# Condensation-dependent interactome of a chromatin remodeler underlies tumor suppressor activities

Yasuhiro Tsukamoto [1,2,3,10] ✉, Atsuki Kawamura[2,13], Ayhan Yurtsever[4,13], Hidefumi Suzuki[5,13], Nichole Marcela Rojas-Chaverra [1], Hiroki Sato[1,11], Daisuke Ino [2,12], Takehiko Ichikawa[4], Weilin Wei[4], Shojiro Haji [3], Dominic Chih-Cheng Voon[6,7], Akinobu Matsumoto [8], Kunio Matsumoto [1,4], Hidehisa Takahashi [5], Noriyuki Kodera [4], Takeshi Fukuma [4], Yoshihiro Ogawa [3] ✉, Masaaki Nishiyama [2,9] & Katsuya Sakai [1,4] ✉

Chromatin remodelers are vital for cellular functions like transcription by modulating nucleosome accessibility. Although biological condensates regulate these processes, the contribution of chromatin remodelers to condensation mechanisms remains poorly understood. Here, we examine the role of the E1321 frameshift mutation in *CHD1*, a chromatin remodeler, which is often targeted in cancers. This mutation truncates CHD1's C-terminus, leading to an oncogenic transcriptome and promoting tumorigenesis. This is due to the loss of an intrinsically disordered region (IDR) crucial for forming CHD1 condensates. These condensates are facilitated by the presence of H3K4me3-modified nucleosomes and RNA, guided to active promoters to regulate gene expression. Furthermore, CHD1 condensates contain long noncoding RNA and histone-modifying proteins, revealing an integral role for CHD1 condensates in epigenetic regulation. Among these components, *MLL* mutations frequently co-occur with *CHD1* mutations in various cancers, suggesting a shared pathway in cancer development. These findings underscore the importance of chromatin remodeler condensation as a regulatory hub in various cellular processes and tumor suppression.

Biomolecular condensates facilitate specific protein-protein interactions at distinct genomic locations through processes like liquid-liquid phase separation[1] or alternative mechanisms[2]. Various nuclear condensates, such as nucleoli, promyelocytic leukemia nuclear bodies, and paraspeckles, play critical roles in genomic functions[3]. Condensates formed by coactivators and transcription factors act as specialized compartments, organizing and concentrating the transcription machinery to establish and maintain cellular identity[4–6]. While chromatin-binding proteins, histone-modified nucleosomes, and RNA are known to contribute to condensate formation[5–8], the role of chromatin remodelers in this process remains poorly understood.

Chromatin dynamics regulate access to the genomic DNA and are crucial for various nuclear processes, including gene transcription during cell fate determination[9,10]. Chromatin remodelers control chromatin reorganization by altering the histone-DNA contacts within the nucleosomes. Studies have shown that chromodomain helicase DNA-binding 1 (CHD1), a member of the CHD family of chromatin remodelers[11–13], is recruited to active gene promoters[14] to regulates gene transcription[15], where it unwinds and opens the chromatin, and

A full list of affiliations appears at the end of the paper. ✉e-mail: ytsukamo@stu.kanazawa-u.ac.jp; ogawa.yoshihiro.828@m.kyushu-u.ac.jp; k_sakai@staff.kanazawa-u.ac.jp

supports RNA polymerase II activity during transcription initiation and elongation[13]. It is also essential for pluripotency in mouse embryonic stem cells[10]. Dysfunctions and mutations in the *CHD1* gene have been linked to cancer and neurodevelopmental disorders[11,16–18]. Notably, the loss of CHD1 function is observed in around 8%–18% of prostate cancer cases, causing widespread alterations in gene expression in a cellular context-specific manner[19,20]. However, the precise mechanisms through which CHD1 abnormalities contribute to cancer development remain unclear.

Here, we showed that the E1321 frameshift (fs) mutation of CHD1, frequently found in cancers, produces a truncated protein lacking its C-terminal intrinsically disordered region (IDR). The truncated CHD1 lost its condensation properties, resulting in oncogenic disruption in transcription and tumorigenesis. Through proximity biotinylation, we revealed that the interactome of CHD1 condensation consists of nascent RNAs, long noncoding RNAs (lncRNAs), and histone-modifying proteins, elucidating its multifaceted involvement in epigenetic processes. Mixed-lineage leukemia (MLL) histone lysine methyltransferase, linked to the CHD1 condensate, was frequently co-mutated with CHD1, which implicates their collaboration in tumorigenesis.

## Results

### E1321fs mutation encodes the C-terminal IDR-truncated CHD1

While *CHD1* is frequently deleted in prostate cancers[19], inactivation in other cancers often occurs through point mutations in the *CHD1* coding region (Fig. 1a). Among these, the E1321fs mutation (*CHD1^E1321fs*) is the most frequently reported in the Cancer Genome Atlas (TCGA) database across diverse cancer types and cell lines (Fig. 1a, b). This mutation is monoallelic in prostate and colon cancer cell lines (22Rv1 and GP5d), with the wild-type (WT) allele retained (Supplementary Fig. 1). Sanger sequencing confirmed the presence of both *CHD1^WT* and *CHD1^E1321fs* transcripts (Fig. 1c). The *CHD1^E1321fs* mutation causes a frameshift thereby introducing a stop codon and the deletion of residues 1342 to 1710 (Fig. 1a, c). To investigate the expression of the mutant protein, we utilized a CRISPR-mediated knock-in strategy to insert the *Venus* sequence, a yellow fluorescent protein, at the mutation site. This resulted in comparable expression levels of CHD1^E1321fs-Venus and CHD1^WT-Venus (Supplementary Fig. 2), suggesting that the mutant protein could avoid mRNA decay.

The truncated C-terminal regions of CHD1 as well as the N-terminal regions are predicted to be IDRs based on analyses using

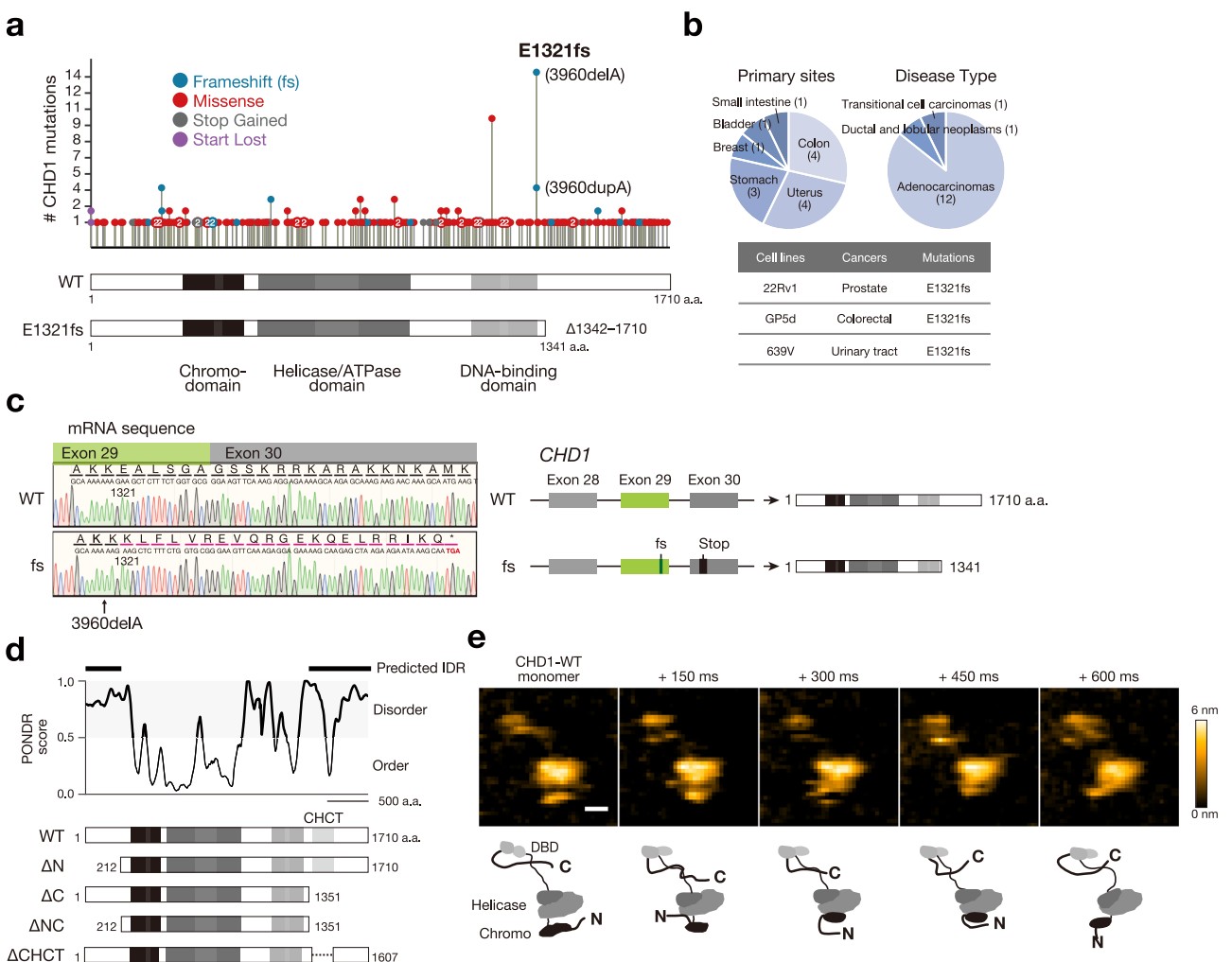

**Fig. 1 | E1321fs mutation encodes the C-terminal IDR-truncated CHD1.**
**a** Schematic showing the somatic mutations in CHD1 across human cancers (TCGA database) with the number of single amino acid mutations. The colored circles specify the mutation types. The functional domains within CHD1 are depicted below. The most prevalent mutation in cancers is a frameshift mutation at E1321 (E1321fs), resulting from adenine deletion or duplication at position 3960. WT: wild-type. **b** Distribution of E1321fs mutation in primary cancer sites and associated cancer types (upper). Identification of cell lines harboring E1321fs mutation (lower).

**c** Heterozygous expression of *CHD1^WT* and *CHD1^E1321fs* mRNA in 22Rv1 cells. *CHD1^E1321fs* mRNA introduces a stop codon at amino acid 1341, resulting in a C-terminus truncated CHD1 protein. **d** PONDR score analysis of CHD1 (upper) and schematic representation of CHD1^WT and domain-deleted variants (lower). **e** HS-AFM captures single-molecule live imaging of CHD1^WT, revealing the presence of IDRs at both the N- and C-terminus. Scale bar, 10 nm. Schematic diagrams depicting the domains of CHD1^WT. Consistent molecular shapes were observed across three independent experiments.

the Predictor of Natural Disordered Regions (PONDR) tool[21] (Fig. 1d). We validated these predictions through direct visualization of each IDR using high-speed atomic force microscopy (HS-AFM)[22], which allows for single-molecule live imaging of recombinant CHD1 protein. Both the N- and C-terminal regions of CHD1 displayed typical IDR characteristics, such as string-like formations, irregular Brownian motion, and a lack of stable folded domains (Fig. 1e, Supplementary Movie 1). Furthermore, the assignment of each domain was supported by HS-AFM experiments using CHD1 variant proteins with specific domain deletions (Supplementary Fig. 3).

## CHD1 forms condensate through its C-terminal IDR

To examine whether CHD1 condensates through its IDR, we conducted in vitro droplet assays (Fig. 2a, Supplementary Fig. 4). CHD1^WT formed spherical condensates, regardless of the fluorescent tag (Supplementary Fig. 4a, b). Formation of these condensates was promoted at higher protein concentrations and was inhibited or disappeared at high salt concentrations (Supplementary Fig. 4c–g, Supplementary Movies 2, 3). Droplet assays using CHD1^WT and IDR deletion variants (ΔN, ΔC, ΔNC) revealed that both N- and C-terminal IDRs promote condensate formation, with the C-terminal IDR being more crucial (Fig. 2a). In vitro nucleosome sliding assays showed that neither IDR was essential for CHD1's nucleosome sliding activity (Supplementary Fig. 5).

In line with in vitro findings, cellular overexpression of exogenous CHD1^WT and variants revealed that both the N- and C-terminal IDRs are necessary for the nuclear condensation of CHD1, with the C-terminal IDR playing a more significant role (Fig. 2b). Fluorescence imaging of HeLa cells expressing CHD1^WT tagged with either Venus or mScarlet at the N- or C-terminus confirmed that the fluorescent tags do not affect CHD1 condensation properties (Supplementary Fig. 6). This condensation depends on the IDRs rather than the Helix C-terminal (CHCT) domain, whose function is unknown[23], as a variant lacking the CHCT domain still exhibited condensation (Fig. 2b). To investigate whether endogenous CHD1^WT condensates within cells, we performed genetic knock-in of Venus into the C-terminal of CHD1 locus in HeLa cells (Fig. 2c). The CHD1^WT-Venus protein formed small nuclear condensates and, in some cells, also larger nuclear condensates (Fig. 2d). As observed for other nuclear proteins[24], we observed that CHD1 forms large nucleolar condensates in response to heat shock, which gradually dissipate over the course of several hours (Supplementary Fig. 7a–d). Since heat shock is known to induce solid-like condensates[2], we assessed the dynamics of CHD1^WT-Venus condensates formed under native and heat-shock conditions using fluorescence recovery after photobleaching (FRAP). Native CHD1 condensates displayed rapid recovery within seconds (Fig. 2e, Supplementary Fig. 7e, f, Supplementary Movie 4), whereas heat shock-induced CHD1 condensates showed no fluorescence recovery over several minutes (Supplementary Fig. 7e, f). These observations showed that native and heat shock-induced CHD1 condensates have distinct biochemical properties, and may serve different functions.

Due to the lack of specific antibodies for the CHD1^E1321fs protein variant, we could not investigate the condensation patterns of endogenous CHD1^E1321fs via immunostaining. Instead, we evaluated the fluorescence patterns in HeLa cells with knock-in of either CHD1^E1321fs-Venus or CHD1^WT-Venus (Supplementary Fig. 2). The CHD1^E1321fs–Venus showed a more uniform nuclear distribution compared to the more condensed patterns of CHD1^WT–Venus (Supplementary Fig. 8), consistent with exogenous protein experiments.

## H3K4me3 nucleosome and RNA stimulates CHD1 condensation

Histone marks can lead to chromosome compartmentalization by promoting phase separation[8,25]. To investigate the functional relevance of the IDR, we tested whether H3K4me3-modified nucleosomes induce CHD1 condensation, as CHD1 binds specifically to H3K4me3[26,27]. We conducted droplet assays by titrating DNA, nucleosomes, H3K4me3-, H3K27ac-, and H3K9me3-modified nucleosomes against CHD1. The results indicated that H3K4me3 nucleosomes significantly increased both the number and size of CHD1 condensates, while others had minimal effects (Fig. 3a, Supplementary Fig. 9). The slight increase in CHD1 condensation with H3K9me3-modified nucleosomes may be attributed to potential interactions between chromodomains and H3K9me3[28].

Recent studies have shown that nascent RNAs at active promoters regulate transcriptional condensate formation[6,29]. Given CHD1's role at gene promoters in transcription[13,30], we tested whether RNA could enhance CHD1 condensation by titrating EF1α mRNA against CHD1. Droplet assays revealed that RNA not only promoted CHD1 condensation but was also incorporated into the condensates (Fig. 3b, Supplementary Fig. 10). High-resolution AFM analysis showed that CHD1 condensates formed thin disk-like structures with fluctuating central protrusions (Fig. 3c, left), while RNA presence resulted in spherical condensates with heights of 50–100 nm (Fig. 3c, right, Supplementary Fig. 11).

To explore the roles of H3K4me3 nucleosomes and RNA in CHD1 condensation in cells, we treated HeLa cells with RNA polymerase II inhibitors and a methyltransferase inhibitor. α–Amanitin, an RNA polymerase II inhibitor, reduced CHD1 condensation in the nucleus without altering CHD1 protein levels (Supplementary Fig. 12a–h). Actinomycin D, an RNA polymerase I and II inhibitor, significantly decreased cellular RNA levels and reduced CHD1 condensation, accompanied by a marked reduction in both H3K4me3 and CHD1 protein levels, although the underlying mechanism remains unclear (Supplementary Fig. 12a–h). Sinefungin, a methyltransferase inhibitor, decreases the trimethylation level at H3K4 and diminished CHD1 condensation (Supplementary Fig. 12i–k). Collectively, these results suggest that RNA and H3K4me3-modified nucleosomes promote and stabilize CHD1 condensates in cells.

## IDR is necessary to recruit CHD1 to the promoters of tumor suppressors

CHD1 is recruited to transcription start sites (TSS) and intragenic and intergenic enhancer-like sites through transcription-coupled mechanisms[14,31]. We investigated how CHD1 condensation affects its genomic distribution by performing ChIP-seq analysis on CHD1 knockout mCAT-HeLa cells that were reintroduced with either CHD1^WT or CHD1^ΔNC. The results showed that CHD1^WT was predominantly localized to active promoters, consistent with previous reports[14,31] (Fig. 4a, b, Supplementary Fig. 13). In contrast, while the binding of CHD1^ΔNC to TSS was reduced (Fig. 4a, b, Supplementary Fig. 13), it showed enhanced binding with noncoding regions, such as intergenic and intron regions (Fig. 4a–c). Notably, CHD1^ΔNC exhibited reduced binding at promoters of tumor suppressor genes, such as TP53 and cyclin-dependent kinase inhibitor 1B (CDKN1B) (Fig. 4d), leading to decreased mRNA expression (Fig. 4e).

To identify the nucleosome-free regions in CHD1^WT or CHD1^ΔNC mCAT-HeLa cells, we performed ATAC-seq (Supplementary Fig. 14). This showed 5083 more open and 2455 more closed chromatin peaks in CHD1^ΔNC cells compared to CHD1^WT (Supplementary Fig. 14a). Genomic Regions Enrichment of Annotation Tool (GREAT) analysis[32,33] indicated that closed chromatin in CHD1^ΔNC is associated with apoptotic pathways, while open chromatin is associated with metabolic pathways (Supplementary Fig. 14b). This suggests that CHD1^ΔNC may have a positive effect on cell growth. However, no significant differences in genome-wide promoter chromatin accessibility were observed (Supplementary Fig. 14c). Rather, the biding sites revealed in our earlier ChIP-seq analysis for CHD1^WT or CHD1^ΔNC positively correlated with chromatin accessibility at individual genes (Supplementary Fig. 14d, e).

Prompted by the reduced recruitment of CHD1^ΔNC at CDKN1B promoter, we assessed the effect of IDR deletion on cellular

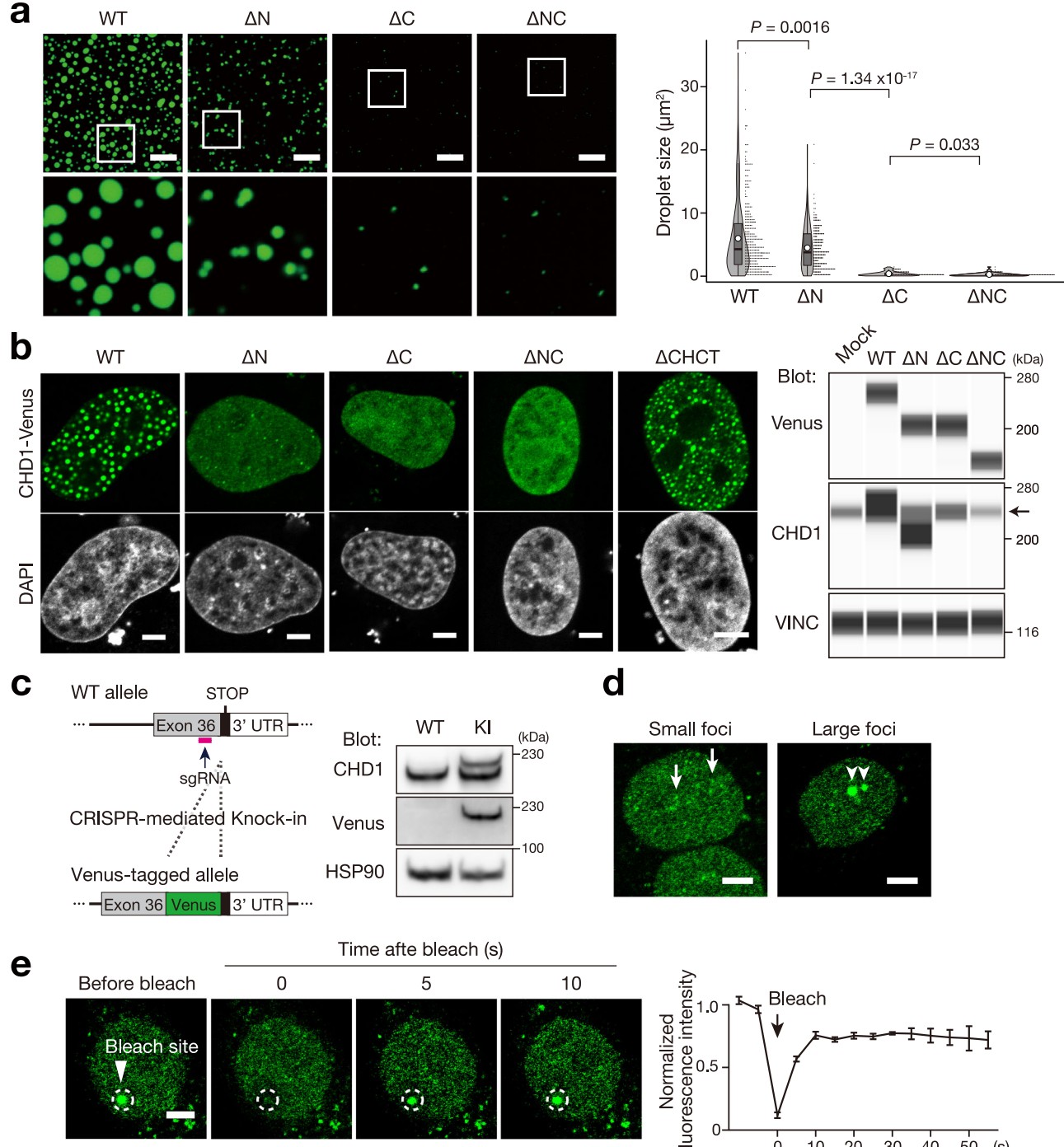

**Fig. 2 | CHD1 forms condensates through the C-terminal IDR. a** Droplet assay of CHD1–Venus proteins (1 μM) highlighting the crucial role of the C-terminal IDR in condensation. Scale bars, 20 μm. Condensates within the $1.125 \times 10^{-2}$ mm² area, obtained from three independently prepared dishes, were analyzed and visualized using violin plots with boxplots, illustrating the mean, median, interquartile range, and 10th–90th percentiles denoted by white circles, black bars, grey boxes, and thin black lines, respectively. The size of each condensate was plotted as a dot on the right and analyzed using unpaired two tailed Student's t-test. **b** Confocal images of HeLa cells expressing exogenous CHD1–Venus, demonstrating that the C-terminus IDR is crucial for nuclear condensate formation (left). Fluorescence localization patterns were consistent across three independent experiments. Scale bars, 5 μm. Western blot analysis performed using the Wes system (ProteinSimple), with anti-Vinculin (VINC) as the loading control (right). The CHD1-Venus variants were expressed at comparable levels to each other, and all exhibit elevated expression

compared to endogenous CHD1 (indicated by the arrow). Note that the CHD1 antibody targets a C-terminal epitope, meaning it does not detect the CHD1^ΔC or CHD1^ΔNC variants. **c** Schematic illustrating the strategy for CHD1^WT–Venus knock-in (KI) HeLa cell generation (left). Western blot analysis of CHD1 and Venus in CHD1^WT–Venus KI HeLa cells, with anti-HSP90 used as loading control (right). KI band was consistently observed in three independent Western blot experiments. **d** Confocal images of CHD1^WT–Venus KI HeLa cells. Arrows indicate small condensates, while arrowheads indicate large condensates (> 1 μm in diameter). Small and large condensates are observed within the nucleoplasm and nucleolus, respectively. Scale bars, 5 μm. **e** Confocal images of Spot-FRAP analysis on CHD1^WT–Venus KI HeLa cells (left panel). Scale bar, 5 μm. FRAP signal in the bleached area before and after bleaching (right panel). Data are presented as the mean ± SEM, $n = 3$ independent experiments. Source data are provided as a Source Data file.

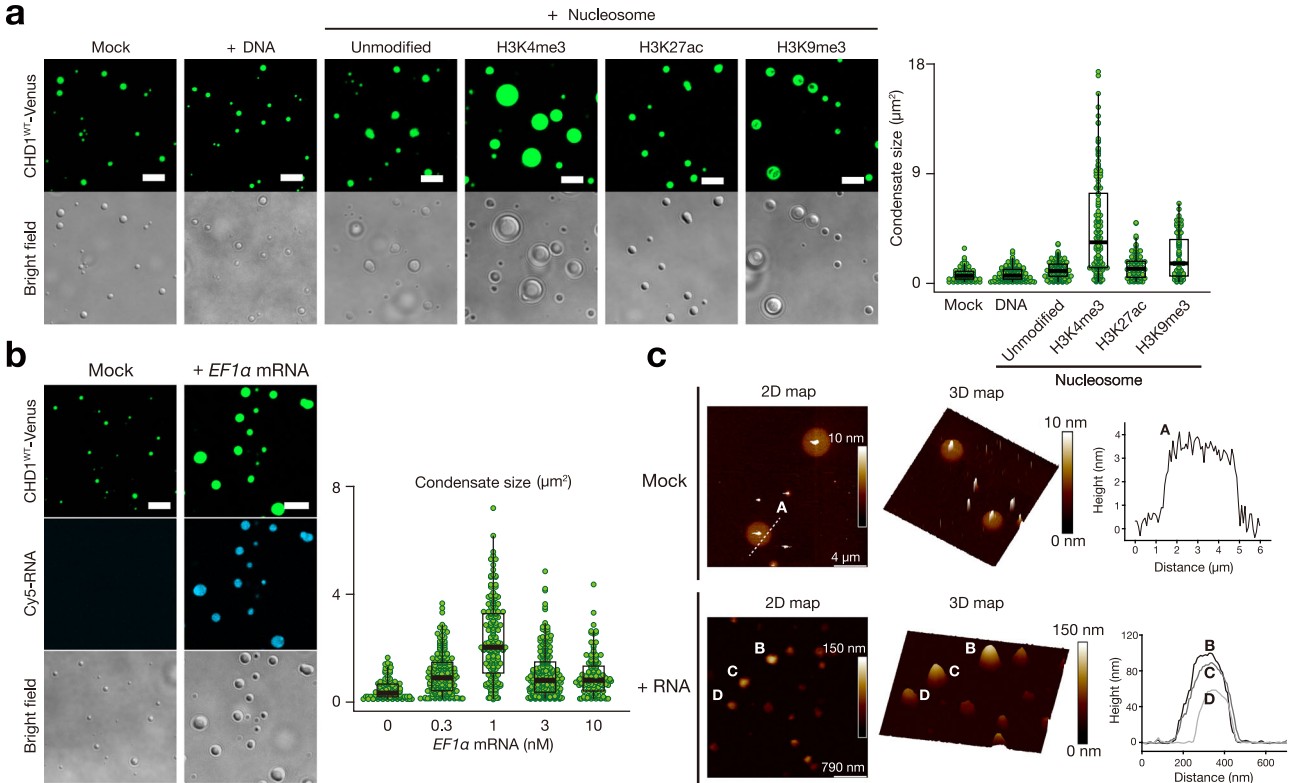

**Fig. 3 | H3K4me3 nucleosome and RNA stimulates CHD1 condensation.**
**a** Droplet assay demonstrating that H3K4me3-modified nucleosomes promote CHD1 condensation. CHD1$^{WT}$–Venus at 30 nM was mixed with 100 nM of Widom 601 DNA, unmodified nucleosomes, H3K4me3-, H3K27ac-, or H3K9me3-modified nucleosomes (left). Scale bars, 5 μm. Condensates within the $1.03 \times 10^{-2}$ mm$^2$ area, obtained from three independently prepared dishes, were represented using boxplots, illustrating the median, interquartile range, and 10$^{th}$–90$^{th}$ percentiles denoted by black bars, white boxes and thin black lines, respectively. The size of each condensate was plotted as a green dot (right). **b** Droplet assay demonstrating RNA-induced condensation of CHD1$^{WT}$ and its incorporation into the condensates. 30 nM of CHD1$^{WT}$–Venus was mixed with 5 mM Tris buffer (Mock) or 1 nM of Cy5-tagged synthetic EF1α RNA (left). Scale bars, 5 μm. Condensates at the indicated

concentration of RNA within the $1.03 \times 10^{-2}$ mm$^2$ area, obtained from three independently prepared dishes, were represented using boxplots with dotplots, illustrating the median, interquartile range, and 10$^{th}$–90$^{th}$ percentiles denoted by black bars, white boxes and thin black lines, respectively (right). **c** Effect of RNA on condensate morphology. High-resolution AFM overview image of CHD1$^{WT}$ protein condensates in the absence of RNA (left), revealing a thin disc-shaped structure with a fluctuating protrusion at the center. The height profile taken along the dashed line. Addition of RNA (right) resulted in the formation of spherical CHD1$^{WT}$ condensates. Height profiles taken across the CHD1$^{WT}$ condensates, indicating heights in the range of 50–100 nm. The images represent results from two independent experiments, which yielded consistent outcomes. Source data are provided as a Source Data file.

proliferation using CHD1 knockout mCAT-HeLa cells with reintroduced CHD1$^{WT}$ or truncated variants (CHD1$^{\Delta N}$, CHD1$^{\Delta C}$, and CHD1$^{\Delta NC}$) (Supplementary Fig. 15a). CHD1 knockout cells showed increased proliferation compared to control treated with a nontargeting single-guide RNA (sgNTC) (Supplementary Fig. 15b). Restoration of either CHD1$^{WT}$ or CHD1$^{\Delta N}$ normalized proliferation, while that of CHD1$^{\Delta C}$ and CHD1$^{\Delta NC}$ did not (Supplementary Fig. 15b), indicating that the C-terminal IDR of CHD1 is crucial for growth suppression. Gene set enrichment analyses (GSEA)[34] indicated that the p53 pathway was downregulated in CHD1 knockout cells and only upregulated by CHD1$^{WT}$ restoration, not by CHD1$^{\Delta NC}$ (Supplementary Fig. 15c). This explains partially why CHD1$^{WT}$, but not CHD1$^{\Delta NC}$, reduced cell proliferation.

Although the chromodomain of CHD1 is responsible for binding to H3K4me2/3 at active promoters[26,27], our ChIP-seq results indicate that the IDRs are also essential for its interaction with active promoters. To better understand the biochemical mechanism, we conducted nucleosome binding assays using CHD1$^{WT}$ and CHD1$^{\Delta NC}$. Interestingly, CHD1$^{\Delta NC}$ exhibited significantly stronger nucleosome binding than CHD1$^{WT}$ (Fig. 5a, dissociation constant ($K_D$): 6.3 nM vs. 221 nM), suggesting that the IDRs of CHD1 negatively regulate its affinity toward nucleosomes. CHD1 is known to bind to nucleosomes through its helicase/ATP domain[12]. Consistent with this, the CHD1 variant lacking this helicase/ATP domain failed to bind to nucleosomes

and formed large nuclear condensates (Fig. 5b). These results suggest that IDRs reduce CHD1's affinity for nucleosomes, enhancing selective binding to H3K4me2/3-modified nucleosomes and active promoters via its chromodomain (Fig. 5c). In contrast, the higher affinity of CHD1$^{\Delta NC}$ for nucleosomes may lead to reduced selectivity for H3K4me2/3 modifications and mis-targeting, potentially explaining its recruitment to intergenic or intronic regions (Fig. 4a–c).

## The C-terminal IDR is essential for tumor suppression

To investigate the effect of C-terminal truncation on tumorigenesis, we used CRISPR-Cas9 system to skip exon 29 (Δex29) in 22Rv1 cells (22Rv1$^{\Delta ex29/\Delta ex29}$), which restored the 1342–1671 region of CHD1 (Fig. 6a, b). This restoration recovered WT-like condensation properties as shown in HeLa cells expressing CHD1Δex29-mScarlet, a red fluorescent protein (Fig. 6c). Tumorigenicity was evaluated by transplanting these cells subcutaneously into nude mice and comparing them with a control line (22Rv1$^{WT/fs}$) treated with a nontargeting single-guide RNA. The 22Rv1$^{\Delta ex29/\Delta ex29}$ cells showed significantly reduced tumor formation compared to the 22Rv1$^{WT/fs}$ cells (Fig. 6d, Supplementary Fig. 16a, b), indicating the C-terminal of CHD1 has tumor-suppressive effects. Immunohistochemistry showed that 22Rv1$^{\Delta ex29/\Delta ex29}$ tumors had lower proliferation and fewer epithelial-mesenchymal transition (EMT)-related proteins (ITGA5, TWIST1/2, SNAI2) compared to 22Rv1$^{WT/fs}$ tumors (Supplementary Fig. 16c–f).

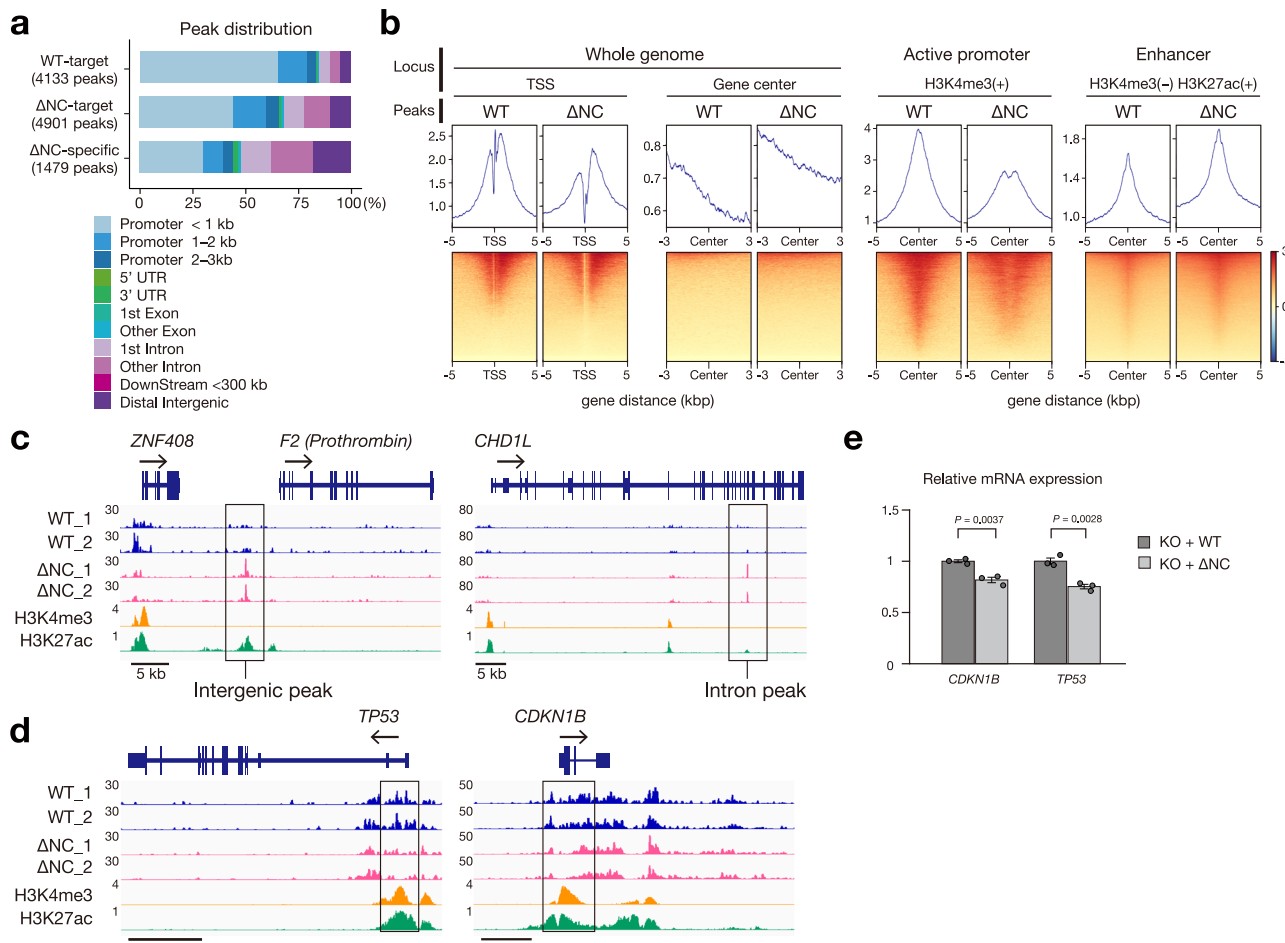

**Fig. 4 | IDR is necessary to recruit CHD1 to active promoters. a** ChIP-seq signals in CHD1$^{WT}$ or CHD1$^{\Delta NC}$-reconstituted CHD1 knockout mCAT-HeLa cells. The distributions of ChIP-seq peak according to the corresponding genomic regions. WT-target: peaks overlapping in CHD1$^{WT}$ samples ($n = 2$). ΔNC-target: peaks overlapping in CHD1$^{\Delta NC}$ samples ($n = 2$). ΔNC-specific: ΔNC target peaks without WT peaks. **b** Heatmap displaying ChIP-seq read densities around transcription start sites (TSS) and gene centers of the entire genome (left), TSS at active promoters (H3K4me3-positive, middle), and unmodified promoter (H3K4me3-negative and H3K27ac-positive, right). CHD1$^{WT}$ peaks were predominantly found at TSS, especially at H3K4me3-positive promoters. In contrast, CHD1$^{\Delta NC}$ peaks displayed a distinct pattern compared to CHD1$^{WT}$, showing a higher tendency to gene body and lower specificity for H3K4me3-positive TSS. **c, d** ChIP-seq binding profiles of WT, ΔNC, H3K4me3, and H3K27ac on intergenic and intronic ΔNC-specific sites (boxed) (**c**) and the promoters of tumor suppressor genes [note the reduced ΔNC ChIP-seq signals at the TSS (boxed) of these gene loci] (**d**). **e** RT-qPCR analysis of *CDKN1B* and *TP53* expression levels normalized to *GAPDH*. Data represent mean ± SEM of $n = 3$ biologically independent samples. Unpaired two-tailed Student's t-test. Source data are provided as a Source Data file.

To determine the transcriptomic changes caused by CHD1$^{E1321fs}$, RNA-seq was conducted on 22Rv1$^{\Delta ex29/\Delta ex29}$ and 22Rv1$^{WT/fs}$ cells. Differentially expressed gene (DEG) analysis showed 432 upregulated and 592 downregulated genes in 22Rv1$^{\Delta ex29/\Delta ex29}$ cells compared to 22Rv1$^{WT/fs}$ cells (Supplementary Fig. 17a–c). Pathway analysis and GSEA highlighted increased expression of p53 pathway components (*BAX*, *DDB*) and tumor suppressors (*CDH17*, *CDKN1A*), while genes related to cell proliferation (*MYC*, *AR*, *EGFR*, *FGFR1*, *IGF1*, *IGF1R*, *IRS2*) and EMT (*SNAI2*, *IGFBP*, *WNT5A*) were downregulated in 22Rv1$^{\Delta ex29/\Delta ex29}$ cells (Fig. 6e, f, Supplementary Fig. 17d). *SP1* and *TP53*-related transcription factors were also higher in 22Rv1$^{\Delta ex29/\Delta ex29}$ cells (Supplementary Fig. 17e). These changes aligned with in vivo results (Supplementary Fig. 16c–f) and were associated with reduced proliferation (Fig. 6g), G1 arrest (Fig. 6h, Supplementary Fig. 18), and lower collagen matrix penetrability (Fig. 6i) in 22Rv1$^{\Delta ex29/\Delta ex29}$ cells.

The deletion of exon 29 leads to the loss of 39 amino acids, 28 of which belong to the DNA-binding domain (Fig. 6a). To clarify whether the CHD1$^{WT}$ and CHD1$^{\Delta ex29}$ exhibit similar functions, we compared the effects of reintroducing CHD1$^{WT}$ and CHD1$^{\Delta ex29}$ on proliferation and gene expression in CHD1 knockout 22Rv1 cells (Supplementary Fig. 19). Both variants similarly inhibited cell growth (Supplementary Fig. 19b) and

upregulated *CDKN1A*, *CDNK1B*, and *BAX* (Supplementary Fig. 19c). These results suggest that the partial loss of the DNA-binding domain does not significantly affect these cellular outcomes.

Collectively, these data suggest that CHD1$^{E1321fs}$ promotes oncogenic gene expression to drive cell proliferation and invasion.

## CHD1 condensates interact with nascent RNAs and lncRNAs

We analyzed CHD1 condensate composition to understand their function. Using antibody-based in situ biotinylation[35,36] with anti-CHD1 antibodies in HeLa cells (Fig. 7a), we identified 6763 RNAs enriched more than fourfold in anti-CHD1 samples compared to IgG controls (Fig. 7b). Of these RNAs, 72.5% were protein-coding (Fig. 7b), with 69.1% corresponding to CHD1 target genes from ChIP-Atlas (Fig. 7c). These results indicate that CHD1 condensates are near transcriptionally active sites and nascent RNA.

Approximately 24.7% of RNAs in proximity to CHD1 were lncRNAs (Fig. 7b), but almost none of them were CHD1 target genes, unlike the coding RNAs (Fig. 7c). GSEA with LncSEA2.0[37] showed that 36% of these lncRNAs were previously shown to interact with CHD1 by RNA immunoprecipitation[38] (Fig. 7d). Additionally, 38.3% of the CHD1-proximal lncRNA can interact with RNA polymerase II subunit A

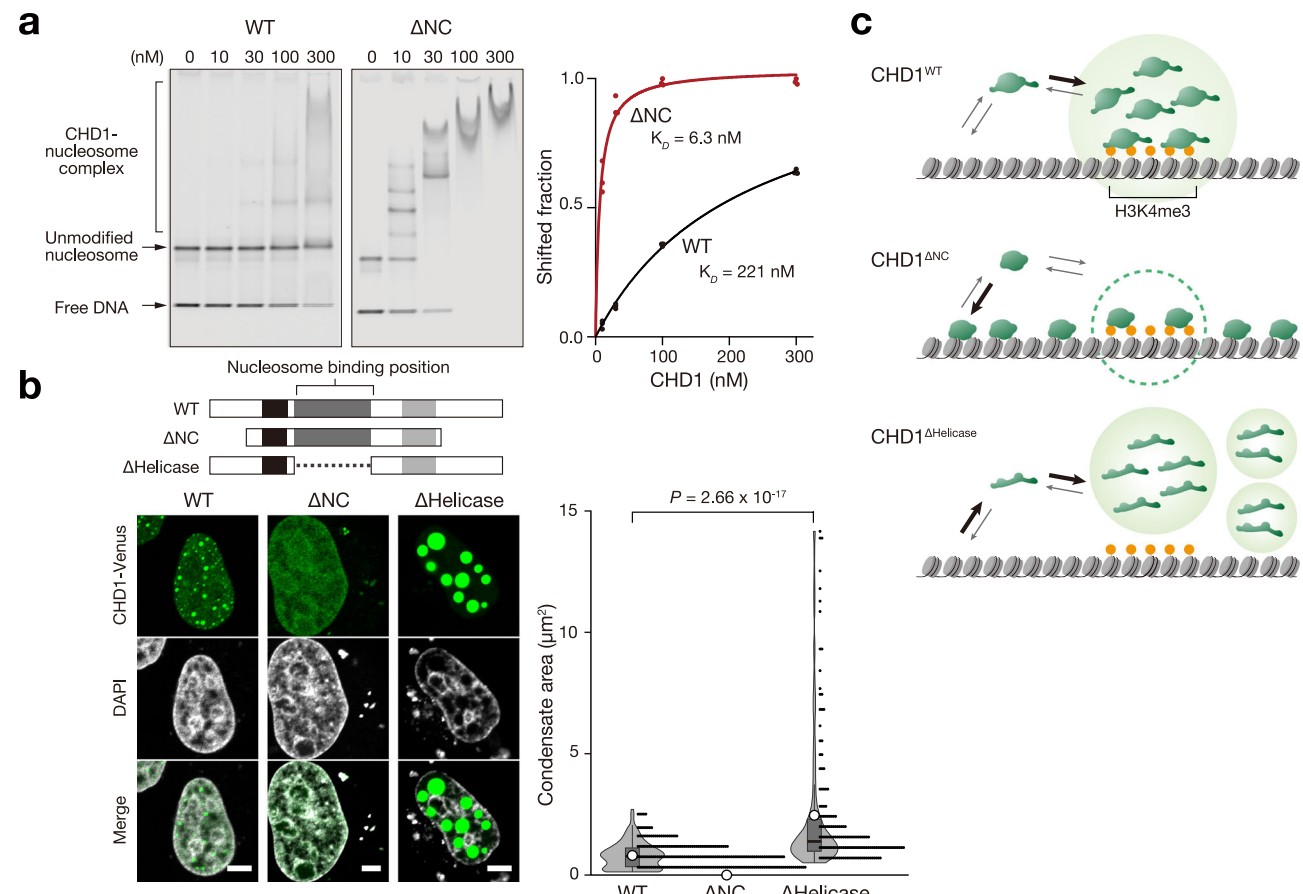

**Fig. 5 | IDR inhibits CHD1-binding to unmodified nucleosome. a** Representative images of nucleosome binding assay showing stronger binding of CHD1$^{\Delta NC}$ protein to nucleosomes compared to CHD1$^{WT}$. The concentration of indicated CHD1 was tested against 5 nM nucleosomes containing IR dye-labeled DNA, separated by native PAGE, and visualized via IR signal detection. Shifted nucleosome fractions ($n = 3$ independent experiments) were fitted to a dose-response curve (right). The curves were compared using two-way ANOVA ($P < 0.0001$). **b** Schematics illustrating the structure of CHD1$^{WT}$, CHD1$^{\Delta NC}$, and CHD1$^{\Delta Helicase}$. Confocal images of HeLa cells expressing WT, ΔNC, and ΔHelicase show negative correlation between CHD1 condensation and its affinity to nucleosomes. Scale bars, 5 μm. Condensate sizes in cells from 10 random fields are represented using violin plots with boxplots, illustrating the mean, median, interquartile range, and $10^{th}$–$90^{th}$ percentiles denoted by white circles, black bars, grey boxes and thin black lines, respectively. The size of each condensate was plotted as a dot on the right and analyzed using unpaired two-tailed Student's t-test. **c** Schematic diagrams showing the nucleosome-binding properties of CHD1$^{WT}$, CHD1$^{\Delta NC}$, and CHD1$^{\Delta Helicase}$. CHD1$^{WT}$ possesses IDRs, which inhibit non-specific binding to unmodified nucleosomes. Condensation driven by the IDR occurs around H3K4me3-nucleosomes, elevating local concentration and ensuring specificity towards H3K4me3-modified regions. CHD1$^{\Delta NC}$ exhibits reduced condensation ability, leading to non-specific binding to nucleosomes through the helicase/ATPase domain. CHD1$^{\Delta Helicase}$ cannot bind to nucleosomes and forms large condensates. Source data are provided as a Source Data file.

(POLR2A) (Fig. 7d), supporting CHD1's role in chromatin remodeling at active transcription sites[10,14,31]. Moreover, 14.8%–38.5% of the CHD1-proximal lncRNAs interacted with diverse histone modifications, including both active and repressive marks (Fig. 7e). In the context of cancer biology, some CHD1-proximal lncRNAs have been linked to microRNA and androgenic receptor (AR) mRNA (Fig. 7f), which might participate in gene repression and prostate cancer, respectively. Additionally, certain lncRNAs were linked to cancer-related processes like metastasis, EMT, cell growth, and the p53 pathway (Fig. 7d, g). These findings suggest a potential functional interaction between lncRNAs and CHD1 condensates.

### IDR-dependent interaction with histone-modifying proteins
To identify proteins interacting with CHD1 in an IDR-dependent manner, we used antibody-based in situ biotinylation with an anti-GFP antibody on CHD1$^{WT}$/CHD1$^{\Delta NC}$-Venus restored HeLa cells, and control HeLa cells (Fig. 8a, b, Supplementary Fig. 20a, b). Compared to control cells, 132 and 137 proteins were significantly enriched in the CHD1$^{WT}$ and CHD1$^{\Delta NC}$-restored cells, respectively (Supplementary Fig. 20a, b). Of these, 48 proteins were specifically enriched by CHD1$^{WT}$ and not by

CHD1$^{\Delta NC}$ (Fig. 8b). Analysis with STRING[39] showed these proteins are involved in histone modification and chromosome organization, including both gene activators (ASH2L, EP400) and repressors (SUZ12, EHMT1, SDS3, GATAD2A) (Fig. 8c).

ChIP-Atlas analysis demonstrated significant overlap in the target genes of CHD1, ASH2L, and SUZ12 (Supplementary Fig. 20c). Immunofluorescence staining of endogenous ASH2L and SUZ12 revealed that they partly colocalized with CHD1 nuclear condensates (Fig. 8d). In CHD1KO mCAT HeLa cells, exogenously expressed CHD1$^{WT}$ exhibited a greater degree of colocalization with SUZ12 compared to CHD1$^{\Delta NC}$ (Supplementary Fig. 21a, b). Similarly, proximity ligation assay (PLA) signals between CHD1 and SUZ12 were significantly stronger with CHD1$^{WT}$ than with CHD1$^{\Delta NC}$ (Supplementary Fig. 21c, d). These results suggest a potential functional interaction between CHD1 and SUZ12 in the context of IDR-mediated CHD1 condensation.

ASH2L is a core component of the MLL family of histone methyltransferases (MLL1–4, or KMT2A–D). Since MLL is frequently targeted in diverged malignancies including leukemia and solid tumors[40], and MLL4/KMT2D forms nuclear condensates[41], we

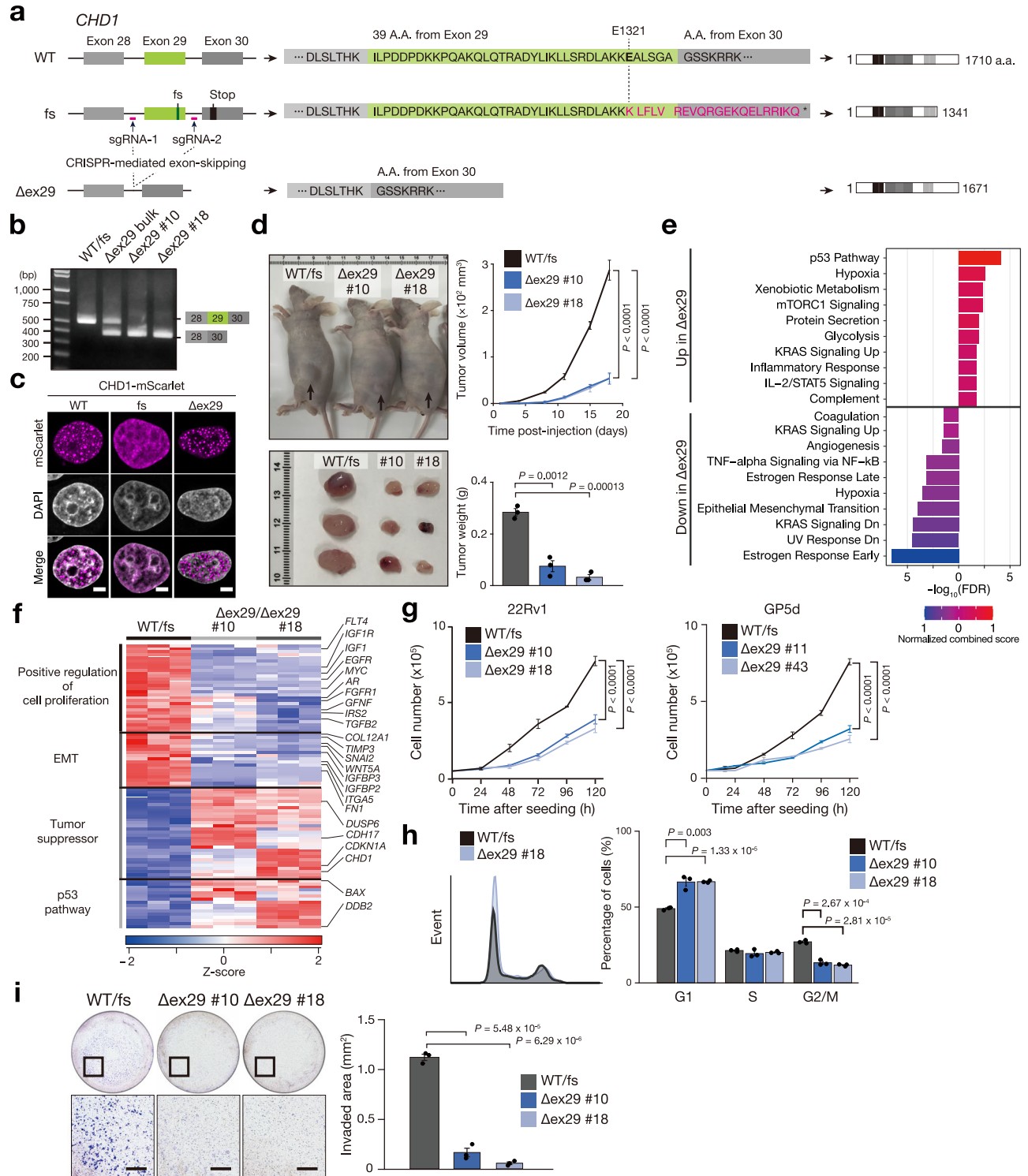

hypothesized a potential functional link between CHD1 and the MLL complex in carcinogenesis. Our analysis of TCGA datasets revealed a notable pattern of co-occurrence of *KMT* and *CHD1* mutations (Supplementary Fig. 22). Specifically, *KMT* mutations were more frequent in cancers with *CHD1* mutations, especially *CHD1*[E1321fs], compared to the general cancer population (Supplementary Fig. 22a). No significant difference was observed in *TP53* mutation frequency between cancers with or without *CHD1* mutations (Supplementary Fig. 22a). Reciprocally, *CHD1* mutations were more common in cancers with KMT mutations compared to those with *TP53* mutations or the overall cancer population (Supplementary Fig. 22b). These data point to a

potential functional compensation between CHD1 and MLL in their roles as tumor suppressors.

## Discussion

This study elucidates the role of IDRs in CHD1-mediated condensation and highlights their importance in tumor suppression (Fig. 9). The prediction of IDRs in CHD family proteins and the identification of disease-associated mutations in these regions suggested the involvement of condensation in the diverse functions of CHD proteins[1]. Our findings represent the evidence that CHD1 forms dynamic condensates through its C-terminal IDR, a process essential for its genome-wide

**Fig. 6 | The C-terminal IDR is essential for tumor suppression. a** Schematic of CRISPR/Cas9-mediated exon 29 skipping to restore the C–terminus of CHD1. A deletion of exon 29 results in the loss of 39 amino acids, 28 of which are part of the DNA-binding domain. **b** RT-PCR analysis of *CHD1* mRNA exhibiting exon 29 skipping in 22Rv1 cells, with consistent results across three independent experiments. WT/fs: parental 22Rv1 cells harboring wild-type and frameshift (fs) alleles. Δex29: 22Rv1 cells treated with CRISPR/Cas9-mediated exon 29 skipping. **c**, Confocal images of HeLa cells expressing exogenous CHD1WT–, CHD1E1321fs–, or CHD1Δex29–mScarlet, demonstrating nuclear condensation of CHD1WT and CHDΔex29, but not CHD1E1321fs. Fluorescence localization patterns were consistent across three independent experiments. Scale bars, 5 μm. **d** Subcutaneous tumor growth of WT/fs 22Rv1 cells with sgRNA nontargeting control and Δex29/Δex29 clones in athymic nude mice (*n* = 3). The right panel presents tumor volume (upper) and weight (lower) as the mean ± SEM from three independent experiments. Tumor volume and weight were analyzed using two-way ANOVA and unpaired two-tailed Student's t-test, respectively. **e** Enrichment analyses performed using Enrichr on DEGs. The

top 10 enriched MSigDB Hallmark 2020 gene sets depict up-regulated or down-regulated DEGs in Δex29/Δex29 clones compared to WT/fs cells. **f** Heat map displaying the gene expression profiles in WT/fs cells and Δex29/Δex29 clones. Tumor suppressor and p53 pathway genes were upregulated, while cell proliferation and EMT-related genes were downregulated. Gene categories and selected genes are indicated on the left and right, respectively. **g** The growth curves of WT/fs cells and Δex29/Δex29 clones were measured in 22Rv1 (left) and GP5d cells (right). Mean ± SEM (*n* = 3 independent experiments); two-way ANOVA. **h** A representative histogram illustrating the cell cycle analysis in WT/fs cells and Δex29/Δex29 clones (left) and phase quantification (right). Mean ± SEM (*n* = 3 independent experiment); unpaired two-tailed Student's t-test. **i** Representative micrographs displaying invaded cells in the transwell invasion assay (left) and the quantitative measurement of the invaded area (right). Scale bars, 500 μm. Mean ± SEM (*n* = 3 independent experiments); unpaired two-tailed Student's t-test. Source data are provided as a Source Data file.

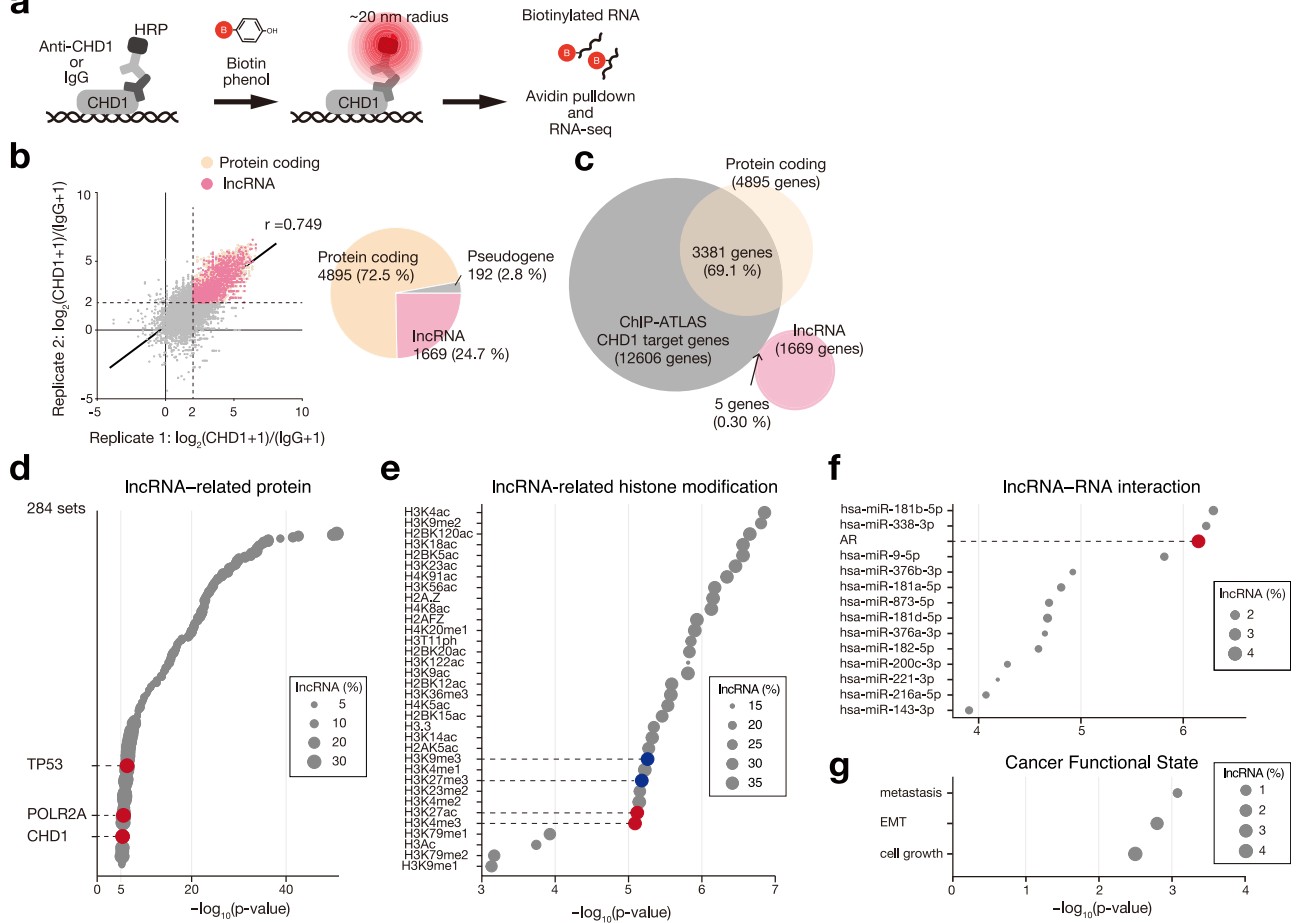

**Fig. 7 | CHD1 condensates interact with lncRNAs associated with epigenetics and cancer. a** Identification of RNAs in the CHD1 condensates using the antibody-based in situ biotinylation. After fixation, permeabilization, and antibody recognition, HRP-conjugated to the second antibody biotinylates RNAs within a ~ 20 nm radius, followed by RNA-seq. **b–g** Analyses of RNAs in the CHD1 condensates. **b** Scatter plot of biotinylation RNA-seq, comparing the log₂ read-count of anti-CHD1 antibody to IgG control in HeLa cells (left). CHD1-proximal RNAs were determined by log₂CHD1/IgG > 2, and their classification is shown in a pie chart

(right). **c** Venn diagram illustrating the overlap between potential CHD1 target genes in ChIP-ATLAS and CHD1-proximal RNAs, revealing that almost all CHD1-proximal lncRNAs are distinct from potential CHD1 target genes. **d–g** Gene set enrichment analysis of CHD1-proximal lncRNAs in terms of lncRNA-protein interaction (**d**), lncRNA-histone modification (**e**), lncRNA-RNA interaction (**f**), and cancer functional state (**g**) using LncSEA2.0. Statistical significance was determined by hypergeometric test with FDR-adjusted *P*-values. Source data are provided as a Source Data file.

targeting, interaction with partner proteins, and regulation of gene expression. Furthermore, our results demonstrate that the loss of the C-terminal IDR due to the E1321fs mutation, which is recurrent in cancers, disrupts these functions and contributes to tumorigenesis.

Other chromatin remodelers such as the SWI/SNF complex have been shown to participate in biological condensates formed by the FUS oncofusion protein[42]. Recent studies have highlighted that the IDRs within the ARID1A/B subunits of the SWI/SNF complex are

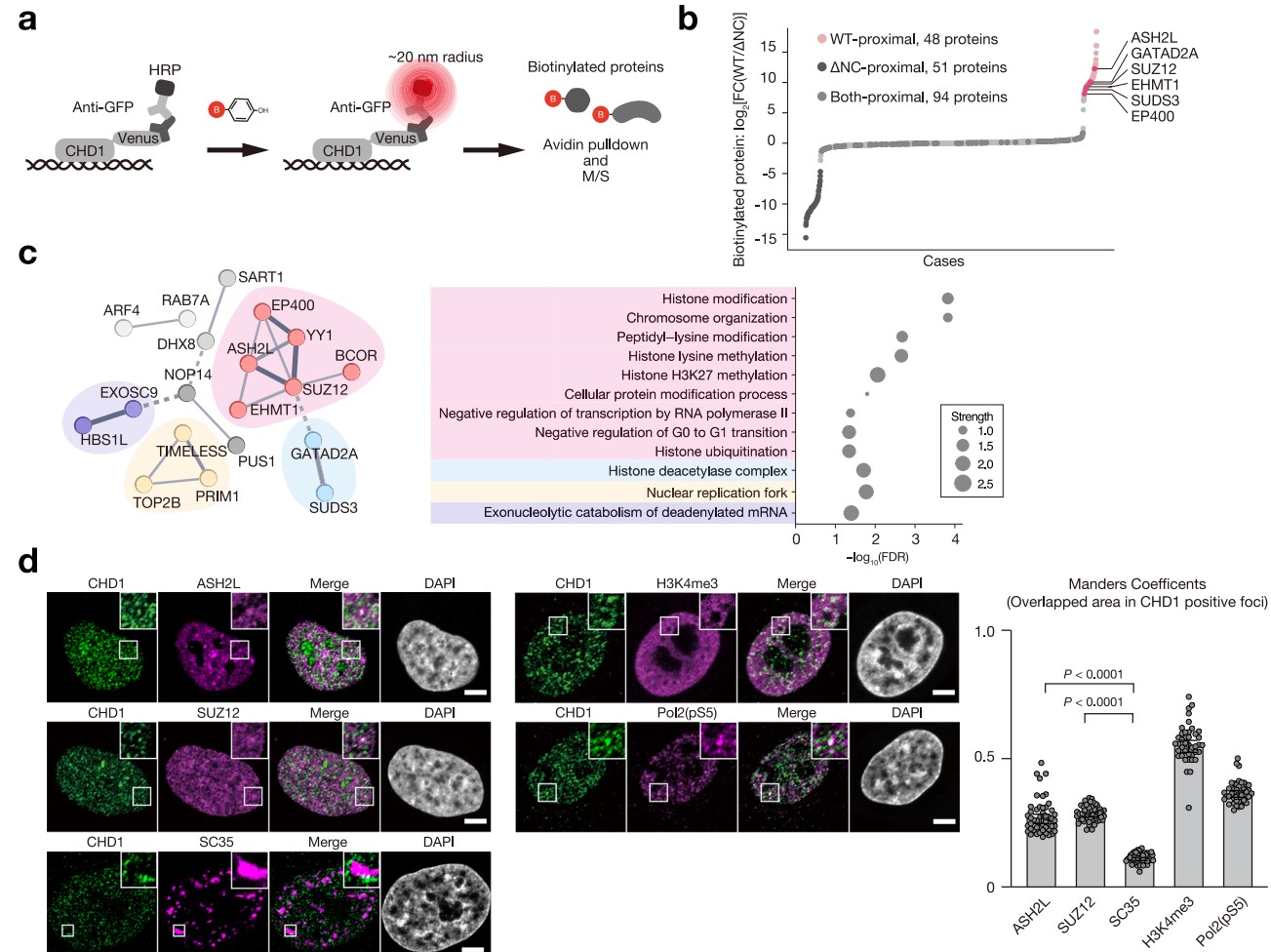

**Fig. 8 | CHD1 condensates interact with active and repressive histone-modifying proteins. a** Identification of proteins in the CHD1 condensates using the antibody-based in situ biotinylation. After fixation, permeabilization, and antibody recognition, HRP-conjugated to the second antibody biotinylates proteins within a ~ 20 nm radius, followed by LC-M/S. **b** Scatter plot of biotinylation M/S, comparing CHD1[WT] LFQ intensity to CHD1[ΔNC] LFQ intensity. The WT- or ΔNC-unique proteins are defined by log$_2$WT/ΔNC > |4| and plotted in pink and dark grey, respectively. The selected proteins are indicated on the right. The other common proteins are plotted in grey. **c** Cluster analysis of WT-unique proximal proteins (histones excluded) using STRING (left). Each color represents a cluster using MCL clustering and line thickness indicates the strength of data support. The cluster's

Gene Ontology terms are displayed as bubble plots (right), indicating the presence of active and repressive histone modifiers among the WT-unique proximal proteins. **d** Confocal images showing the colocalization of CHD1 and ASH2L, SUZ12, SC35, H3K4me3, or RNA polymerase II phosphorylated on serine 5 [Pol2(pS5)] in HeLa cells. Scale bar: 5 μm. Colocalized fractions of CHD1 with indicated proteins were quantified based on Manders' coefficient. ASH2L and SUZ12 are recruited into CHD1 condensates. SC35, a splicing factor involved in mRNA maturation, is not recruited. H3K4me3 and Pol2(pS5) are known to interact with CHD1. Data are means ± SEM. Number of cells analyzed: $n$ = 51 (ASH2L), 51 (SUZ12), 51 (SC35), 41 (H3K4me3), and 42 (Pol2[pS5]). Data were collected from three independent experiments. Two-tailed Mann-Whitney test. Source data are provided as a Source Data file.

essential for nuclear condensate formation that facilitate specific protein-protein interactions crucial for chromatin localization and activity[43]. Our findings, in conjunction with these studies, underscore the vital role of IDR-mediated condensation in chromatin remodelers, demonstrating how condensation supports genomic localization and the recruitment of functional partners.

The formation of transcriptional condensates is vital for regulating gene expression spatially and temporally[4–7,44]. While CHD1's involvement in transcription initiation and elongation is well-established[13,30], its role in transcriptional condensate formation remains unclear. Our in vitro droplet assay showed that CHD1 condensation is specifically enhanced by nucleosomes modified with H3K4me3 and RNA, which are components of active transcription sites. Additionally, interactome analysis revealed that CHD1 condensates contain RNA from target genes, likely newly transcribed from active loci. We also identified lncRNAs within CHD1 condensates that interact with the RNA polymerase II subunit A. The inhibition of RNA polymerase II and methyltransferases affected CHD1 condensation in

cells, suggesting that nascent RNAs and H3K4me3 nucleosomes facilitate CHD1 condensation. Overall, these findings suggest that CHD1 condensation is a vital component of transcriptional condensates.

The IDR-dependent CHD1 interactome includes lncRNAs and histone-modifying proteins, indicating a role in epigenetic regulation. This may account for the global gene expression changes seen in cancers with CHD1 mutations[18,19]. The significant overlap in target genes between CHD1, SUZ12, and ASH2L, along with their partial colocalization in nuclear condensates, suggests that CHD1 may interact with SUZ12-containing repressor complexes or ASH2L-containing MLL activator complexes at specific gene loci. The recruitment of CHD1 to the genome is driven mainly by H3K4me3 marks around TSS[26,27], so its interacting partners are likely influenced by transcription factors and co-factors present at specific gene loci in a cell context-dependent manner. Additionally, the colocalization of CHD1 with certain lncRNAs may also be indirectly mediated by these transcription factors and co-factors, which interact with lncRNAs[45]. Once colocalized, lncRNAs may enhance CHD1 condensation through their

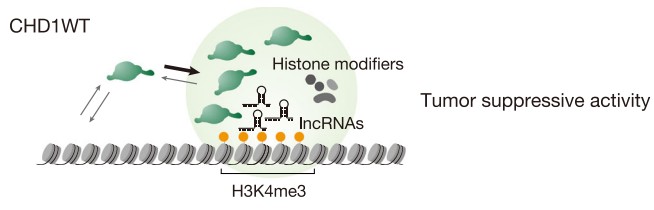

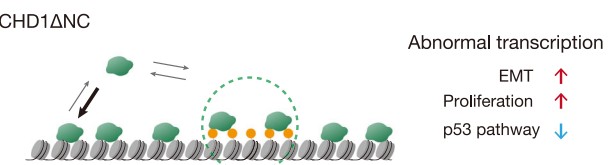

**Fig. 9 | Model illustrating the tumor suppressive role of CHD1 interactome.** CHD1 possesses N- and C-terminal IDR that enable its condensation around H3K4me3 nucleosomes and lncRNA (top). The CHD1 condensate incorporates histone-modifying proteins or their core components, such as ASH2L, EHMT1, and SUZ12, regulating gene expression in a cell context-specific manner. In cancer cells harboring the E1321fs mutation where CHD1 loses its IDR, the mutated CHD1 exhibits reduced condensate formation and fails to efficiently associate with H3K4me3 nucleosomes. Instead, it binds to the genome non-specifically, resulting in abnormal distribution and impaired condensation, which disrupts the recruitment of histone modifiers, leading to dysregulated transcription.

negative charge rather than specific RNA sequences[6], contributing to functional compartmentalization within the nucleus.

Prompted by the remarkable overlap in ASH2L and CHD1 target genes, we examined their loss-of-function in diverse cancer types and found mutations in CHD1 and histone lysine methyltransferases of the MLL complexes frequently co-occurred. Notably, MLL4/KMT2D is known to form phase-separated nuclear condensates via its prion-like domain to promote the formation of enhancer condensates[41]. Given the potential condensation properties of the MLL complex and the presence of ASH2L in CHD1 condensates, we hypothesize that CHD1 and MLL complexes may collaborate in phase separation, potentially lowering the condensation threshold. Disruption of either pathway alone may not suffice for carcinogenesis, but mutations in both could critically impair phase separation and drive cancer development. While our transcriptome and interactome analyses have identified multiple cancer-related pathways and molecules, further functional validation in future studies is required to fully understand these oncogenic processes.

Study limitations: We were unable to pinpoint the biophysical mechanisms behind CHD1 condensation in cells, such as how protein concentration influences condensate size or which specific IDR sequences drive condensation. While in vitro data show a concentration-dependent change in condensate size (Supplementary Fig. 4d, e), accurately assessing this relationship in cells remains challenging due to biological variability and technical constraints. Additionally, we relied on transient overexpression to qualitatively assess CHD1 mutant condensate formation, limiting quantitative comparisons of condensate size under physiological conditions in cells.

## Methods
### PCR analysis
Total RNA (1 µg) isolated from 22Rv1 cells using an RNeasy kit (Qiagen) was subjected to reverse transcription with a RevaTra Ace Kit (Toyobo Bio). The resulting cDNA was subjected to PCR analysis.

### Cells
22Rv1 cells, gifted from I. Tamai, were cultured in RPMI-1640 medium (Nacalai, 30264-56) with 1 mM sodium pyruvate (Wako, 190-14881) and 10% fetal bovine serum (FBS). GP5d cells (ECACC) were cultured in D-MEM supplemented with 10% FBS. HeLa cells, mCAT-HeLa cells, and Plat-E cells were gifts from K. Nakayama and were maintained under an atmosphere of 5% $CO_2$ at 37 °C in D-MEM (Nacalai, 08459-64) supplemented with 10% FBS. mCAT-HeLa cells and Plat-E were previously described in ref. 46. mCAT-HeLa cells were expressing mouse cationic amino acid transporter 1, serving as the mouse ecotropic retroviral receptor. For RNA polymerase II inhibition, $5 \times 10^5$ HeLa cells were seeded on 6-well plate. Twenty-four hours later, the cells were treated with α−amanitin (AMA, 10 µg/ml for eight hours; Wako, 010-22961), actinomycin D (ActD, 20 µg/ml for two hours, Nacalai, 00851-44) or dimethyl sulfoxide (DMSO) as a control. Cells were examined for possible mycoplasma contamination using the MycoAlert Mycoplasma Detection Kit (Lonza, LT07-118) and were negative.

### PONDR analysis
Predictions for the degree of disordered profile were performed using PONDR prediction (www.pondr.com). Predicted disorder values along the length of human CHD1 protein was determined using PONDR VL3 algorithm. High values indicate greater predicted disorder.

### Genome editing
The CRISPR-Cas9 system was used to generate genetically modified cells. Target-specific sequences were cloned into modified pX330, coding Cas9-P2A-puromycin. To generate the CHD1–Venus KI line, homology-directed repair templates were cloned into pBlueScript using NEBuilder HiFi DNA Master Mix (NEB, E2621). The homology repair template consisted of Venus cDNA sequence flanked on either side by around 1000 bp homology arms amplified from genomic DNA using PCR. The following sgRNA sequences with PAM sequence in parentheses were designed using the web tool CRISPRdirect (crispr.dbcls.jp)[47] and were used for CRISPR-Cas-targeting: sgRNA_Non_Targeting: CCGGGTCTTCGAGAAGACCT; sgRNA_CHD1_exon_7: AGATTTATGGATTGTCGGAT (TGG); sgRNA_CHD1_exon_36: AAAGTACACCGGAGCATACC (TGG); sgRNA_CHD1_intron_28: AGCATCTTCTAAACGCACCA (AGG); sgRNA_CHD1_intron_29: GAGAAAACTCGTTTAGCGTG (CGG); sgRNA_CHD1_exon_29: TTTGATGAGGTAGTCTGCAC (GGG). All oligonucleotides used in this study were synthesized by Eurofins. To generate genetically modified cell lines, $2-5 \times 10^5$ cells were transfected with 2 µg Cas9-P2A-Puro plasmid (sgRNA Non_Targeting) for nontargeting cells, 2 µg Cas9-P2A-Puro plasmid (sgRNA_CHD1_exon_7) for CHD1KO cell lines, 1 µg Cas9-P2A-Puro plasmid (sgRNA_CHD1_intron_28) and 1 µg Cas9-P2A-Puro plasmid (sgRNA_CHD1_intron_29) for CHD1-exon29-deleted cell lines or 1.25 µg Cas9-P2A-Puro plasmid (sgRNA_CHD1_exon_36 or sgRNA_CHD1_exon_29) and 1.25 µg non-linearised homology repair template for CHD1KI cell lines using XtremeGENE 9 DNA (Sigma–Aldrich, 6365787001). Forty-eight hours after transfection, cells were selected by puromycin for 48 h. The selected cells were seeded by limiting dilution. Individual clones were picked, and CHD1KO and CHD1–Venus KI clones were verified by Sanger sequence and immunoblotting.

### Tumor formation in mice
All animal procedures were approved by Kanazawa University's Institutional Animal Care and Use Committee (IACUC; approval Nos. AP-143214, AP-173859, and AP-183967). Mice were housed in AAALAC-accredited facilities and maintained on a 12 h light:12 h dark-light cycle with room temperatures of 21−23 °C with 40−60% humidity. Tumor diameter and other endpoints did not exceed the limits permitted by IACUC. All mice were maintained under specific pathogen−free conditions. 22Rv1 cells ($2.0 \times 10^4$, $1.0 \times 10^5$, or $5.0 \times 10^5$) mixed with Matrigel (1:1) were implanted into the flanks of 5-week-old male athymic nude mice (BALB/cAJcl-nu/nu, CLEA). Only male mice were used in this study because the tumor model is based on human prostate cancer, a disease specific to the male reproductive system. Tumors were

measured using a digital calliper two times a week, and tumor volumes were calculated using the formula: tumor volume = length × width × height × 0.5. Mice were euthanised when the tumor reached 150 mm³.

### RNA-seq analysis

Total RNA was extracted from 22Rv1 cells and mCAT HeLa cells using an RNeasy kit (Qiagen). Messenger RNA (1 µg) was purified from total RNA with the use of a NEBNext Poly(A) mRNA Magnetic Isolation Module (New England Biolabs). The purified RNA was used to prepare a cDNA library with the use of a NEBNext Ultra Directional RNA Library Prep Kit for Illumina (New England Biolabs). The library was then sequenced using a NovaSeq 6000 system (Illumina). The raw sequencing data quality was checked with FastQC (version 0.11.5), and trimming of adapter sequences was performed with Trimmomatic (version 0.39). The total amount of each transcript was calculated using a series of programs, including HISAT2 (version 2.2.1), Subread featureCounts (version 2.0.6) and DESeq2 (version 1.38.3). RNA-seq reads were mapped against the human (hg38) genome. GSEA was performed as described previously with the use of GSEA software (version 4.3.2) and MSigDB hallmark gene sets[34]. Gene Ontology and enrichment analysis of DEGs was performed using Metascape (http://metascape.org/)[48] and Enrichr (https://maayanlab.cloud/Enrichr/)[49].

### Cell growth assay

Genetically edited 22Rv1 cells and GP5d cells (5 × 10⁴) were subcultured in a 24-well plate and mCAT HeLa cells (1 × 10⁵) in a six-well plate. Cell number was counted by Luna II automated cell counter (Logos biosystem). Experiments were repeated in triplicate.

### Cell cycle analysis

22Rv1 cells were cultured at a density of 5 × 10⁵ in a 6-well plate for 48 h. The 5 × 10⁵ cells were harvested, and fixed with ice-cold 70% ethanol overnight at −20 °C. Cells were centrifuged and washed twice with phosphate buffer saline (PBS), and then incubated in PBS containing propidium iodide (BD Pharmingen), 20 µg/ml Ribonuclease A (RNase A, Nacalai 30141), and 0.1% Triton X-100 at 30 °C in the dark for 30 min. The stained cells were analyzed using the BD FACS Canto II and FlowJo software.

### Transwell invasion assay

The invasion of 22Rv1 cells was determined by CytoSelect Collagen Cell Invasion Assay (Cell Biolabs, CBA-110-COL). Briefly, 1.5 × 10⁵ cells in serum-free RPMI-1640 medium without phenol red were transferred into the upper part of a Boyden chamber coated with a uniform layer of dried bovine type I collagen matrix, while the bottom part medium contained 10% FBS as a chemoattractant. Cells were incubated for 48 h, and the non-invaded cells were removed from the top of the membrane. The invaded cells were stained with provided cell stain solution, and we quantified the stained area within an 8.8 mm diameter circle in the chamber (9 mm diameter).

### Cloning and protein expression

Full-length human CHD1 sequences were amplified and cloned into pBlueScript II SK(+). The N, C-terminal, or domain-deletion mutations are introduced by inverse PCR. The sequences were cloned into a modified pFastBac vector (Addgene: 55220). The construct contains an N-terminal 6 × His tag, a maltose-binding protein (MBP) tag, a 10 × Asn linker sequence, and a tobacco etch virus (TEV) protease cleavage site. All sequences were verified by sequencing. Purified plasmids were transfected into DH10Bac (Invitrogen) to generate bacmids containing full-length, N, C-terminal-deleted, or domain-deleted CHD1. Bacmids were prepared from positive clones using blue/white selection. Sf21 cell work was performed according to standard protocols in Sf-900 II SFM (Thermo Fisher) supplemented to 5% (v/v) with FBS and 1 × with penicillin, streptomycin, and amphotericin B (Nacalai 02892-54). P0

and P1 virus production were performed according to the Bac-to-Bac system (Invitrogen). Sf21 cells (5 × 10⁷/100 ml) were incubated for 24 h at 27 °C on a rotary shaker (125 rpm) and then infected with 5 ml of P1 virus for protein expression. The infected cells were grown for 48 h and harvested by centrifugation (2000 rpm [350 × g], 4 °C, 10 min). The cell pellet was flash-frozen and stored at −80 °C.

### Protein purification

Protein purifications were performed at 4 °C. The frozen Sf21 pellet was thawed and lysed by pipetting in lysis buffer (300 mM NaCl, 20 mM Na·HEPES pH 7.8, 10% (v/v) glycerol, 1 mM 1,4-Dithiothreitol (DTT), 50 mM imidazole, 1% (v/v) Nonidet P-40, ethylenediaminetetraacetic acid (EDTA)-free protease inhibitor (Nacalai 03969), 10 µg/ml DNase I (Nippon gene, 314-08071), 10 µg/ml RNase A). Lysates were placed on ice for 20 min. Samples were spun at 13,000 rpm (15,000 × g) at 4 °C for 30 min. The clarified supernatant was transferred to a new tube, and the spin was repeated. The supernatant from the second spin was filtered by a 0.45 µm filter membrane (Millipore). The filtered sample was applied onto a HisTrap HP 1 ml (Cytiva), pre-equilibrated in wash A buffer (300 mM NaCl, 20 mM Na·HEPES pH 7.8, 10% (v/v) glycerol, 1 mM DTT, 50 mM imidazole). After sample application, the column was washed with 10 column volume (CV) lysis buffer, 10 CV wash B buffer (300 mM NaCl, 20 mM Na·HEPES pH 7.8, 10% (v/v) glycerol, 1 mM DTT, 100 mM imidazole). The protein was eluted with elution buffer (300 mM NaCl, 20 mM Na·HEPES pH 7.8, 10% (v/v) glycerol, 1 mM DTT, 400 mM imidazole). Peak fractions were pooled and placed at 4 °C overnight in the presence of 100 units of TEV protease (New England Biolabs). The cleaved sample was concentrated using an Amicon Ultra-4 100,000 MWCO centrifugal concentrator (Millipore) and filtered by 0.45 µm filter membrane. The concentrated sample was subjected to gel filtration chromatography through a Superose 6 Increase 10/300 GL (Cytiva) on an ÄKTA go (GE Healthcare) in HPLC buffer (300 mM NaCl, 20 mM Na·HEPES pH 7.8, 10% (v/v) glycerol, 1 mM DTT). Peak fractions were concentrated to 1–2 µM by Amicon Ultra-0.5 mL 100,000 MWCO centrifugal concentrator (Millipore). The purified CHD1 was used within 24 h when evaluating chromatin remodeling activity and binding to nucleosomes and within 48 h in the other in vitro assays. When CHD1ΔC and ΔNC were purified, imidazole concentration was reduced (10 mM imidazole in lysis buffer, 10 mM imidazole in wash A buffer, and 40 mM imidazole in wash B buffer).

### High-speed AFM

The high-speed AFM observation of protein dynamics was performed as described previously[22]. Briefly, samples were diluted to a concentration of ~2 nM in 50 mM NaCl, 10 mM Tris HCl pH 7.5 buffer. Samples were deposited on a freshly cleaved mica surface. Five minutes after deposition, the sample was washed three times with 10 mM Tris HCl pH7.5. The sample was imaged in liquid.

### Droplet assay

Droplet assays without DNA and nucleosomes were performed by diluting recombinant protein to Droplet A buffer (final concentration: 20 mM Na·HEPES pH 7.8, 300 mM NaCl, 10% (v/v) glycerol, 5% (v/v) PEG8000). Droplet assays containing RNA, DNA, and nucleosomes (Active motif, 31295, 31586, 31600, 81002) were performed by diluting substrates and enzymes to desired concentration in Droplet B buffer (final concentration: 20 mM Na·HEPES pH 7.8, 150 mM NaCl, 1 mM MgCl₂, 10% (v/v) glycerol, 5% (v/v) PEG8000). The total RNA in Fig. 3c and Supplementary Fig. 11 was isolated from HeLa cells using an RNeasy kit (Qiagen). Diluted CHD1 was added to the bovine serum albumin (BSA)-coated glass of a 3.5 cm glass bottom dish (Matsunami, D11130H). The reaction was incubated for 60 min at room temperature to allow condensates in the solution to settle on the glass imaging surface. When recombinant CHD1 was stained with antibodies, the

enzymes were added to the Poly L-Lysine-coated glass of a 3.5 cm glass bottom dish and incubated for 60 min. The sample was fixed with 1% paraformaldehyde (PFA) for 10 min at room temperature, washed twice with PBS, and placed in 5% BSA containing PBS for 20 min. We incubated primary antibodies for 30 min. After washing three times with PBS, the sample was incubated with secondary antibodies for 30 min at room temperature, washed three times again, and stained with Rhodamine B. All the samples were imaged on a Nikon A1R confocal microscope. The condensate particle size of fluorescence-tagged enzymes was determined by comparing it to that of bright-field images.

## Immunostaining

Cells were cultured on the 3.5 cm glass bottom dish (Matsunami, D11130H), coated with Cellmatrix Type IA (Nitta Gelatin). Cells were fixed with 1% paraformaldehyde for 10 min at room temperature, washed twice with PBS, permeabilized with 0.5% Triton-X100 for 20 min at room temperature, placed in 5% BSA blocking solution for 20 min, incubated with primary antibodies (1:200 dilution) in 5% BSA blocking solution overnight at 4 °C. After washing three times with PBS containing 0.1% Triton-X100, cells were incubated with secondary antibodies (1:2000 dilution) for 30 min at room temperature and washed three times again with 0.1% Triton-X100. The nuclei were stained by 4′,6-Diamidine-2′-phenylindole dihydrochloride (DAPI). Images were captured at × 60 or 63 using a Nikon A1R, LSM900 or LSM980 confocal microscope. For quantification of colocalization of CHD1 and other proteins, the image analysis was performed with ImageJ. To focus on small compartments, a median filter of 1 pixel radius was first applied, and then thresholding using the Otsu method and analyzed by Manders' coefficient with Coloc2 plugin. The same antibodies and imaging settings (objective, light path, laser power, gain, offset, frame size, zoom, speed) were used for each sample on the same day for quantification. For the intensity quantification, at least two independent and forty cells were analyzed.

## Immunohistochemistry (IHC)

Tumor tissues were fixed in 4% paraformaldehyde at 4 °C overnight, then transferred to 70% ethanol and processed through increasing ethanol concentrations for paraffin embedding. The resulting paraffin blocks were sectioned into 4 μm slices, dried, melted at 60 °C, and rehydrated for immunohistochemistry. Antigen retrieval was performed with pH 9.0 TE Buffer (10 mM Tris, 1 mM EDTA), followed by peroxidase quenching using 3% hydrogen peroxide and blocking with 3% BSA in PBS. Primary antibodies used were anti-Ki-67 antibody (1:200 dilution), anti-Integrin α5 (1:400 dilution), and anti-SLUG/SNAI2 (1:400 dilution). For the anti-TWIST1/2 antibody (1:500 dilution), a pH 8.0 TE Buffer (10 mM Tris, 1 mM EDTA) was used for the antigen retrieval. The slides were incubated overnight, followed by incubation with secondary antibody, an anti-rabbit IgG HRP-conjugated antibody (1:200) or an anti-mouse IgG HRP-conjugated antibody (1:200) for 30 min at room temperature and visualized using 3,3′-diaminobenzidine (DAB, Vector Laboratories, Newark, USA), counterstaining with Mayer's Hematoxylin (Fujifilm, Japan) for 3 min, and imaged using an Olympus BX51 microscope (Olympus, Japan). Image J was used for analysis. For anti-Integrin α5, ten fields at 20× magnification were analyzed per sample. Images were converted to grayscale, and the blue channel was used for optimal contrast. The percentage of positive area was measured and averaged, resulting in a percentage for each sample. For anti-Ki-67, ten non-overlapping images at 40× magnification from the tumor's hotspots were captured per sample. The number of positive Ki-67-stained cells and the total number of cells were counted to calculate the Ki-67 index, expressed as a percentage. The index values from all images were averaged for each sample. For anti-TWIST1/2 and anti-SLUG/SNAI2,

ten random fields at 40× magnification were examined per sample. TWIST1/2 samples were categorized as low or high based on staining intensity, while SLUG/SNAI2 samples were classified as negative, low/medium, or high. The average staining intensity for each sample was expressed as a percentage.

## Antibodies

Rabbit polyclonal anti-CHD1 (Abcam, ab244391, immunofluorescence); rabbit monoclonal anti-CHD1 (Cell Signaling, 4351, immunoblotting); mouse monoclonal anti-CHD1 (Santa Cruz, sc-271626, immunofluorescence and biotinylation RNA-seq); rabbit polyclonal anti-GFP (MBL, 598, immunoblotting, ChIP, biotinylation M/S and PLA); rabbit monoclonal anti-Vinculin (Cell Signaling, 13901, immunoblotting); mouse monoclonal anti-HSP90 (BD Biosciences, 610419, immunoblotting); rabbit monoclonal anti-ASH2L (Cell Signaling, 5019, immunofluorescence); rabbit monoclonal anti-SUZ12 (Cell Signaling, 3737, immunofluorescence); mouse monoclonal anti-SUZ12 (Santa Cruz, sc-271325, immunofluorescence, PLA);rabbit polyclonal anti-H3K4me3 (Abcam, ab8580, immunofluorescence, immunoblotting); rabbit polyclonal anti-Histone H3 (Abcam, ab1791, immunoblotting); mouse recombinant anti-RNA pol II CTD pSer5 (Active motif, 91119, immunofluorescence); normal mouse IgG (Santa Cruz, sc-2025, PLA); alpha-tubulin (Sigma–Aldrich, T6074, immunoblotting); Rabbit monoclonal anti-Ki-67 (Abcam, ab16667, IHC); Rabbit monoclonal anti-Integrin α5 (Abcam, ab150361, IHC); Mouse monoclonal anti-SLUG/SNAI2 (Santa Cruz, sc-166476, IHC); Rabbit polyclonal anti-TWIST1/2 (GeneTex, GTX127310, IHC); Goat polyclonal anti-mouse IgG HRP-conjugated antibody (R&D systems, HAF007, IHC); Goat polyclonal anti-rabbit IgG HRP-conjugated antibody (R&D systems, HAF008, IHC).

## FRAP assay

Fluorescence recovery after photobleaching (FRAP) experiments were performed on a Nikon A1R confocal microscope. Live cell FRAP experiments were performed in FluoreBrite D-MEM medium (Thermo Fisher Scientific) at room temperature, and the fluorescence signal was bleached using the 488 nm laser at maximum intensity. Each FRAP experiment was repeated in triplicate. Intensity recovery traces obtained from the region of interest were background corrected, and all traces were normalised.

## Sliding assay

Widom 601 nucleosome positioning sequence-containing DNA fragments[50] were amplified by PCR for nucleosome reconstitution. DNA fragments used in the sliding and nucleosome binding assays were constructed by IR700 dye-labelled oligonucleotide (Eurofins genomics). All DNA concentrations were determined by nanodrop spectrophotometry. To reconstitute nucleosomes, DNA and recombinant human histone octamer (EpiCypher and AMED-BINDS) were mixed in TE plus 2 M NaCl and then assembled by stepwise salt dialysis into TE plus 1.5 M/ 1 M/ 0.75 M/ 0 M NaCl at 4 °C as described previously[51]. Sliding assays were performed as described previously[52]. Purified CHD1 enzymes were mixed with 5 nM IR dye-conjugated nucleosomes in Reaction buffer (20 mM HEPES pH 8.0, 50 mM NaCl, 3 mM MgCl$_2$, 1 mM DTT, 0.1 mg/ml BSA, 1 mM ATP) on ice. Reactions proceeded for 30 min at 30 °C. Reactions were stopped by transfer into an ice bucket and addition of 2 μl of Quench buffer (20 mM Na-HEPES pH 8.0, 50 mM NaCl, 50 mM EDTA, 240 nM unlabeled-Widom 601sequence DNA) to a 10 μl reaction volume. Reactions proceeded for 25 °C 15 min before native PAGE. Samples were loaded into a 0.5 × TBE native PAGE (SuperSep Ace 6%, Wako) and electrophoresed at 150 V for 90 min. Gels were visualized by scanning on an Odyssey Infrared Imaging System (LICOR). Experiments were repeated in duplicate. For quantitation, the signal of the nucleosome-remodeled band was measured relative to the total nucleosome signal.

## Virus preparation and transduction

pMXs-IRES-Puro retroviral vectors were transfected into Plat-E cells using PEI Max (PSI, 24765-1). pBOBI-P2A-Puro lentiviral vectors were co-transfected with packaging plasmids (LP1, LP2 and VSVG) into HEK293T cells using Lipofectamine LTX and PLUS Reagent (Thermo Fisher Scientific). Sixteen hours later, cells were washed once with PBS, and a fresh growth medium was added. The viral supernatant from 60-h post-transfection was concentrated by centrifugation ($8000 \times g$, 4 °C, 16 h), and the pellets were diluted in a culture medium with 4 µg/ml polybrene (Merck, H9268). For mCAT HeLa cell transduction, $2 \times 10^5$ mCAT HeLa cells were transduced with the virus solution for 24 h. sgNTC cells were infected with a virus carrying an empty vector, whereas CHD1KO cells were infected with viruses expressing either an empty or CHD1 variants. Cells were selected with 12 µg/mL puromycin for 48 h and maintained with 5 µg/mL puromycin.

## Chromatin Immunoprecipitation (ChIP)

ChIP was performed essentially as described previously[53]. mCAT-HeLa cells were fixed with a final concentration of 0.5% formaldehyde for 10 min at room temperature, and then the formaldehyde was quenched with a final concentration of 125 mM glycine. After washing by PBS, the cells were, suspended in ChIP buffer (5 mM HEPES-KOH (pH 8.0), 200 mM KCl, 1 mM CaCl₂, 1.5 mM MgCl₂, 5% sucrose, 0.5% Nonidet P-40, EDTA-free protease inhibitor), incubated for 10 min on ice, subjected to ultrasonic treatment with the use of an Astrason XL-2020 ultrasonic processor, and digested with micrococcal nuclease for 40 min at 30 °C. After adding EDTA to a final concentration of 0.1 mM, each digested sample was centrifuged at $15,000 \times g$ for 10 min at 4 °C. The resulting supernatant was incubated with rotation for 6 h at 4 °C with antibodies conjugated to magnetic beads. After washing twice with ChIP buffer and Tris-EDTA buffer, bound proteins were eluted from the beads, and cross-links were reversed by incubation overnight at 65 °C with 1% sodium dodecyl sulfate (SDS) in Tris-EDTA buffer. The reverse-crosslinked DNA was purified using Nucleo Spin Gel and PCR Clean-Up (Takara Bio) and subjected to ChIP-seq analysis.

## ChIP-seq analysis

The immunoprecipitated DNA was used for ChIP-seq library preparation using the ThruPLEX DNA-Seq Kit (Takara Bio, R400674). The DNA was purified by Agencourt AMPure XP (Beckman Coulter). A final size selection was performed using 2% agarose E-Gel (Invitrogen), and the library was recovered by MinElute PCR Purification Kit (Qiagen). Libraries were sequenced using the NovaSeq 6000 (Illumina) on paired-end 150 bp mode. Available ChIP-seq data obtained with antibodies to H3K4me3 and H3K27ac in HeLa cells (DRR014675 and DRR014677) were reanalysed. The reads were uniquely mapped to the human (hg38) genome with the use of Bowtie software (version 2.4.5), and duplicated reads were removed with samtools (version 1.14). Markedly enriched genome regions were identified using the MACS peak caller (version 2.2.7.1, with the option '-gsize hs -nomodel -extsize 160 -p 1e-5'). Peak annotation was performed with ChIPseeker (version 1.34.1). After using the S3norm method[54] to normalize sequencing depths and signal-to-noise ratios in silico, heatmaps were generated with deeptools computeMatrix and plotHeatmap (version 3.5.1). We assigned a specific gene as a CHD1-target gene when the CHD1 ChIP peaks were identified within a 1000 bp region proximal to the TSS, utilizing bedtools (version 2.31.0).

## Assay for toransp[sic]ase accessible chromatin using sequencing (ATAC-seq) analysis

ATAC-seq was performed in biological duplicate using the ATAC-Seq Kit (Active Motif, 53150). Briefly, $1 \times 10^5$ mCAT-HeLa cells were used to isolate nuclei. After lysing the cells in ATAC Lysis buffer, nuclei were incubated with tagmentation Master Mix at 37 °C for 30 min and the DNA was purified with the DNA purification column. PCR amplification of tagmented DNA was performed to make libraries with the appropriate indexed primers. After SPRI bead clean-up, the DNA libraries were sequenced using the NovaSeq 6000 (Illumina) on paired-end 150 bp mode. The raw sequencing data were trimmed by Trimmomatic (version 0.39), and paired end reads were aligned to the human (hg38) genome with the use of Bowtie software (version 2.4.5), and duplicated reads were removed with samtools (version 1.14). Markedly enriched genome regions were identified using the MACS peak caller (version 2.2.7.1, with the option '-gsize hs --nomodel --shift -50 --extsize 100 -p 1e-5'). Peak annotation was performed with ChIPseeker (version 1.34.1). We utilized the S3norm method[54] to normalize sequencing depths and signal-to-noise ratios in silico and visualized the alignments using the Integrative Genomics Viewer (IGV, version 2.1.6).

## Nucleosome binding assays

Purified CHD1 was mixed with 5 nM IR dye-conjugated nucleosome in EMSA buffer (10 mM HEPES pH 8.0, 75 mM NaCl, 5 mM MgCl₂, 10% (v/v) glycerol, 1 mM DTT, 0.1 mg/ml BSA) on ice. Reactions proceeded for 30 min at 30 °C prior to native PAGE and LICOR scanning as described above. Experiments were repeated in triplicate. For quantitation, the signal of the CHD1-bound band was measured relative to the total substrate signal.

## Fluorescently-tagged nucleic acid preparation

Fluorescent RNA for droplet assays was prepared as described before[6]. pTRI-Xef1, containing the Xenopus elongation factor 1 alpha gene, was transcribed using the MEGAscript T7 kits (Invitrogen) according to the manufacturer's instructions. Reactions included a Cy5-labeled UTP (Enzo LifeSciences ENZ-42506) at a ratio of 1:10 labelled UTP: unlabeled UTP. The MEGAclear Transcription Clean-Up Kit (Invitrogen) was used to purify the RNA following the manufacturer's introductions. Labelled RNA was run on 1% agarose gels in TAE buffer to verify a single band of the correct size.

## High-resolution AFM

The local structural characteristics of the CHD1 condensate formations were investigated using a Bruker BioScope Resolve AFM system. The system was operated in PeakForce Tapping mode with ScanAsyst in fluid, and it was equipped with a NanoScope V controller. The AFM images were acquired using AC240-NG cantilevers (purchased from OPUS) with a nominal spring constant of 2 N/m and a nominal tip radius of <8 nm. We experimentally determined the individual spring constant of each cantilever using the standard thermal tune method. AFM image rendering and data processing were performed by using the NanoScope Analysis software (version 1.9, Bruker, Billerica, MA, USA). The PeakForce tapping amplitude, and frequency, and setpoint were 100–200 nm, 2 kHz, and 0.5–2 nN, respectively. During the AFM observation, a 100 µL condensate of the CHD1 protein solution, with a concentration of 90 nM, was applied to a freshly cleaved mica surface. Following a 60 min incubation period at room temperature, the sample was rinsed multiple times with 20 mM of HEPES buffer and subjected to AFM investigations. In addition to the topographical characterization, we performed nanomechanical measurements using PeakForce tapping AFM (PF-QNM) to provide insights into the structural organization of the condensates. To characterize the nanomechanical properties of condensates, Young's modulus mapping was performed using a contact mechanics model provided by Sneddon et al. [55].

## Antibody-based in situ biotinylation assay

Antibody-based in situ biotinylating was performed as described previously[36]. HeLa cells were fixed with 4% paraformaldehyde and permeabilized with 0.5% Triton X-100-containing PBS. The cells were blocked with BSA and horse serum, and incubated with primary antibodies in the same protocol of immunostaining described above. The

cells were washed with 0.5% Triton X-100-containing PBS, and the cells were then incubated with horseradish peroxidase (HRP)-conjugated anti-mouse (CST, #7076) or HRP-conjugated anti-rabbit IgG (CST, #7074). After washing, HRP-based in situ biotinylation was performed by incubating the cells in biotinylation buffer (200 μM biotin-tryamide and 0.0015% $H_2O_2$ in PBS) for 1 min, which catalyzed the biotinylation of molecules within a ~20 nm radius. The biotinylation reaction was stopped by washing cells with PBS immediately, and the cells were harvested by scraping in 1% SDS Lysis buffer (150 mM NaCl, 1% Triton X-100, 0.5% Sodium deoxycholate (Doc), 1% SDS, 50 mM Tris-HCl (pH 8.0)). The samples were sheared by sonication using Bioruptor Sonicator (Diagenode, Denville, NJ) for 25 min. For mass-spectrometry analysis of biotinylated proteins, the sheared lysate was boiled for 20 min at 95 °C for reverse cross-linking, and debris was removed by centrifugation. For RNA purification for RNA-seq, the boiling step was skipped. After centrifugation, the lysate was diluted into 0.5% SDS concentration, followed by avidin pull-down using SoftLink™ Soft Release Avidin Resin (Promega, V2011). The avidin resin was washed with 0.5% SDS RIPA buffer (150 mM NaCl, 1% Triton X-100, 0.5% Doc, 0.5% SDS, 50 mM Tris-HCl (pH 8.0)), 0.5 M NaCl RIPA buffer (500 mM NaCl, 1% Triton X-100, 0.5% Doc, 0.1% SDS, 50 mM Tris-HCl (pH 8.0)), 1.2 M NaCl RIPA buffer (1.2 M NaCl, 1% Triton X-100, 0.5% Doc, 0.1% SDS, 50 mM Tris-HCl (pH 8.0)), and 0.5% SDS RIPA buffer again.

### Biotinylation RNA-seq analysis

After biotinylation and avidin pull-down, the bound protein-RNA complex was eluted and reverse-crosslinked by incubation for 1 h at 55 °C in TE buffer containing 1% SDS, 0.2 M NaCl and proteinase K. The pulled-down RNAs were isolated by Trizol LS (Invitrogen, 10296028) and ethanol precipitation. cDNA library was prepared by NEBNext Ultra Directional RNA Library Prep Kit for Illumina (New England Biolabs). Immunoprecipitated and input materials were analyzed by RNA-seq. Before library preparation, equal amount of *Drosophila* S2 RNA was added to each RNA sample as spike-in control. The library was then sequenced using a NovaSeq 6000 system (Illumina) on paired-end 150 bp mode. The raw sequencing data quality was checked with FastQC (version 0.11.5), and trimming of adapter sequences was performed with TrimGalore (version 0.6.7). The reads were uniquely mapped to the human (GRCh38/hg38) genome and *Drosophila* (dm6) genome with the use of Bowtie2 software (version 2.4.5). Aligned reads were sorted using samtools (version 1.14). To split reads into human and *Drosophila* reads and calculate scaling factors, the Spiker split_bam.py script (version 1.0.3) was run[56]. The human reads were converted to gene counts using featureCounts (with the option '-t gene') based on the GENECODE v38 annotation and normalized to the minimum of *Drosophila* reads. We excluded genes with an average read count of 10 or below, and then add the counts to one. Gene set enrichment analysis of lncRNA was performed using LncSEA2.0 (https://bio.liclab.net/LncSEA)[37].

### Proteomic analysis

For peptide sample preparation for mass-spectrometry, the avidin resin was rinsed with distilled water for three times, and resuspend with 2 M urea in 50 mM $NH_4HCO_3$. After incubating the resin with 100 mM DTT for 30 min at 37 °C, and with 250 mM iodoacetamide for 15 min at room temperature, add 0.5 μg Trypsin in 50 mM $NH_4HCO_3$ and incubate overnight at 37 °C for peptide digestion. The trypsin-digested peptides were purified using C18 and styrenedivinylbenzene filters. Proteins were identified by LC-MS/MS analysis using Orbitrap Elite, Hybrid Ion Trap-Orbitrap Mass Spectrometer (Thermo Fisher Scientific Inc.) coupled with an UltiMate 3000 HPLC system (Thermo Fisher Scientific). Peptide samples were loaded on a trap column (100 μm × 20 mm, C18, 5 μm, 100 Å, Thermo Fisher Scientific) and separated on a Nano HPLC capillary column (75 μm × 18 cm, C18, 3 μm, Nikkyo Technos, Tokyo, Japan) at a flow rate of 300 nL/min. Solvent A

was 0.1% formic acid in 2% acetonitrile, while solvent B was 0.1% formic acid in 95% acetonitrile. The peptide samples were eluted using a gradient beginning with 2% B for 0–5 min, then 2% to 33% B for 5–120 min, followed by 90% B for 10 min, and finally equilibration with 2% B for 20 min. The data were acquired using a survey scan performed in a mass range from 380 to 1500 m/z. The top 15 peaks were selected for fragmentation. Peptide identification, protein identification, and label-free quantification were performed using MaxQuant software (v1.6.7.0) against human protein sequences in the UniProt Knowledgebase (UniProtKB/SwissProt). Search parameters were as follows: enzyme, trypsin; variable modifications, oxidation (M), Acetyl (Protein N-term); fixed modifications, carbamidomethyl (C). A false discovery rate of less than 5% was adopted as the acceptance criteria for identifications. A single biological replicate was analyzed for this condition.

### Proximity Ligation Assay (PLA)

The PLA was performed using Duolink PLA Fluorescence protocol (Sigma–Aldrich). Briefly, cells were seeded, fixed, and permeabilized as described in the immunostaining method, and then blocked according to the manufacturer's instructions. Primary antibodies targeting the proteins of interest were applied and incubated overnight at 4 °C. After washing with 1× Wash Buffer A, Duolink PLA probes were then added and incubated for 60 min, followed by ligation and amplification steps to generate fluorescence signals. After washing with Wash Buffer B, the nuclei were stained by DAPI. Images were captured at 63× magnification using an LSM900 confocal microscope under the same settings. PLA signals were quantified by observing all z-axis slices within the nucleus. Random fields of view were selected, and 30 cells were counted. This procedure was repeated in two independent experiments to ensure reproducibility and statistical significance.

### Statistical analysis and reproducibility

Prism 6 (GraphPad Software) was used to calculate statistics. Results were considered significant at a *P*-value of <0.05. Relevant statistical methods for individual experiments are detailed within figure legends. No statistical method was used to predetermine sample size. No data were excluded from the analyses. The experiments were not randomized. The investigators blinded to allocation during experiments and outcome assessment.

### Reporting summary

Further information on research design is available in the Nature Portfolio Reporting Summary linked to this article.

## Data availability

Source data are provided with this paper. All data are available in the main text or the supplementary information. This paper does not report original code. RNA-seq, ChIP-seq and ATAC-seq data have been deposited in the NCBI sequence read archive under the accession number PRJNA1013253, PRJNA1013747, PRJNA1014223 PRJNA1014249 and PRJNA1148813. Publicly available ChIP-seq datasets analyzed in this study were obtained from the DDBJ Sequence Read Archive under the following accession numbers: DRR014675 and DRR014677. Mass spectrometry data have been deposited in the JPOST Repository under accession code JPST004000. All data are available in the main text or the supplementary information. Any additional information required to reanalyze the data reported in this paper is available from the lead contact upon request. Source data are provided with this paper.

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

## Acknowledgements

We would like to thank K. Tsukamoto, Y. Tanabe, G. Batbayar, A. Ishimura, H. Konno, N. Yilmaz, K. Tomoda, T. Daikoku, H. Kawasaki and S. Yamanaka for technical assistance and discussion; T. Hara, A. Kida, R. Kohno, K. Isozaki, C. Tambo and M. Sora for technical assistance; This study was supported by Ministry of Education, Culture, Sports, Science and Technology (MEXT), Japan, a Grant-in-Aid for JSPS Scientific Research (19J23485) to Y.T., (21K18250) to K.M., and (JP21H02847 and 22H05592) to M.N., PRIME grant from the Japan Agency for Medical Research and Development (AMED) (JP22gm6310008) to M.N., Yasuda Medical Foundation to M.N., Research Support Project for Life Science and Drug Discovery (Basis for Supporting Innovative Drug Discovery and Life Science Research (BINDS)) from AMED, and the World Premier International Research Center Initiative (WPI), MEXT, Japan.

## Author contributions

Y.T., A.K., D.I, D.C.V., T.F., A.M., and K.S. conceived and designed the research; Y.T. performed experiments and analyzed data; H.Suzuki performed antibody-based in situ biotinylation assay; A.Y., W.W., and T.I. performed high-resolution AFM; N.M.R.C. performed immunohistochemistry; H.Sato performed immunofluorescence imaging; Y.T., H.Suzuki, H.T., N.K., T.F, K.M., S.H., K.S.; M.N. and Y.O. reviewed data; Y.T., D.C.V., and K.S. wrote the paper; All authors discussed the data and commented on the manuscript.

## Competing interests

The authors declare no competing interests.

## Additional information

[1]Division of Tumor Dynamics and Regulation, Cancer Research Institute, Kanazawa University, Kanazawa, Japan. [2]Department of Histology and Cell Biology, Graduate School of Medical Sciences, Kanazawa University, Kanazawa, Japan. [3]Department of Medicine and Bioregulatory Science, Graduate School of Medical Sciences, Kyushu University, Fukuoka, Japan. [4]Nano Life Science Institute (WPI-NanoLSI), Kanazawa University, Kanazawa, Japan. [5]Department of Molecular Biology, Graduate School of Medical Science, Yokohama City University, Yokohama, Japan. [6]Inflammation and Epithelial Plasticity Unit, Cancer Research Institute, Kanazawa University, Kanazawa, Japan. [7]Innovative Cancer Model Research Unit, Institute of Frontier Sciences Initiative, Kanazawa University, Kanazawa, Japan. [8]Division of Biological Science, Graduate School of Science, Nagoya University, Nagoya, Japan. [9]Social Brain Development Research Unit, Next Generation Medical Development Research Core, Institute for Frontier Science Initiative, Kanazawa University, Kanazawa, Japan. [10]Present address: Gladstone Institute of Cardiovascular Disease, Gladstone Institutes, San Francisco, CA, USA. [11]Present address: Department of Life and Environmental System Science, Graduate School of Nanobioscience, Yokohama City University, Yokohama, Japan. [12]Present address: Department of Pharmacology, Graduate School of Medicine, The University of Osaka, Osaka, Japan. [13]These authors contributed equally: Atsuki Kawamura, Ayhan Yurtsever, Hidefumi Suzuki. ✉e-mail: ytsukamo@stu.kanazawa-u.ac.jp; ogawa.yoshihiro.828@m.kyushu-u.ac.jp; k_sakai@staff.kanazawa-u.ac.jp

