## [Peer Review file · Nature Communications]

Condensation-dependent interactome of a chromatin remodeler underlies tumor suppressor activities

Corresponding Author: Dr Katsuya Sakai

Version 0:

Reviewer comments:

Reviewer #1

(Remarks to the Author)

In this manuscript, authors uncovered a novel biologic function of chromatin remodeler, in particular CHD1, on forming transcriptional condensate, which is crucial for gene regulation and tumor development. This study demonstrated that CHD1 forms liquid-state condensates via its C-terminal IDR, which is frequently frameshift mutated in cancers. Besides, authors found that CHD1 condensates are facilitated by H3K4me3-modified nucleosomes and RNA and localized at active promoters. Also, authors showed the frameshift mutations E1321fs promotes cancer cell proliferation and tumor growth. Importantly, using unbiased in situ biotinylation, authors identified that CHD1 interacts with lncRNAs and MLL histone modifying proteins inside the condensates, providing new insights into the components and function of CHD1 condensates during tumor development.

Overall, this study is well-designed, novel and significant. These findings advanced our understanding the biology and epigenetic basis of CHD1 and other chromatin remodelers. The in-depth mechanistic study also underscores the importance of identification of CHD1 genetic status in cancer patients. I have a few suggestions to make their conclusions more robust.

Major comments:

1. Fig 2 showed that exogenous CHD1-Venus forms nuclear condensation in HeLa cells and depletion of C- or N-terminal IDRs reduces the formation of CHD1 condensation. Does the endogenous wildtype CHD1 protein or E1321fs mutant displayed differential nuclear condensation in cancer cells?
2. In Fig 4, authors demonstrated that H3K4me3 nucleosome and RNA stimulates CHD1 condensation. However, all results are based in vitro assays. The conclusion could be strengthened by additional evidence from cellular assays. For examples, inhibit RNA synthesis or histone methyltransferases that modify H3K4me3 in HeLa cells, and then determine CHD1 condensation.
3. In Fig 5, ChIP-seq showed that CHD1 Δ NC has reduced binding affinity to TP53 and CDKN1B. Determination of mRNA level of these genes in CHD1 WT vs Δ NC cells is needed to verify IDR's role in activating transcription of tumor suppressor genes.
4. In Fig 7, authors used CHD1 Δ ex29 to rescue E1321fs-induced protein truncation. However, compared to parent 22Rv1 cells containing one copy of wildtype CHD1 and one copy of E1321fs, the engineered CHD1 Δ ex29/ Δ ex29 22Rv1 cells loss the entire exon 29 on both CHD1 gene copies. CHD1 Δ ex29/ Δ ex29 showed inhibitory effects on cell proliferation, tumor growth, and changed downstream genes and pathways. However, additional evidence is needed to consolidate the conclusion that the rescued E1321fs mutant, but not loss of functional domain encoded by exon 29, led to these effects.
5. Fig 7c showed that mScarlet-tagged CHD1 Δ ex29 forms nuclear condensation in 22Rv1 cells. mScarlet-tagged CHD1fs and mScarlet-tagged CHD1WT should be used as control groups to draw the conclusion.
6. Phase separation is considered often mediated by IDRs, and previous studies indicated that IDR mutations disrupt phase separation in key cellular processes and CHD1 family proteins were predicted to phase separate (PMID: 33357399). In this manuscript, authors found that CHD1 and CHD7 can form condensates and IDR is required for this process, consistent with previous prediction. When demonstrating the novelty of finding CHD1 condensation via IDR (Line 353-357), prior studies should also be considered and discussed.

Minor comments:

7. Quantification of CHD1 colocalization with other proteins is needed in Fig 9D.
8. Line 300-301, the conclusion that CHD1 and MLL “functionally compensatory in their respective roles as tumor suppressors” is overclaimed, since it is just based on the co-occurrence of CHD1 and MLL mutations in cancer
9. Please provide a description of Predictor of Naturally Disordered Regions (PONDR) analyses in Method.
10. For Extended Data Fig. 2C, please indicate the color key of the image.

Reviewer #2

(Remarks to the Author)

Tsukamoto et al demonstrate that the recurrent frameshift-induced truncation of the C-terminal IDR of CHD1 leads to a reduction in CHD1 phase separation, protein and RNA interactions, and tumor suppressor activities. The authors have performed a comprehensive study of the recurrent frameshift mutation (E1321) in CHD1 implicated in cancer (Fig 1). The authors show that the disordered C-terminal region of CHD1, which is truncated by the frameshift is required for the formation of biomolecular condensates in vitro and in cells (Fig 2). They also compare the condensate formation properties of other CHD family proteins to CHD1 finding that only CHD1 and CHD7 form condensates in their assay conditions (Fig 3.). The authors demonstrate the contributions of H3K4me3 nucleosomes and RNA in promoting CHD1 condensate formation (Fig 4). In addition, the authors present evidence on how the disordered N- and C-terminal regions of CHD1 are required for highly selective binding of CHD1 to H3K4me3 modified nucleosomes (Fig. 6). Tsukamoto et al., provide a comprehensive panel of genomics (Fig 5), transcriptomics (Fig 7), and antibody-based in situ biotinylation assays for RNA-seq (Fig 8) and proteomics analysis (Fig 9). Finally, the authors present a model based on their results where loss of either disordered terminal region in CHD1 leads to reduced condensate formation, and loss of its interacting partners (H3K4me3 nucleosomes, lncRNAs, and histone modifiers), which results in dysregulated transcription (Fig. 10).

The major strengths of this paper are the focus on disease-relevant human genetics, the examination of this mutant CHD1 in diverse experimental context (from cell-free to animal studies), and the use of cutting-edge techniques to address important questions bridging chromatin, condensate, and cancer biology. This work constitutes a significant contribution to these fields. The authors go beyond even recently published papers on BAF remodeler phase separation to connect disease mutations with phenotypes in cells and in mice. Overall, the authors present compelling evidence in support of their model, which will be of interest to the readership at Nature Communications. Once the authors address the following specific comments, the manuscript will be suitable for publication.

- 1) Is there evidence at the protein level that the truncated protein is expressed in cancer cells? The data presented in Extended Data Figure 1 demonstrates that the stop-gain is in the final exon, which should escape non-sense mediated decay, but it would substantially improve the manuscript if the authors could show that this truncated protein is present in cancer cell lines. A western blot for CHD1 in 22Rv1 cells or other cell lines could address this.
- 2) For the exon skipping experiments presented in figure 7, it will be helpful to present the amino acid differences between WT, fs, and dex29. This will help the reader understand this experiment and its potential caveats. It will also be helpful to provide additional rationale for why this strategy was taken instead of an editing strategy to restore the sequence to WT. The results are compelling, but modifying the text and figures to more clearly setup the experiment will improve the manuscript.
- 3) In its current form Figure 3 does not seem to fit within the manuscript. It is difficult for the reader to understand why these experiments are being presented. What could help is to include a diagram and data for CHD1 in panel A. Is there something that distinguishes CHD1 from the other family members that do not form condensates? Is there something shared between CHD1 and CHD7? Are the C-terminal disordered regions of different family members very different from one another? Is CHD1 unique from the others in its association with human disease?
- 4) Also related to Figure 3, the pattern of condensates in CHD7 overexpression does not resemble the pattern of CHD1 observed by IF. Instead of presenting the overexpression in 3C, it will be more relevant to present co-IF for CHD1 and CHD7.
- 5) Line 170. The authors should provide justification for testing delta N and C IDRs instead of just delta-C, which the previous figures used.
- 6) Do the genes upregulated on Figure 7 panel F for the WT/fs mutant contain associated peaks for CHD1 delta-NC from experiments on Figure 5? The experiments use different versions of the CHD1 protein, but it might be helpful to put the data in context with other experiments performed in the manuscript.
- 7) For Figure 9D, quantification and negative control need to be included.
- 8) Figure 9E and 9F also require a negative control. In other words, what is the standard to know you are not obtaining false

positives in these statistical tests? These data could also be put in supplementary and replaced by quantification for panel D because they do not directly relate to the title of the figure.

9) The authors need to cite papers from the BAF (SWI/SNF) chromatin remodeler literature implicating those chromatin remodelers in condensate formation. PMID: 34018649 and 37788668

10) The authors should be cautious about stating “condensate-dependent” when what they mean is “IDR-dependent”. In particular related to the interactions observed by proximity biotinylation in figures 8 and 9.

Reviewer #3

(Remarks to the Author)

In this work, Tsukamoto and collaborators investigated the contribution of CHD1 condensates to tumor progression in a model of prostate cancer. They dissect the potential molecular mechanisms that guide CHD1 to assemble into condensates by LLPS, and they characterize the contribution of the C-term IDR in modulating the interactions with other chromatin factors and paraps with lincRNAs during this process. The molecular and biological insights are relevant in the field as the performed analyses tackle a key question related to the contribution of IDR in modulating the function of chromatin factors both in physiological and pathological conditions. The results obtained do not fully support the raised conclusions, and the authors need to perform a series of controls to strengthen the robustness of the results and the significance of their data. In addition, the work contained multiple approaches and experimental settings which are poorly controlled or not fully in line with the flow of the manuscript, which reduced the interest and the relevance of the findings; we strongly suggest limiting the presented data to the results that are properly integrated and controlled (see below).

In sum, although the proposed role of CHD1 condensates in contribution to the growth of (prostate) cancer cells, further experiments are required to improve the soundness of the raised conclusions. Indeed, there are important technical limitations and a lack of appropriate controls that reduce the impact and the robustness of this study that need to be addressed before considering the manuscript ready for publication.

Major criticisms:

1. The expression of the exogenous CHD1-Venus showed the dependency on the presence of both N- and C-terminus CHD1 in the assembly of large widespread droplets. However, if comparing the observed pattern upon KI the Venus in the locus of CHD1, the overexpressed isoform gives rise to larger droplets, which may depend on the expression level of the recombinant protein (Fig. 2b, c). To address this discrepancy, it would be relevant to compare the level of the exogenous OE constructs with respect to the endogenous CHD1. Are these stable OE cells? How homogenous is the expression level within the cell population? Would it be relevant to define the assembly of droplets in dependency on the IDRs in another cancer-related (prostate) model? CHD1 expression varies quite broadly in different tumors and within the same prostate cancer samples, which is further highlighted by the relative expression in prostate cancer cell lines, with 22Rv1 expressing a relatively low level if compared with PC-3 or VCaP.

1. Regarding the KI experiment, author should clarify whether they analyze the CHD1 clustering in a single clone only, and if so it would be necessary to test different clones to ascertain the clonal-independency for the described phenotype. In addition, the FRAP assay, as performed, is not sufficient to describe the assembly of CHD1 in condensates through LLPS. We strongly encourage to deep into this, by defining the biophysical properties of the endogenous CHD1 by discriminating between LLPs and other mechanisms governing the assembly of condensates. For example, the authors should consider the surface tensile forces of the droplets to determine the behavior of CHD1 within cells (DOI:<https://doi.org/10.1016/j.molcel.2020.02.005>).

2. The comparison between the CHD family members does not add much to the paper and, as presented, is inconclusive. There are too many differences in terms of protein size, the possible extent of the relative IDR, expression levels, and functionality. The presented data lack the minimum number of controls to support the raised conclusions (replicates, comparison of protein level, analyses of the endogenous proteins for all the members, etc.). I would strongly suggest to simply remove this part of the work.

3. The authors should better explain the rationale for the selected molar ratio between the recombinant CHD1 and the nucleosome in the droplet assay (Fig 4a). The same concern holds true for the spiked RNA molecules, as the relative abundance of RNA can vary the number, size, and timing of droplet assembly. They should repeat the experiment carefully, considering these variables and determining their contribution to the CHD1 droplet formation. In addition, it is not clear which RNA has been labeled, the rationale of this choice, and whether the length of the RNA, its potential secondary structures, or sequence biases may determine the contribution of RNA species in guiding CHD1 assembly into droplets. Are RNA molecules involved in the nucleation or in the growth of the droplets? Time course analyses could help distinguish these aspects of LLPS.

4. ChIP-seq analyses suggest that the C-term IDR contributes to CHD1 chromatin recruitment at active promoters. Although of interest, the presented data do not support entirely the raised conclusion. As highlighted in ED Fig. 7a, the rescue of CHD1 KO cells with either the WT or the C-term IDR deletion showed a remarkable difference in the protein levels, which per se can explain the retrieved results. The authors should repeat these assays using a cellular model with comparable (and properly quantified) expression levels of both WT and IDR-deleted IDR, which should be similar to the endogenous level, thus representing a rescue experiment setting. To better normalize the data, this kind of analysis requires the usage of calibrated ChIP-seq by spiking-in mouse (or Drosophyla) cells expressing a-chromatin protein before performing the IP. In addition, as CHD1 is a chromatin remodeler, it is essential to determine whether any possible alteration of its chromatin function through the IDR involves a perturbation of the histone turnover at the NFR of promoters. This can be assessed

either by MNase-seq or by ATAC-seq. The drawn conclusion on the contribution of the IDR on the cell cycle control and p53 pathway suffers from the same limitation (not properly rescuing CHD1 level); please address this issue.

5. The in vivo tumorigenic assays to determine the contribution of CDH1E1321fs are highly relevant and well-controlled. However, using only three mice/conditions for a short time window is not fully appropriate to dissect the contribution of the mutant to the complex process of tumorigenesis. If possible, we suggest repeating the analyses by injecting a lower number of cells and extending the analyses at later points to ensure that different kinetics of tumor growth can explain the phenotype. In addition, it is recommended to perform a phenotypic characterization of the formed tumors to determine the proliferative index (either using Ki67 or H3Ser10ph), the level of apoptosis, and eventually, the EMT phenotype/invasiveness.

6. In relation to the mapping of lincRNAs interacting with CHD1, although the analyses are of high relevance, they do not support the conclusion that this class of RNA species supports the CHD1 assembly in condensates. They indicate that some lincRNAs (whose promoter is not a target of CHD1) are in proximity to CHD1 but do not tell much about their function nor their contribution to the formation of condensates. Either the authors provide some data to support these conclusions, or this part of the manuscript should be reconsidered. For example, they could test the specificity of some of the identified lincRNAs in guiding the assembly of the droplets using an in vitro droplet assay. The same approach could be tested in vitro by perturbing the relative abundance of the most (Cancer-related) relevant lincRNAs.

7. Regarding the differential contribution of chromatin factors interacting with CHD1 through the C-term IDR, the proteomic analyses should be better supported by orthogonal experiments. For example, the IF analyses require co-localization analyses (Fig. 9c). We would also suggest to repeat the same experiments using the rescued cells, after controlling that the same expression level is achieved (that should be similar to the endogenous CHD1).

Version 1:

Reviewer comments:

Reviewer #1

(Remarks to the Author)

In the revised version, the authors satisfactorily addressed all my initial comments with convincing original data. The current version of the manuscript is now very strong and acceptable for publication.

Reviewer #2

(Remarks to the Author)

The authors have addressed all of the concerns raised in my initial review. The authors have added a significant amount of new data in support of their conclusions. The revised manuscript is suitable for publication.

Reviewer #3

(Remarks to the Author)

The authors provided supportive data to address some of the raised concerns, which improve the robustness of the manuscript.

However, some important issues have not been addressed. Specifically, referring to the specific raised points:

1. The author did not address the discrepancy between the size of the droplets obtained from the overexpressed isoform compared to the KI system. They indeed showed that they have heterogeneous expression of the both N- and C-terminus CHD1, yet they are confident that the expression level did not impact on the protein distribution. This conclusion is actually not supported by the presented data in which they do not show quantitative measurements of the number and size of the droplets in respect to the expression level. The comparison of the low- vs high-expressing cells do show a clear difference in the distribution and number of the formed droplets. Moreover, the Delta-N construct delineate a very peculiar distribution of the protein, which seemingly resides in heterochromatin regions: why? Additionally, they did not show the relative level of OE with respect to the endogenous level, as requested.

2. The author concluded that although different cell lines expressed different levels of CHD1, this does not impact on its distribution. This conclusion is not fully supported by the data as they did not quantify the number and size of the droplets and relate it to the overall change in protein abundance, that should be also confirmed by IF, at least. What happens if the same cancer cell line they perturb the level of the endogenous CHD1?

3. The authors stated they analyzed different KI clones, which behaved similarly without major differences. Nice to know, but they did not provide any data supporting this statement. They instead show data on HeLa cells in which they OE the construct to then perform FRAP assay: why? This experiment does not address the original question. On the same point, they argue that determining the biophysical properties of CHD1 is outside the scope of this work. I disagree completely as the drawn conclusions of this work rely on the potential contribution of CHD1 to form condensates, which modulated the chromatin remodeling and can influence tumor progression.

4. The authors still consider necessary to describe the behavior of the other CHD proteins. They perform additional data that instead of clarifying the potential differences among this diverse family of remodellers, it adds only confusion. For example, how do they explain the inconsistency of the pattern distribution of the endogenous Chd7 vs the OE form, which showed a clear heterochromatin pattern? This figure is irrelevant for the manuscript, there are no quantifications (just descriptive) and inconsistent.

5. The authors provided relevant controls to better titer the abundance of the recombinant CHD1 and the co-mixed nucleosomes, strengthening the significance of these data. Yet they should comment why another methylated nucleosome

(H3K9me3) increased the droplet formation. Is it possible that the in vitro assay conditions do not permit to assess the specificity of CHD1 chromodomains towards the K4me3? This point should be discussed.

6. Regarding the role of nascent RNAs in modulating CHD1 condensates, the authors argue that CHD1 behaves similarly to other chromatin factors such as MED1 and BRD4 whose condensates formation is modulated by electrostatic-driven RNA-protein interaction (PMID: 33333019). However, the authors failed to address our criticisms as they do not provide any data supporting this hypothesis. For example, if this prediction is correct, then other polyanionic molecules should give rise to similar results. In the same mentioned work, the hypothesis of charge balancing predicted that RNA synthesis should impact on condensates formation, and they showed experimentally that RNAP II inhibition increased the size of the condensates, which is the opposite of what measured in this manuscript (Extended Data Fig. 6a-e). How do the authors explain this contradictory results (respect to the charge balancing mechanisms)? Referring to the same experiments, they concluded that RNA synthesis inhibition reduced CHD1 condensates; however, the shown representative IF images indicated a clear decrease of CHD1 abundance, which seemingly was not detected by WB. This incongruity needs to be addressed by using independent biological replicates and by performing adequate statistical analyses. The control of the drug efficacy in inhibiting transcription elongation is incorrect as they are measuring the steady state of total RNA (which is dominated by the large abundance of rRNAs and tRNAs) and not the RNA synthesis. In sum these set of experiments did not address the raised criticisms.

7. The protein level of the rescued experiments has not been quantified by WB or other means. This point should be carefully addressed, as it can strongly impact on the data interpretation. We understand the difficulties of having a single Ab recognizing all the protein mutants, but the author can reach quantitative information's by performing shotgun proteomics using as reference peptides shared among all the CHD1 isoforms. Similarly, the authors did not perform calibrated ChIP-seq nor orthogonal approach to properly analyze the retrieved data. Without these controls, the comparative analyses do not support the raised conclusions.

8. The additional experiments related to the in vivo tumorigenic assay fully addressed the raised criticisms.

9. The authors tuned down their conclusions on the potential role of lincRNAs in mediating CHD1 condensates, in line with the raised criticism.

10. The proteomics results through which the authors identified other chromatin factors interacting with CHD1 have been properly validated, although the limitation regarding the rescue experiments related to the proper quantification of CHD1 variants somehow undermine the relevance of these findings. In addition, the colocalization analyses by MCC lack the appropriate statistical tests to support the conclusions. Indeed, by considering as n the number of analyzed cells, they measured the cell-to-cell variability among the two conditions, but they did not test the null hypothesis for which they need to consider the number of replicates. As in this case they have two biological replicates, they have to increase the numerosity of independent biological replicates before applying the appropriate nonparametric statistical test.

Version 2:

Reviewer comments:

Reviewer #3

(Remarks to the Author)

Although the author has not fully addressed all raised concerns, I acknowledge their effort in responding to the criticisms. That said, I remain convinced that the following points require further discussion or revision:

Protein Abundance and Droplet Size: The relationship between protein abundance and droplet size has not been experimentally addressed. Presenting non-quantitative data is insufficient to resolve this issue. The author should include a discussion section acknowledging this limitation. Importantly, no one compelled the author to rely on transient overexpression, which they use to justify inconsistencies in their data, when a more rigorous approach—such as deriving stable single clones—would have allowed for proper analysis.

Biophysical Properties of Condensates: As the author has not adequately determined the biophysical properties of the condensates, all references to liquid-liquid phase separation (LLPS) should be removed as agreed with authors. Additionally, the manuscript must explicitly state as a limitation that the biophysical mechanisms driving condensate formation could not be determined.

CHD Protein Droplet Assays: The decision to not quantify droplet assays for the other CHD proteins suggests that the data lack consistency. Arguing that this is the first report of their ability to form condensates is not a valid justification for presenting poorly controlled experiments. Scientific conclusions must be based on robust, well-controlled data. Unless proper quantification is performed, these data should be removed.

By addressing these concerns transparently and acknowledging the study's limitations, the manuscript will be significantly strengthened.

Version 3:

Reviewer comments:

Reviewer #3

(Remarks to the Author)

The authors properly addressed all the raised points by including new data and/or by commenting on some intrinsic limitations of the study, as suggested. I am fully satisfied with the revised manuscript, and I would like to thank the authors for acknowledging the relevance of the raised points and for addressing them accordantly.

Point-By-Point Response to the Reviewers' Comments

We thank the reviewers for the careful evaluation of our manuscript, providing insightful feedback aimed at strengthening our manuscript. In the subsequent paragraphs, we address the comments raised by each reviewer. Our responses are in blue.

Reviewer #1:

In this manuscript, authors uncovered a novel biologic function of chromatin remodeler, in particular CHD1, on forming transcriptional condensate, which is crucial for gene regulation and tumor development. This study demonstrated that CHD1 forms liquid-state condensates via its C-terminal IDR, which is frequently frameshift mutated in cancers. Besides, authors found that CHD1 condensates are facilitated by H3K4me3-modified nucleosomes and RNA and localized at active promoters. Also, authors showed the frameshift mutations E1321fs promotes cancer cell proliferation and tumor growth. Importantly, using unbiased in situ biotinylation, authors identified that CHD1 interacts with lncRNAs and MLL histone modifying proteins inside the condensates, providing new insights into the components and function of CHD1 condensates during tumor development.

Overall, this study is well-designed, novel and significant. These findings advanced our understanding the biology and epigenetic basis of CHD1 and other chromatin remodelers. The in-depth mechanistic study also underscores the importance of identification of CHD1 genetic status in cancer patients. I have a few suggestions to make their conclusions more robust.

Response: The authors are grateful to the reviewer for careful evaluation and positive assessment of our work.

Major comments:

1. Fig 2 showed that exogenous CHD1-Venus forms nuclear condensation in HeLa cells and depletion of C- or N-terminal IDRs reduces the formation of CHD1 condensation. Does the endogenous wildtype CHD1 protein or E1321fs mutant displayed differential nuclear condensation in cancer cells?

Response: Given the absence of antibodies exclusively recognizing the CHD1^{E1321fs} protein variant, investigating the distinct condensation patterns between endogenous CHD1^{E1321fs} and CHD1^{WT} via immunostaining is presently unfeasible. Consequently, we employed the CRISPR-mediated knock-in method to introduce the CHD1^{WTs}-Venus or CHD1^{E1321fs}-Venus gene into HeLa cells, thereby express CHD1^{E1321fs}-Venus protein at endogenous levels. The CHD1 E1321fs protein demonstrated a more homogeneous distribution within the nucleus compared to the CHD1 WT-Venus protein, which exhibited more condensed patterns (See Extended Data Fig. 4, below). These results have been added in the revised manuscript (lines 142-148, Extended Data Fig. 4).

Extended Data Fig. 4 | Nuclear condensation of CHD1^{WT}-Venus and CHD1^{E1321fs}-Venus. a, Live cell imaging of HeLa cells with knock-in of either CHD1^{E1321fs}-Venus or CHD1^{WT}-Venus. Nuclei were stained with Silicon

Rhodamine (SiR)-DNA for live-cell imaging. The Otsu thresholding method was utilized to derive binary values from fluorescence intensity, enabling quantification of CHD1 occupancy in the nucleus. Scale bars, 5 μm . **b**, Forty cells were analyzed from two clones. Each cell's percentage was plotted as a dot and analyzed using an unpaired Student's t-test.

2. In Fig 4, authors demonstrated that H3K4me3 nucleosome and RNA stimulates CHD1 condensation. However, all results are based in vitro assays. The conclusion could be strengthened by additional evidence from cellular assays. For examples, inhibit RNA synthesis or histone methyltransferases that modify H3K4me3 in HeLa cells, and then determine CHD1 condensation.

Response: We appreciate the reviewer's suggestion and assessed the effect of RNA polymerase II inhibitors and a methyltransferase inhibitor on CHD1 condensation in cells (see Extended Data Fig. 6, below). Amanitin, an RNA polymerase II inhibitor, reduced CHD1 condensation in the nucleus without altering CHD1 protein levels (Extended Data Fig. 6 a-e). Actinomycin D, which inhibits both RNA polymerase I and II, significantly decreased cellular RNA levels and reduced CHD1 condensation, accompanied by a marked reduction in both H3K4me3 and CHD1 protein levels, although the underlying mechanism remains unclear (Extended Data Fig. 6 a-e). Sinefungin, a methyltransferase inhibitor, decreases the trimethylation level at H3K4 and diminished CHD1 condensation (Extended Data Fig. 6f-h). These results suggest that RNA and H3K4me3-modified nucleosomes promote and stabilize CHD1 condensates in cells. These results have been added in the revised manuscript (lines 183-193, Extended Data Fig. 6).

Extended Data Fig. 6 | Inhibition of RNA polymerases or histone methyltransferases leads to a reduction in CHD1 condensation within cells. a, Representative immunofluorescence images showing the CHD1 condensates under the effects of RNA polymerase II inhibition. Scale bars, 5 μm . The fluorescence intensity profiles (bottom) were taken along the white dashed line. High condensation areas of CHD1 were defined using a cutoff intensity of 1.0×10^5 (indicated by the dashed line). **b**, The percentage of high condensation areas of CHD1 in each nucleus

was plotted as individual dots and analyzed using an unpaired Student's t-test. **c**, Western blot analysis for CHD1 and H3K4me3. **d,e** The effects of RNA polymerase II inhibition on cell viability (**d**), and total cellular RNA amount (**e**). **f**, Representative immunofluorescence images showing the CHD1 condensates under the effects of sinefungin (Sine), a methyl-transferase inhibitor. Scale bars, 5 μ m. **g**, Western blot analysis for CHD1 and H3K4me3 in the presence of sinefungin. **h**, Percentage of high condensation areas of CHD1 (a cutoff intensity of 6.0×10^4) in each nucleus was plotted as individual dots and analyzed using an unpaired Student's t-test (right).

3. In Fig 5, ChIP-seq showed that CHD1 Δ NC has reduced binding affinity to TP53 and CDKN1B. Determination of mRNA level of these genes in CHD1 WT vs Δ NC cells is needed to verify IDR's role in activating transcription of tumor suppressor genes.

Response: In response to the reviewer's request, we measured the expression of CDKN1B and TP53 mRNA and found that, consistent with the reduced occupancy at these loci, the reconstitution of CHD1 Δ NC resulted in lower expression of these mRNAs, relative to the reconstitution of WT CHD1 (see Fig. 4e, below). These results have been added in the revised manuscript (line 206, Fig. 4e).

Fig. 4e | RT-qPCR analysis of *CDKN1B* and *TP53* expression levels normalized to *GAPDH*.

4. In Fig 7, authors used CHD1 Δ ex29 to rescue E1321fs-induced protein truncation. However, compared to parent 22Rv1 cells containing one copy of wildtype CHD1 and one copy of E1321fs, the engineered CHD1 Δ ex29/ Δ ex29 22Rv1 cells loss the entire exon 29 on both CHD1 gene copies. CHD1 Δ ex29/ Δ ex29 showed inhibitory effects on cell proliferation, tumor growth, and changed downstream genes and pathways. However, additional evidence is needed to consolidate the conclusion that the rescued E1321fs mutant, but not loss of functional domain encoded by exon 29, led to these effects.

Response: Firstly, following is a brief explanation of why we used the knock-in of CHD1 Δ ex29 instead of restoring the WT sequence. Initially, we attempted to restore the WT sequence using CRISPR-mediated editing, as illustrated below. This approach mirrors the strategy used to knock-in CHD1^{WT}-Venus in both alleles. However, in 22Rv1 prostate cancer cells, the efficiency of knock-in was notably lower than in HeLa cells, resulting in WT sequence knock-in for only one allele across all screened cells. While CRISPR-mediated cleavage occurred efficiently in both alleles, the allele where the WT sequence failed to integrate resulted in indel mutations, causing a premature termination codon. Therefore, we chose CRISPR-mediated exon skipping. This strategy avoids the need to introduce foreign sequences via knock-in, thereby efficiently inducing exon 29 skipping in both alleles.

Figure for the reviewers only. Schematic illustrating the strategy for CHD1^{WT}-Venus KI HeLa cell generation, which unsuccessfully resulted in the generation of an indel mutation in one of the alleles.

To clarify this reviewer's concern whether the CHD1^{WT} and CHD1^{Δex29} exhibit similar outcomes, we compared the effects of lentivirus-mediated expression of CHD1^{WT} and CHD1^{Δex29} on proliferation and gene expression in 22Rv1 CHD1-knockout cells (see Supplementary Fig. 11, below). Both variants similarly suppressed growth (Supplementary Fig. 11b) and upregulated CDKN1A, CDKN1B, and BAX (Supplementary Fig. 11c), indicating that the partial loss of the functional domain does not impact these outcomes. These results have been added in the revised manuscript (lines 275–282, Supplementary Fig. 11).

Supplementary Fig. 11 | Effect of CHD1^{Δex29} and CHD1^{WT} on growth and gene expression related to cell cycle inhibition. **a**, Western blot analysis showing CHD1 expression levels in 22Rv1^{Δex29/Δex29} cells and CHD1KO 22Rv1 cells, with the latter transduced with either a puromycin resistance gene (CHD1KO bulk) or CHD1^{WT}-p2a-puro. **b**, Growth curves of 22Rv1 cells. **c**, RT-qPCR analysis of expression levels for *CDKN1A*, *CDKN1B* and *BAX*, normalized to *GAPDH*.

5. Fig 7c showed that mScarlet-tagged CHD1^{Δex29} forms nuclear condensation in 22Rv1 cells. mScarlet-tagged CHD1^{fs} and mScarlet-tagged CHD1^{WT} should be used as control groups to draw the conclusion.

Response: We apologize for not providing adequate experimental controls. To address this, we have now included images of exogenously expressed mScarlet-tagged CHD1^{WT}, CHD1^{E1321fs}, and CHD1^{Δex29} (see Fig. 7c, below). The results support our conclusion that CHD1^{WT} and CHD1^{Δex29} strongly condense in the nucleus, while CHD1^{E1321fs} weakly condenses and is more evenly distributed. These results have been added in the revised manuscript (lines 252–253, Fig. 6c).

Fig. 7c | Confocal images of HeLa cells expressing exogenous CHD1^{WT}-mScarlet, CHD1^{fs}-mScarlet, or CHD1^{Δex29}-mScarlet, demonstrating nuclear condensation of CHD1^{WT} and CHD1^{Δex29}, but not CHD1^{E1321fs}. Scale bars, 5 μm.

6. Phase separation is considered often mediated by IDRs, and previous studies indicated that IDR mutations disrupt phase separation in key cellular processes and CHD1 family proteins were predicted to phase separate (PMID: 33357399). In this manuscript, authors found that CHD1 and CHD7 can form condensates and IDR is required for this process, consistent with previous prediction. When demonstrating the novelty of finding CHD1 condensation via IDR (Line 353–357), prior studies should also be considered and discussed.

Response: We thank the reviewer for raising this important point and have cited the paper (PMID: 33357399) in the result section (line 151) and the first paragraph of the Discussion section (lines 347–349) in the revised manuscript.

Minor comments:

7. Quantification of CHD1 colocalization with other proteins is needed in Fig 9D.

Response: To address the reviewer's comment, we repeated the experiment and quantified CHD1 colocalization with other proteins by Manders coefficients for each protein (see Fig. 9d, below). We used SC35, a splicing factor involved in mRNA maturation, as a negative control, as it is not recruited in CHD1-condensates. Conversely, H3K4me3 and RNA polymerase II phosphorylated at serine 5 were used as positive controls as these proteins are known interaction partners of CHD1. We have also provided more details in the Methods section (lines 596–603). These results have been added in the revised manuscript (lines 322–324, Fig. 8d).

Fig. 8. | **d**, Confocal images showing the colocalization of CHD1 and ASH2L, SUZ12, SC35, H3K4me3, or RNA

polymerase II phosphorylated on serine 5 [Pol2(pS5)] in HeLa cells. Scale bars, 5 μm . Colocalized fractions of CHD1 with indicated proteins were quantified based on Manders' coefficient. ASH2L and SUZ12 are recruited into CHD1 condensates. SC35, a splicing factor involved in mRNA maturation, is not recruited. H3K4me3 and Pol2(pS5) are known to interact with CHD1. Data are presented as means \pm SEM of forty cells in at least two independent experiments.

8. Line 300–301, the conclusion that CHD1 and MLL “functionally compensatory in their respective roles as tumor suppressors” is overclaimed, since it is just based on the co-occurrence of CHD1 and MLL mutations in cancer

Response: We have revised the statement from “This provides compelling clinical evidence that CHD1 and MLL might be functionally compensatory in their respective roles as tumor suppressors” to “These data point to a potential functional compensation between CHD1 and MLL in their roles as tumor suppressors.” (lines 342–343).

9. Please provide a description of Predictor of Naturally Disordered Regions (PONDR) analyses in Method.

Response: As requested, we have provided the information in the Methods section (lines 435–438).

10. For Extended Data Fig. 2C, please indicate the color key of the image.

Response: Accordingly, we have added the color key to the image, which indicates the height.

C

Extended Data Fig. 1 | c, Domain-deleted CHD1 variants visualized by HS-AFM. c, HS-AFM results showing domain assignment in CHD1 variants through single-molecule live imaging. Scale bar, 10 nm. Schematic diagrams depicting the domains of domain-deleted variants.

Reviewer #2 (Remarks to the Author):

Tsukamoto et al demonstrate that the recurrent frameshift-induced truncation of the C-terminal IDR of CHD1 leads to a reduction in CHD1 phase separation, protein and RNA interactions, and tumor suppressor activities. The authors have performed a comprehensive study of the recurrent frameshift mutation (E1321) in CHD1 implicated in cancer (Fig 1). The authors show that the disordered C-terminal region of CHD1, which is truncated by the frameshift is required for the formation of biomolecular condensates in vitro and in cells (Fig 2). They also compare the condensate formation properties of other CHD family proteins to CHD1 finding that only CHD1 and CHD7 form condensates in their assay conditions (Fig 3.). The authors demonstrate the contributions of H3K4me3 nucleosomes and RNA in promoting CHD1

condensate formation (Fig 4). In addition, the authors present evidence on how the disordered N- and C-terminal regions of CHD1 are required for highly selective binding of CHD1 to H3K4me3 modified nucleosomes (Fig. 6). Tsukamoto et al., provide a comprehensive panel of genomics (Fig 5), transcriptomics (Fig 7), and antibody-based in situ biotinylation assays for RNA-seq (Fig 8) and proteomics analysis (Fig 9). Finally, the authors present a model based on their results where loss of either disordered terminal region in CHD1 leads to reduced condensate formation, and loss of its interacting partners (H3K4me3 nucleosomes, lncRNAs, and histone modifiers), which results in dysregulated transcription (Fig. 10).

The major strengths of this paper are the focus on disease-relevant human genetics, the examination of this mutant CHD1 in diverse experimental context (from cell-free to animal studies), and the use of cutting-edge techniques to address important questions bridging chromatin, condensate, and cancer biology. This work constitutes a significant contribution to these fields. The authors go beyond even recently published papers on BAF remodeler phase separation to connect disease mutations with phenotypes in cells and in mice. Overall, the authors present compelling evidence in support of their model, which will be of interest to the readership at Nature Communications. Once the authors address the following specific comments, the manuscript will be suitable for publication.

Response: The authors are grateful to the reviewer for careful evaluation and positive assessment of our work.

1) Is there evidence at the protein level that the truncated protein is expressed in cancer cells? The data presented in Extended Data Figure 1 demonstrates that the stop-gain is in the final exon, which should escape non-sense mediated decay, but it would substantially improve the manuscript if the authors could show that this truncated protein is present in cancer cell lines. A western blot for CHD1 in 22Rv1 cells or other cell lines could address this.

Response: As the reviewer highlighted, the detection of the C-terminal truncated CHD1 protein is a critical observation. We recognized its importance during the initial preparation of our manuscript. All antibodies used in Western blotting for CHD1 specifically target the C-terminus, thereby exclusively detecting CHD1^{WT} (see below Fig. a: Antibody from CST). Antibodies recognizing regions other than the C-terminus, which could potentially detect both CHD1^{WT} and CHD1^{E1321fs}, performed poorly in Western blotting (see Fig. b: Antibody from biobyte, St John's Lab, or Abnova). Consequently, we developed custom antiserum by immunizing rabbits with an N-terminal peptide of CHD1 and applied it in Western blotting (see Fig. c: house-made antibody). This method revealed bands corresponding to CHD1^{WT} protein and a smaller band likely representing CHD1^{E1321fs}. However, due to band ambiguity, definitive conclusions could not be drawn.

Figure for the reviewers only. **a**, Schematic showing CHD1 antibody recognition sites. Almost all CHD1 antibodies recognize CHD1 C terminus, which CHD1^{E1322fs} does not have. **b**, **c**, Western blot analysis of CHD1 expression in 22Rv1 cells. House-made CHD1 antibody was designed to recognize the N-terminus of CHD1, but its specificity for CHD1 was low (**c**). CHD1 antibodies designed to recognize other than the C-terminus also exhibit weak binding to CHD1 (**b**).

Due to challenges in obtaining suitable antibodies for Western blotting, we aimed to demonstrate that the CHD1^{E1322fs} mutation evades mRNA decay. To investigate this, we employed a knock-in strategy to insert the Venus fluorescent protein between the CHD1^{E1322fs} mutation and the resulting STOP codon (see Supplementary Fig. 2a, below). Through Western blotting (see Supplementary Fig. 2b, below) and fluorescence observations (see Extended Data Fig. 4, below), we confirmed comparable expression levels of the CHD1^{E1322fs}-Venus to that of CHD1^{WT}-Venus. Although indirect, these results suggest that the CHD1^{E1322fs} protein is produced by evading mRNA decay. These results have been added in the revised manuscript (lines 93-96, Supplementary Fig. 2; lines 142-148, Extended Data Fig. 4).

Supplementary Fig. 2 | Expression of the $CHD1^{E1321fs}$ -Venus knock-in protein suggests that it might be resistant to mRNA decay mechanisms. a, The schematic outlines the strategy used for generating HeLa cells with knock-in of either $CHD1^{E1321fs}$ -Venus or $CHD1^{WT}$ -Venus. **b**, Western blotting using anti-CHD1 antibody (which specifically detects $CHD1^{WT}$) and anti-Venus antibody.

Extended Data Fig. 4 | Nuclear condensation of $CHD1^{WT}$ -Venus and $CHD1^{E1321fs}$ -Venus. a, Live cell imaging of HeLa cells with knock-in of either $CHD1^{E1321fs}$ -Venus or $CHD1^{WT}$ -Venus. Nuclei were stained with Silicon Rhodamine (SiR)-DNA for live-cell imaging. The Otsu thresholding method was utilized to derive binary values from fluorescence intensity, enabling quantification of CHD1 occupancy in the nucleus. Scale bars, 5 μ m. **b**, Forty cells were analyzed from two clones. Each cell's percentage was plotted as a dot and analyzed using an unpaired Student's t-test.

2) For the exon skipping experiments presented in figure 7, it will be helpful to present the amino acid differences between WT, fs, and dex29. This will help the reader understand this experiment and its potential caveats. It will also be helpful to provide additional rationale for why this strategy was taken instead of an editing strategy to restore the sequence to WT. The results are compelling, but modifying the text and figures to more clearly setup the experiment will improve the manuscript.

Response: We acknowledge the reviewers' comments regarding the loss of 39 amino acids in exon skipping. To address this, we have now included a detailed comparison of the amino acid differences between samples as in Fig. 6a (below) and explained that "A deletion of exon 29 results in the loss of 39 amino acids, 28 of which are part of the DNA-binding domain." In the figure legend.

Fig. 6 | a, Schematic representation of CRISPR/Cas9-mediated exon 29 skipping to restore the C-terminus of CHD1. A deletion of exon 29 results in the loss of 39 amino acids, 28 of which are part of the DNA-binding domain.

Regarding our choice of methods, we initially attempted to restore the sequence to WT using CRISPR-mediated editing, as illustrated below. This approach mirrors the strategy used to knock in CHD1^{WT}-Venus in both alleles. However, in 22Rv1 prostate cancer cells, the efficiency of knock-in was notably lower than in HeLa cells, resulting in WT sequence knock-in for only one allele across all screened cells. While CRISPR-mediated cleavage occurred efficiently in both alleles, the allele where WT sequence failed to integrate resulted in indel mutations, causing premature termination codon. Therefore, we chose CRISPR-mediated exon skipping. This strategy avoids the need to introduce foreign sequences via knock-in, thereby efficiently inducing exon 29 skipping in both alleles.

Figure for the reviewers only. Schematic illustrating the strategy for CHD1^{WT}-Venus KI HeLa cell generation, which was unsuccessful due to the inadvertent generation of an indel mutation in one of the alleles.

To provide further evidence that CHD1^{Δex29} is functionally comparable to CHD1^{WT}, we compared their effects following lentiviral transduction into 22Rv1 CHD1-knockout cells (see Supplementary Fig. 11, below). Both variants similarly suppressed growth (Supplementary Fig. 11b) and upregulated CDKN1A, CDKN1B, and BAX (Supplementary Fig. 11c), indicating that the partial loss of the functional domain does not impact these outcomes. These results have been added in the revised manuscript (lines 275–282, Supplementary Fig. 11).

Supplementary Fig. 11 | Effect of CHD1^{Δex29} and CHD1^{WT} on growth and gene expression related to cell cycle inhibition. **a**, Western blot analysis showing CHD1 expression levels in 22Rv1^{Δex29/Δex29} cells and CHD1KO 22Rv1 cells, with the latter transduced with either a puromycin resistance gene (CHD1KO bulk) or CHD1^{WT}-p2a-puro. **b**, Growth curves of 22Rv1 cells. **c**, RT-qPCR analysis of expression levels for *CDKN1A*, *CDKN1B* and *BAX*, normalized to *GAPDH*.

3) In its current form Figure 3 does not seem to fit within the manuscript. It is difficult for the reader to understand why these experiments are being presented. What could help is to include a diagram and data for CHD1 in panel A. Is there something that distinguishes CHD1 from the other family members that do not form condensates? Is there something shared between CHD1 and CHD7? Are the C-terminal disordered regions of different family members very different from one another? Is CHD1 unique from the others in its association with human disease?

Response: Accordingly, we added a diagram and data for CHD1 in panel A (see Supplementary Fig. 4a, right). Additionally, we examined the condensation properties of all CHD family proteins, including CHD3, CHD5, and CHD9, to address which members have the capability to form condensates. Our findings show that only CHD1, CHD7, and CHD9 exhibit condensation capability (see Supplementary Fig. 4a, below). These experiments provide insight into which CHD family members undergo liquid-liquid phase separation, however since the mechanism remains unresolved, we have moved these results to Supplementary Fig. 4 and excluded them from the main figures.

Supplementary Fig. 4 | Condensation properties of the CHD family. **a**, Schematic representations of CHD family protein domains (left, upper) and analysis of PONDR scores indicating IDRs (left, lower). Confocal images of HeLa cells expressing Venus-CHD proteins demonstrate the condensation property of CHD1, CHD7 and CHD9 (right column). Scale bars, 5 μm.

Like the reviewer, we were very much intrigued if CHD1, CHD7, and CHD9 share certain cryptic features within their IDRs, a hidden "sequence grammar" as it were. As such, we employed localCIDER to analyze the net charge per residue across these IDRs but did not find any notable common patterns. To uncover significant sequence features, more comprehensive methods such as the NARDINI+ algorithm (PMID: 37788668) and functional validation through mutant studies may be necessary. Given the constraints of the revision period and the fact that this is not the primary focus of our study, we have chosen to leave this question for future investigation.

4) Also related to Figure 3, the pattern of condensates in CHD7 overexpression does not resemble the pattern of CHD7 observed by IF. Instead of presenting the overexpression in 3C, it will be more relevant to present co-IF for CHD1 and CHD7.

Response: We observed that exogenously expressed CHD1, CHD7, and CHD9 formed condensates, as shown in Supplementary Fig. 4c (below). At the reviewer's request, we also stained for endogenous CHD1, CHD7, and CHD9 in cells to examine their condensation properties and colocalization, which is presented in Supplementary Fig. 4b (below). Our findings indicate that endogenous CHD1, CHD7, and CHD9 also have condensation capabilities. Notably, CHD1 showed a higher degree of colocalization with CHD9 than with CHD7. These results have been incorporated into the revised manuscript (lines 156–161, Supplementary Fig. 4b, c).

Supplementary Fig. 4 | b, Confocal images showing the colocalization of fluorescence-tagged CHD1 with CHD7 or CHD9 in HeLa cells. Scale bars, 5 μm. **c**, Immunofluorescence images showing the colocalization of endogenous CHD1 with CHD7 or CHD9 in HeLa cells. Scale bars, 5 μm. The colocalization efficiency was assessed using Manders' coefficients (right).

5) Line 170. The authors should provide justification for testing delta N and C IDRs instead of just delta-C, which the previous figures used.

Response: We prioritized CHD1^{ΔNC} over CHD1^{ΔC} to achieve a clearer functional contrast with CHD1^{WT} for the following reasons. Our preliminary ChIP-seq experiments had earlier revealed that both CHD1^{ΔC} and CHD1^{ΔNC} exhibited reduce accumulation at promoter regions and increased accumulation at intronic and intergenic regions compared to CHD1^{WT} (see Figure below). Notably, CHD1^{ΔNC} showed more pronounced deviations in these accumulation patterns compared to CHD1^{ΔC}. Furthermore, in droplet assay, CHD1^{ΔNC} demonstrated significantly lower phase separation capability than CHD1^{ΔC}.

Figure for the reviewer only. The distributions of ChIP-seq peak according to the corresponding genomic regions. ChIP-seq signals in CHD1^{WT}, CHD1^{ΔC} and CHD1^{ΔNC}-reconstituted CHD1KO mCAT-HeLa cells.

6) Do the genes upregulated on Figure 7 panel F for the WT/fs mutant contain associated peaks for CHD1 delta-NC from experiments on Figure 5? The experiments use different versions of the CHD1 protein, but it might be helpful to put the data in context with other experiments performed in the manuscript.

Response: We analyzed 590 differentially expressed genes from RNA-seq data (22Rv1 cells, CHD1^{WT/fs} vs. CHD1^{Δex29}) and 634 genes with specific peaks from ChIP-seq data (HeLa cells,

CHD1^{WT} vs. CHD1^{ΔNC}. We identified only 10 overlapping genes (see Fig. a, below). We attribute this limited overlap to significant differences in cellular contexts. However, the top 10 clusters in pathway and process enrichment analyses of *CHD1^{ΔNC}*-specific target genes (ChIP-seq) include genes related to cell proliferation (see Fig. b, c, below), which is also observed in the RNA-seq analysis (Fig. 6f). This indicates that both *CHD1^{WT/fs}* and *CHD1^{ΔNC}* may promote proliferation in their respective cell types. In conclusion, while the reviewer's suggestion to describe the results in a unified context is appreciated, we have chosen not to incorporate it due to the difficulty in drawing clear conclusions from the current data.

Figure for the reviewer only. **a**, Venn diagram showing down-regulated DEGs in Δ ex29 cells shared among *CHD1^{ΔNC}*-specific target genes in HeLa cells, indicating few genes are identical. **b,c**, Pathway and process enrichment analyses of *CHD1^{ΔNC}*-specific target genes using Metascape revealing that the genes include cell cycle related genes (b), particularly those driving the cell cycle (c). Light green bars indicate the leaf nodes of GO term “regulation of cell cycle phase transition”.

7) For Figure 9D, quantification and negative control need to be included.

Response: Accordingly, we quantified CHD1 colocalization with other proteins by Manders' coefficients for each protein (see Fig. 8d, below). We used SC35, a splicing factor involved in mRNA maturation, as a negative control, as it is not detected by in situ biotinylation M/S. Conversely, H3K4me3 and RNA polymerase II phosphorylated at serine 5 are known as CHD1 related protein, so we used them as positive controls. Additionally, we have provided more detailed information in the Methods (lines 596–603). These results have been added in the revised manuscript (lines 322–324, Fig. 8d).

Fig. 8 | d, Confocal images showing the colocalization of CHD1 and ASH2L, SUZ12, SC35, H3K4me3, or RNA polymerase II phosphorylated on serine 5 (pS5) in HeLa cells. Scale bars, 5 μ m. Colocalized fractions of CHD1 with indicated proteins were quantified based on Manders' coefficient. ASH2L and SUZ12 are recruited into CHD1

condensates. SC35, a splicing factor involved in mRNA maturation, is not recruited. H3K4me3 and Pol2(pS5) are known to interact with CHD1. Data are means \pm SEM of forty cells in at least two independent experiments.

8) Figure 9E and 9F also require a negative control. In other words, what is the standard to know you are not obtaining false positives in these statistical tests? These data could also be put in supplementary and replaced by quantification for panel D because they do not directly relate to the title of the figure.

Response: In the original Figures 9E and F, the negative control was not explicitly described. We have now added this clarification in the figure legend (see Extended Data Fig. 10, below) and in the Results section (lines 336–342), indicating that p53 serves as the negative control. To summarize, mutations in KMTs are notably more common in cancers with CHD1 mutations compared to the general cancer population. However, p53 mutations do not show a significant difference between cancers with and without CHD1 mutations. Conversely, mutations in CHD1 are significantly more frequent in cancers with KMT mutations compared to all cancers, but there is no significant difference in the incidence of p53 mutations between cancers with and without CHD1 mutations. In response to the reviewers' suggestion, these data have been relocated to Extended Data Fig. 10 with an appropriate title.

Extended Data Fig. 10 | Co-occurrence of mutations between CHD1 and KMTs in human cancers. a, Percentages of MLL/KMT2 family mutations and TP53 mutations in all cases across the NCI Genomic Data Commons, CHD1 mutations cases, and CHD1E1321fs cases. TP53 is used as a standard. Two-tailed binominal test. **b,** Percentages of CHD1 mutations in all cases, MLL/KMT2 family and TP53 mutations cases. Two-tailed binominal test. **b,** Percentages of CHD1 mutations in all cases, MLL/KMT2 family and TP53 mutations cases. Two-tailed binominal test.

9) The authors need to cite papers from the BAF (SWI/SNF) chromatin remodeler literature implicating those chromatin remodelers in condensate formation. PMID: 34018649 and 37788668.

Response: We thank the reviewer for bringing this to our attention. The paper with PMID 37788668 was not published at the time of our original submission. We have now added both references in the second paragraph of the Discussion section (lines 356–360). Additionally, we have expanded our discussion to address the ongoing exploration of the role of IDRs in chromatin remodelers and their involvement in condensate formation (lines 360–363).

10) The authors should be cautious about stating “condensate-dependent” when what they mean is “IDR-dependent”. In particular related to the interactions observed by proximity biotinylation in figures 8 and 9.

Response: We thank the reviewer for this expert's opinion. We have revised the term “condensation-dependent” to “IDR-dependent” to reflect this distinction accurately.

Reviewer #3 (Remarks to the Author):

In this work, Tsukamoto and collaborators investigated the contribution of CHD1 condensates to tumor progression in a model of prostate cancer. They dissect the potential molecular mechanisms that guide CHD1 to assemble into condensates by LLPS, and they characterize the contribution of the C-term IDR in modulating the interactions with other chromatin factors and paraps with lincRNAs during this process. The molecular and biological insights are relevant in the field as the performed analyses tackle a key question related to the contribution of IDR in modulating the function of chromatin factors both in physiological and pathological conditions. The results obtained do not fully support the raised conclusions, and the authors need to perform a series of controls to strengthen the robustness of the results and the significance of their data.

Response: We appreciate the reviewer's recognition of the study's importance. In response to the concerns regarding experimental controls, we have conducted additional experiments with appropriate controls to address these issues. Below, we detail the specific changes and improvements made.

Major criticisms:

1. The expression of the exogenous CHD1-Venus showed the dependency on the presence of both N- and C-terminus CHD1 in the assembly of large widespread droplets. However, if comparing the observed pattern upon KI the Venus in the locus of CHD1, the overexpressed isoform gives rise to larger droplets, which may depend on the expression level of the recombinant protein (Fig. 2b, c). To address this discrepancy, it would be relevant to compare the level of the exogenous OE constructs with respect to the endogenous CHD1. Are these stable OE cells? How homogenous is the expression level within the cell population?

Response: In the exogenous expression experiments shown in Fig. 2 and 3, we used transient transfection to introduce the Venus-fused CHD1 protein. After plasmid transfection, nearly all cells expressed the Venus-tagged proteins and displayed fluorescence, although expression levels varied (see figure below). We specifically selected images of cell nuclei with saturated CHD1 expression for both CHD1^{WT}-Venus and CHD1^{ΔNC}-Venus in Fig. 2 and 3. Even at saturated expression levels, CHD1^{ΔNC}-Venus exhibited less condensation and a more uniform distribution throughout the nucleus (see Figure b, below). In contrast, CHD1^{WT}-Venus displayed distinct condensation patterns at both high (see Figure a-A, below) and low expression levels (see Figure a-B, below). These results clearly demonstrate that the condensation properties of CHD1 are mediated by its IDR, independent of expression levels.

Figure for the reviewers only. Confocal images of HeLa cells expressing -CHD1^{WT}-Venus (a) or CHD1^{ΔNC}-Venus (b) demonstrate their condensation property. Confocal images were taken at laser power of 1% (upper)

and 0.1% (lower). Maximum laser power in LSM900 is 4 %. Scale bars, 20 μm . Even in the cells where fluorescence is detected at the lowest laser power, indicating the highest level of protein overexpression, condensates are not observed in CHD1^{ΔNC}-Venus overexpressed cells. Note that CHD1^{WT}-Venus exhibited distinct condensation patterns at both high (a-A) and low expression (a-B) levels. In contrast, CHD1^{ΔNC}-Venus displayed less condensation and a more uniform distribution throughout the nucleus even at high expression level (b-C).

2. Would it be relevant to define the assembly of droplets in dependency on the IDRs in another cancer-related (prostate) model? CHD1 expression varies quite broadly in different tumors and within the same prostate cancer samples, which is further highlighted by the relative expression in prostate cancer cell lines, with 22Rv1 expressing a relatively low level if compared with PC-3 or VCaP.

Response: Accordingly, we examined the expression levels of CHD1 across various prostate and colorectal cancer cell lines (see Figure a, below). While all cell lines expressed CHD1, the expression levels varied. We then examined the nuclear condensation of endogenous CHD1 by immunostaining three prostate cancer cell lines with differing CHD1 expression levels (See Figure b, below). The results showed a similar punctate distribution of endogenous CHD1 in these cells, suggesting that the observed differences in expression levels did not significantly affect the ability of CHD1 to undergo phase separation.

Figure for the reviewers only. CHD1 expression in prostate and colorectal cancers. a, Western blot analysis for CHD1. HSP90 is used as a loading control. **b**, Confocal immunofluorescence images of endogenous CHD1 in prostate cancer cell lines. Scale bars, 5 μm .

3. Regarding the KI experiment, author should clarify whether they analyze the CHD1 clustering in a single clone only, and if so it would be necessary to test different clones to ascertain the clonal-independency for the described phenotype.

Response: We analyzed multiple CHD1-Venus knock-in clones and found consistent condensation properties across them. To address the reviewer's concern about potential clonal variation in the FRAP assay, we additionally performed the assay on HeLa cells transiently expressing CHD1-Venus (see Figure below). This was done using non-clonal, heterogeneous cell populations (n=5). The FRAP kinetics were consistent across these cells, suggesting that the dynamic properties of CHD1 condensation are not influenced by clonal variation.

Figures for the reviewer only. FRAP analysis of overexpressed CHD1-Venus in HeLa cells. Confocal images of Spot-FRAP analysis (left panel). Scale bar, 5 μ m. FRAP signal in the bleached area before and after bleaching (right panel). Data are presented as the mean \pm SEM, $n = 5$ independent experiments.

3 (CONTINUED). In addition, the FRAP assay, as performed, is not sufficient to describe the assembly of CHD1 in condensates through LLPS.

Response: We recognize that distinguishing between LLPS and other mechanisms of protein condensation in live cells is a complex and debated issue, as highlighted in recent literature (e.g., "Evaluating Phase Separation in Live Cells: Diagnosis, Caveats, and Functional Consequences," PMID: 31594803). This is a general and fundamental question, and there is still no consensus on how to separate LLPS from other mechanisms, which is beyond the scope of this manuscript.

In response to the reviewer's concerns, we have made the following revisions: We have removed references to 'phase separation' and 'liquid-like' behavior from the Results section to avoid overinterpretation of our data. Concerning the Introduction, it has been updated to clarify the current limitations of our study and to better align with the focus of our research (see below).

Original introduction: Biomolecular condensates enable specific protein-protein interactions at distinct genomic locations through a process called liquid-liquid phase separation. The formation of various nuclear condensates, such as nucleoli, promyelocytic leukemia nuclear bodies, and paraspeckles, are essential for specific genomic functions. Phase-separated condensates formed by coactivators and transcription factors serve as specialized compartments to efficiently organize and concentrate the transcription machinery, significantly contributing to establishing and maintaining cellular identity.

Revised introduction (line49-55): Biomolecular condensates facilitate specific protein-protein interactions at distinct genomic locations through processes like liquid-liquid phase separation¹ or alternative mechanisms (PMID: 31594803)². Various nuclear condensates, such as nucleoli, promyelocytic leukemia nuclear bodies, and paraspeckles, play critical roles in genomic functions. Condensates formed by coactivators and transcription factors act as specialized compartments, organizing and concentrating the transcription machinery to establish and maintain cellular identity.

3 (CONTINUED). We strongly encourage to deep into this, by defining the biophysical properties of the endogenous CHD1 by discriminating between LLPS and other mechanisms governing the assembly of condensates. For example, the authors should consider the surface tensile forces of the droplets to determine the behavior of CHD1 within cells.

Response: We appreciate the reviewer's suggestion to further explore the biophysical properties of CHD1 condensates in vivo (in cells). However, directly measuring surface tension in vivo using atomic force microscopy (AFM) is currently technically unfeasible. While AFM is effective for measuring Young's modulus, which indicates material stiffness and elastic deformation, it

cannot measure surface energy, which represents surface tension. Typically, techniques such as optical tweezers or microrheology are used to evaluate surface tension. However, there is no specific literature on the applicable phase boundaries or conditions for our system. Therefore, incorporating these measurements into our current study within the available timeframe is not feasible.

In response to the reviewer's feedback, we have added new results to demonstrate the dynamic nature of CHD1 condensates *in vivo* (Supplementary Fig. 3, see below). We analyzed CHD1 condensates under both native and heat shock conditions. Consistent with previous findings (PMID: 31199242), CHD1 formed large nucleolar condensates upon heat shock, which gradually dissipated over several hours (Supplementary Fig. 3a-c). Given that heat shock is known to induce solid-like condensates (PMID: 31594803), we investigated the dynamics of CHD1^{WT}-Venus condensates using FRAP. Under native conditions, CHD1 condensates showed rapid fluorescence recovery within seconds (Fig. 2e, Supplementary Fig. 3e, f, below), while those formed under heat shock conditions exhibited no fluorescence recovery over several minutes (Supplementary Fig. 3e, f, below). These findings underscore the dynamic nature of native CHD1 condensates and suggest different mechanisms and functions for CHD1 condensates formed under native versus heat shock conditions. These results have been incorporated into the revised manuscript (lines 130-140, Supplementary Fig. 3).

Supplementary Fig. 3 | Heat shock induced CHD1 to form more stable gel-like condensates, different from dynamic native state. **a**, Experimental design of heat shock experiment. **b**, Heat shock induces large amorphous CHD1 condensates in nucleoli. Confocal images of untreated or heat-shocked (1 hour, 45°C) cells stained with an anti-CHD1 antibody. Scale bars, 10 μm. **c**, Confocal images of CHD1-Venus KI HeLa cells fixed at indicated time points after heat-shock recovery. Scale bars, 20 μm. **d**, Percentage of HeLa cells with CHD1 nucleolar

accumulation. Data are presented as the mean \pm SEM, $n = 2$ independent experiments. **e**, Confocal images of Spot-FRAP analysis of CHD1-Venus in asynchronous cells (upper) or heat-shocked cells (lower). Scale bars, 5 μm . **f**, FRAP signal in the bleached area before and after bleaching. The bleaching intensity was the same in each experiment. Data are presented as the mean \pm SEM, $n = 3$ independent experiments.

4. The comparison between the CHD family members does not add much to the paper and, as presented, is inconclusive. There are too many differences in terms of protein size, the possible extent of the relative IDR, expression levels, and functionality. The presented data lack the minimum number of controls to support the raised conclusions (replicates, comparison of protein level, analyses of the endogenous proteins for all the members, etc.). I would strongly suggest to simply remove this part of the work.

Response: As the reviewer pointed out, our initial data did not encompass all CHD family members and lacked information on endogenous proteins. To address this, we performed transient exogenous expression of CHD-Venus fusion proteins for all CHD family members, including CHD3, CHD5, and CHD9, in addition to those originally studied (see Supplementary Fig. 4a, right). These experiments were repeated multiple times and across several cells, consistently yielding similar results. Our findings indicate that only CHD1, CHD7, and CHD9 demonstrated condensation capability.

Supplementary Fig. 4 | a, Schematic representations of CHD family protein domains (left, upper) and analysis of PONDR scores indicating IDRs (left, lower). Confocal images of HeLa cells expressing Venus-CHD proteins demonstrate the condensation property of CHD1, CHD7 and CHD9 (right column). Scale bar, 5 μm .

Moreover, we examined the condensation properties of these proteins through immunostaining of endogenous proteins (see Supplementary Fig. 4b, below). Our results demonstrate that endogenous CHD1, CHD7, and CHD9 have condensation capabilities. Notably, CHD1 co-localized more with CHD9 than with CHD7. Additionally, exogenously expressed fluorescence-tagged CHD1 co-localized well with fluorescence-tagged CHD9, but not with fluorescence-tagged CHD7 (see Supplementary Fig. 4c, below). We believe these new data provide novel insights into which CHD family members undergo condensation. Notwithstanding the novelty of these new data, we have moved these results to Supplementary Fig. 4 and excluded them from the main figures, in keeping with the reviewer's recommendation.

Supplementary Fig. 4 | b, Immunofluorescence images showing the colocalization of CHD1 with CHD7 or CHD9 in HeLa cells. Scale bars, 5 μ m. The colocalization efficiency was assessed using Manders' coefficients (right). **c**, Confocal images showing the colocalization of fluorescence-tagged CHD1 with CHD7 or CHD9 in HeLa cells. Scale bars, 5 μ m.

5. The authors should better explain the rationale for the selected molar ratio between the recombinant CHD1 and the nucleosome in the droplet assay (Fig 4a). They should repeat the experiment carefully, considering these variables and determining their contribution to the CHD1 droplet formation.

Response: As requested by the reviewer, we repeated the droplet assays under varying conditions, titrating concentrations of DNA (Widom 601 sequence), nucleosomes, H3K4me3 nucleosomes, H3K27ac nucleosomes, and H3K9me3 nucleosomes at 10, 30, and 100 nM against 30 nM of CHD1 (Added as Supplementary Fig. 5 and Fig. 3a, see below). We observed that H3K4me3 nucleosomes significantly increased both the number and size of CHD1 condensates compared to other conditions across these concentration ranges. The differences between conditions were most pronounced at 100 nM. Consequently, we determined the molar ratio of 100 nM test samples (DNA, nucleosomes, H3K4me3 nucleosomes, H3K27ac nucleosomes, and H3K9me3 nucleosomes) against 30 nM of CHD1.

Supplementary Fig. 5 | H3K4me3-modified nucleosomes promote CHD1 condensation in dose-dependent manner. a, Confocal images of CHD1^{WT}-Venus condensates. 30 nM of CHD1^{WT}-Venus was mixed with DNA or nucleosomes at the indicated concentration. Scale bars, 5 μ m. **b, c**, The occupancy (**b**) and average size (**c**) of CHD1^{WT}-Venus condensates in 1.03×10^{-2} mm² area. Seven fields were randomly selected.

Fig. 3 | H3K4me3 nucleosome and RNA stimulates CHD1 condensation. a, Droplet assay demonstrating that H3K4me3-modified nucleosomes promote CHD1 condensation. CHD1-Venus at 30 nM was mixed with 100 nM of Widom 601 DNA, unmodified nucleosomes, H3K4me3-, H3K27ac-, or H3K9me3-modified nucleosomes (left). Scale bars, 5 µm. Condensates within the 1.03×10^{-2} mm² area are represented using boxplots, illustrating the median, interquartile range, and 10th–90th percentiles denoted by black bars, white boxes and thin black lines, respectively. The size of each condensate was plotted as a green dot (right).

We observed aggregation defined by the non-circular shapes (the arrow in the Fig. 1 below), when 300 nM of DNA (Widom 601 sequence) was added to 30 nM of CHD1. Therefore, the concentration of the test samples was limited to below 100 nM.

Fig. 1 for the reviewer only. The excess of DNA induces CHD1 phase transition from condensates to aggregation. 30 nM of CHD1WT-Venus was mixed with TAMRA (Tetramethylrhodamine)-labeled DNA. Droplet assay results demonstrating that CHD1 condensates uptake DNA but start to form aggregates with 300 nM of DNA. Scale bars, 5 µm.

6. The same concern holds true for the spiked RNA molecules, as the relative abundance of RNA can vary the number, size, and timing of droplet assembly. In addition, it is not clear which RNA has been labeled, the rationale of this choice, and whether the length of the RNA, its potential secondary structures, or sequence biases may determine the contribution of RNA species in guiding CHD1 assembly into droplets.

Response: The RNA used in this study was Xenopus EF1α mRNA, as described in the Method section. Firstly, we determined the concentration of the RNA below 30 nM because aggregation was observed when 30 nM of EF1α mRNA was added to 30 nM of CHD1 (the arrow in the Fig. 2 below).

Fig. 2 for the reviewer only. The excess of RNA induces CHD1 phase transition from condensates to aggregation. 30 nM of CHD1^{WT}-Venus was mixed with Cy5-labeled *Xenopus EF1α* mRNA. Droplet assay results demonstrating that CHD1 condensates uptake RNA but start to form aggregates with 30 nM of DNA. Scale bars, 5 μ m.

We conducted droplet assays under various conditions by titrating the concentrations of EF1 α mRNA at 0.3, 1, 3, and 10 nM against 30 nM of CHD1 (Added as Fig. 3b and supplementary Fig. 6, see below). We found that EF1 α mRNA consistently increased CHD1 condensation within these concentration ranges. Interestingly, CHD1 condensation was most enhanced at 1 nM of EF1 α mRNA, forming a bell-shaped curve around this concentration. This is consistent with previous reports that RNA-induced condensation is primarily driven by charge and has an optimal concentration (Henninger, J. E. et al. RNA-Mediated Feedback Control of Transcriptional Condensates. Cell 184, 207–225 e224 (2021). PMID: 33333019). The resulting data have been included as revised Fig. 3b and Supplementary Fig. 6.

Fig. 3 | b, Droplet assay demonstrating RNA-induced condensation of CHD1 and its incorporation into the condensates. 30 nM of CHD1^{WT}-Venus was mixed with 5 mM Tris buffer (Mock) or 1 nM of Cy5-tagged synthetic EF1 α RNA (left). Scale bars, 5 μ m.

Supplementary Fig. 6 | RNA promotes CHD1 condensation in dose-dependent manner. **a**, Confocal images of droplet assay at indicated concentration of RNA. Scale bars, 5 μm. **b**, **c**, The occupancy (**b**) and average size (**c**) of CHD1^{WT}-Venus condensates in 1.03×10^{-2} mm² area. Seven fields were randomly selected.

6 (continued). Are RNA molecules involved in the nucleation or in the growth of the droplets? Time course analyses could help distinguish these aspects of LLPS.

Response: Using a confocal microscope, we observed that CHD1 condensates form almost instantaneously, within seconds, making it challenging to capture these events with time-lapse imaging. In our time-lapse analyses spanning seconds to hours, we did not observe a significant increase in the number or size of CHD1 condensates (data not shown). However, endpoint observations revealed that the addition of RNA led to an increase in both the size and number of CHD1 condensates, suggesting that RNA may play a crucial role in both nucleation and growth. Although it would be interesting to explore the physicochemical influence of RNA on CHD1 condensation in detail, we hope the reviewer appreciates that it is beyond the main focus of our study.

7. ChIP-seq analyses suggest that the C-term IDR contributes to CHD1 chromatin recruitment at active promoters. Although of interest, the presented data do not support entirely the raised conclusion. As highlighted in ED Fig. 7a, the rescue of CHD1 KO cells with either the WT or the C-term IDR deletion showed a remarkable difference in the protein levels, which per se can explain the retrieved results. The authors should repeat these assays using a cellular model with comparable (and properly quantified) expression levels of both WT and IDR-deleted IDR, which should be similar to the endogenous level, thus representing a rescue experiment setting.... The drawn conclusion on the contribution of the IDR on the cell cycle control and p53 pathway suffers from the same limitation (not properly rescuing CHD1 level).

Response: The data referenced by the reviewer were obtained using a commercial CHD1 antibody that specifically recognizes the C-terminus of CHD1, which does not detect CHD1^{ΔNC}. We have clarified this point both in the original figure and its legend to prevent

misunderstandings. Furthermore, protein levels of CHD1^{WT} and CHD1^{ΔNC} in the cell lines used in the experiment are nearly identical, as demonstrated by immunoblotting against the HA tag (refer to the relocated original figure now in Supplementary Fig. 9a, and see the figure below from an independent experiment).

7 (continued). To better normalize the data, this kind of analysis requires the usage of calibrated ChIP-seq by spiking-in mouse (or Drosophyla) cells expressing a-chromatin protein before performing the IP.

Response: Several published studies have indicated that there is not a universally accepted methodology for spike-in normalization for transcription factor ChIP-seq (Doerfler, P. A. et al. Nat Genet 53, 1177–1186 (2021), PMID: 34341563). Therefore, we employed the S3norm method for in silico normalization of ChIP-seq data, which accounts for sequencing depth and signal-to-noise ratio (Xiang, G. et al. Nucleic Acids Res 48, e43 (2020), PMID: 32086521). This method allowed for a quantitative comparison of ChIP signals between experimental groups (see Methods, lines 722–724. Our revised results, presented in Fig. 4b–d (see below), are consistent with the original findings, supporting that our overall conclusions remain unchanged. Notably, significant differences between experimental groups were observed only within the defined regions in panels c and d, while the surrounding areas exhibited similar ChIP peak profiles. These updated findings have been incorporated into the revised manuscript (Fig. 4b–d).

Fig. 4 | IDR is necessary to recruit CHD1 to active promoters. b, Heatmap displaying ChIP-seq read densities around transcription start sites (TSS) and gene centers of the entire genome (left), TSS at active promoters (H3K4me3-positive, middle), and unmodified promoter (H3K4me3-negative and H3K27ac-positive, right). CHD1-WT peaks were predominantly found at TSS, especially at H3K4me3-positive promoters. In contrast, CHD1- Δ NC peaks displayed a distinct pattern compared to CHD1-WT, showing a higher tendency to gene body and lower specificity for H3K4me3-positive TSS. **c, d,** ChIP-seq binding profiles of WT, Δ NC, H3K4me3, and H3K27ac on intergenic and intronic Δ NC-specific sites (boxed) (**c**) and the promoters of tumor suppressor genes [note the reduced Δ NC ChIP-seq signals at the TSS (boxed) of these gene loci] (**d**).

7 (continued). In addition, as CHD1 is a chromatin remodeler, it is essential to determine whether any possible alteration of its chromatin function through the IDR involves a perturbation of the histone turnover at the NFR of promoters. This can be assessed either by MNase-seq or by ATAC-seq.

Response: We appreciate the reviewer's insightful comment. To address this, we performed ATAC-seq to investigate the differences in nucleosome-free regions between CHD1^{WT} or CHD1^{ΔNC}-transduced mCAT-HeLa cells. (Added as Supplementary Fig. 8, see below). ATAC-seq revealed substantial changes in open chromatin (5083 peaks) and closed chromatin (2455 peaks) in CHD1^{ΔNC} cells compared to CHD1^{WT} cells (see Supplementary Fig. 8a, below). GREAT analysis revealed that the ATAC-seq peaks lost in CHD1^{ΔNC} (closed chromatin) are enriched for terms related to the apoptotic pathway, while those gained in CHD1^{ΔNC} (open chromatin) are enriched for terms related to the metabolic/catabolic pathway (see Supplementary Fig. 8b, below). This suggests that CHD1^{ΔNC} may have a positive effect on cell growth.

Additionally, at the level of individual genes, ChIP-revealed binding sites of CHD1^{WT} or CHD1^{ΔNC} exhibit a positive correlation with chromatin accessibility as revealed by ATAC-seq. Specifically, at sites where CHD1^{WT} binds but CHD1^{ΔNC} does not, ATAC-seq signals are higher in cells expressing CHD1^{WT} compared to CHD1^{ΔNC} (see Supplementary Fig. 8d, e, below).

To explore whether the reduction in CHD1^{ΔNC} ChIP-seq peaks around transcription start sites (TSS) compared to CHD1^{WT} correlates with decreased chromatin accessibility at promoter regions across the genome, we analyzed ATAC-seq data from promoter regions in both CHD1^{WT} and CHD1^{ΔNC} cells. The analysis showed no significant differences in chromatin accessibility (see Supplementary Fig. 8c, below). It is possible that CHD1 may contribute to local chromatin opening for RNA polymerase II during transcription elongation—a dynamic process that could vary among different cell populations and might not be captured by ATAC-seq in the whole cell population. These findings have been incorporated into the revised manuscript (lines 208–218, Supplementary Fig. 8).

Supplementary Fig. 8 | Chromatin accessibility differences between CHD1^{WT} and CHD1^{ΔNC}-expressing cells.

a, Comparison of ATAC-seq peaks in CHD1^{ΔNC}-expressing HeLa cells versus CHD1^{WT}-expressing HeLa cells. CHD1^{ΔNC}-expressing cells exhibited a gain of 5,083 peaks and a loss of 2,455 peaks relative to CHD1^{WT}-expressing cells. **b**, Gene ontology analysis of the peaks gained or lost in CHD1^{ΔNC}-expressing cells. **c**, Metaplot showing the distribution of ATAC-seq signal within a 3 kb window around TSS (left) and within active promoters (right) across the genome. **d**, **e**, Representative ATAC-seq peaks at target genes of CHD1^{WT} (**d**) and CHD1^{ΔNC} (**e**).

8. The in vivo tumorigenic assays to determine the contribution of CDH1E1321fs are highly relevant and well-controlled. However, using only three mice/conditions for a short time window is not fully appropriate to dissect the contribution of the mutant to the complex process of tumorigenesis. If possible, we suggest repeating the analyses by injecting a lower number of cells and extending the analyses at later points to ensure that different kinetics of tumor growth can explain the phenotype.

Response: In response to the reviewers' suggestions, we conducted additional tumor formation assays in mice, reducing the number of injected cells to allow for a more prolonged observation of tumor development. Regardless of the quantity of injected tumor cells (5.0×10^5 , 1.0×10^5 , 2.0×10^4), 22Rv1 $\Delta ex29/\Delta ex29$ cells showed significantly reduced tumor formation compared to the 22Rv1 $^{WT/fs}$ cells (Added as Extended Data Fig. 7a, b, see below). This confirms the contribution of CHD1 E1321fs to tumorigenicity.

Extended Data Fig. 7 | 22Rv1 $\Delta ex29/\Delta ex29$ clones exhibit reduced tumor growth and less pronounced EMT phenotypes in mice. a, b, Subcutaneous tumor growth of WT/fs 22Rv1 cells with sgRNA nontargeting control and $\Delta ex29/\Delta ex29$ clones in athymic nude mice ($n = 4$). 1.0×10^5 (a) and 2.0×10^4 (b) cells were injected. The right panels present tumor volume and weight as the mean \pm SEM from four independent experiments. Differences in

tumor volume and weight were analyzed using two-way ANOVA and unpaired Student's t-test, respectively. **c-f**, Representative immunohistochemistry images of tumor tissue for Ki-67 (brown, **c**) as a marker of proliferative cells, and integrin alpha 5 (ITGA5, brown, **d**) TWIST1/2 (brown, **e**) or SNAI2 (brown, **f**) as a marker of EMT. The tissues were counter stained with Mayer's Hematoxylin and Eosin. Scale bars, 100 μ m. The right panels present quantitative analysis of percentages of positive cells (Ki-67, TWIST1/2, or SNAI2) or positive area (ITGA5). Student's t-test.

8 (continued). In addition, it is recommended to perform a phenotypic characterization of the formed tumors to determine the proliferative index (either using Ki67 or H3Ser10ph), the level of apoptosis, and eventually, the EMT phenotype/invasiveness.

Response: In response to the reviewer's recommendations, we performed a series of immunohistochemistry analyses to characterize tumors based on proliferative markers (Ki-67), apoptosis levels (TUNEL staining), and the epithelial-to-mesenchymal transition (EMT) phenotype using ITGA5, TWIST1/2, and SNAI2. Our findings revealed that tumors harboring E1321fs demonstrated a significantly higher Ki-67 index and increased expression of EMT-related proteins compared to Δ ex29 cells (Added as Extended Data Fig. 7 c-f, see above). We did not observe a significant difference in the levels of apoptotic cells (data not shown). These results are consistent with our RNA-seq data, which indicated upregulation of EMT-related genes and genes associated with cell proliferation in the E1321fs mutant. These results have been added in the revised manuscript (lines 258-260, Extended Data Fig. 7c-f).

9. In relation to the mapping of lincRNAs interacting with CHD1, although the analyses are of high relevance, they do not support the conclusion that this class of RNA species supports the CHD1 assembly in condensates. They indicate that some lincRNAs (whose promoter is not a target of CHD1) are in proximity to CHD1 but do not tell much about their function nor their contribution to the formation of condensates. Either the authors provide some data to support these conclusions, or this part of the manuscript should reconsider.

Response: As noted by the reviewer, our study has not definitively demonstrated the functional interaction between the proximity biotinylation-identified lincRNA and CHD1. In response, we have acknowledged this limitation in the Results section and adjusted our conclusions accordingly as below.

Revised statement in the Result section: Approximately 24.7% of RNAs in proximity to CHD1 were lincRNAs (Fig. 7b), but almost none of them were CHD1 target genes, unlike the coding RNAs (Fig. 7c). ~~This suggests that their recruitment might be functionally important for the CHD1 condensates. GSEA with LncSEA2.0..... These findings highlight the diverse roles of lincRNAs in CHD1 condensates and their potential involvement in various cellular processes and cancer. These findings suggest a potential functional interaction between lincRNAs and CHD1 condensates. (lines 295-308)~~

Revised statement in the Discussion section: While our transcriptome and interactome analyses have identified multiple cancer-related pathways and molecules, further functional validation in future studies is required to fully understand these oncogenic processes. (lines 412-415)

9 (Continued). For example, they could test the specificity of some of the identified lincRNAs in guiding the assembly of the droplets using an in vitro droplet assay. The same approach could be tested in vitro by perturbing the relative abundance of the most (Cancer-related) relevant lincRNAs.

Response: The consideration of potential differences among various lncRNAs in facilitating CHD1 condensation is intriguing. However, we believe that RNA's role in promoting liquid-liquid phase separation is primarily driven by its fundamental physical property—namely, its negative charge (Henninger, J. E. et al. Cell 184, 207–225 (2021), PMID: 33333019)—rather than by specific characteristics of individual RNA sequences. We have added discussion as below.

Revised statement in the Discussion section (lines 396–401): Additionally, the colocalization of CHD1 with certain lncRNAs may also be indirectly mediated by these transcription factors and co-factors, which interact with lncRNAs⁴⁸(PMID: 28959731). Once colocalized, lncRNAs may enhance CHD1 condensation through their negative charge rather than specific RNA sequences⁶(PMID: 33333019), contributing to functional compartmentalization within the nucleus.

10. Regarding the differential contribution of chromatin factors interacting with CHD1 through the C-term IDR, the proteomic analyses should be better supported by orthogonal experiments. For example, the IF analyses require co-localization analyses (Fig. 9c).

Response: To address this, we quantified CHD1 colocalization with SUZ12 by Manders' coefficients for each protein (Added as Fig. 8d, see below). We used SC35, a splicing factor involved in mRNA maturation, as a negative control, as it is not recruited in CHD1-condensates. Conversely, H3K4me3 and RNA polymerase II phosphorylated at serine 5 are known as CHD1 related protein, so we used them as positive controls. These results have been added in the revised manuscript (lines 322–324, Fig. 8d).

Fig. 8 | d, Confocal images showing the colocalization of CHD1 and ASH2L, SUZ12, SC35, H3K4me3, or RNA polymerase II phosphorylated on serine 5 (pS5) in HeLa cells. Scale bars, 5 μ m. Colocalized fractions of CHD1 with indicated proteins were quantified based on Manders' coefficient. ASH2L and SUZ12 are recruited into CHD1 condensates. SC35, a splicing factor involved in mRNA maturation, is not recruited. H3K4me3 and Pol2(pS5) are known to interact with CHD1. Data are means \pm SEM of forty cells in at least two independent experiments.

10 (continued). We would also suggest to repeat the same experiments using the rescued cells, after controlling that the same expression level is achieved (that should be similar to the endogenous CHD1).

Response: As requested by the reviewer, we investigated whether SUZ12 co-localizes with CHD1 condensates in HeLa cells rescued with CHD1^{WT} or CHD1^{ΔNC}. These cells exhibited similar expression levels of CHD1^{WT} and CHD1^{ΔNC} following lentivirus infection, as shown right.

To assess the co-localization between CHD1 and SUZ12, we performed immunofluorescence staining for both proteins. As illustrated in Extended Data Fig. 9a, b (below), CHD1^{WT} displayed greater co-localization with SUZ12 compared to CHD1^{ΔNC}. These results have been added in the revised manuscript (lines 324–326, Extended Data Fig. 9a, b).

To further support the evidence of co-localization between CHD1 and SUZ12 through another orthogonal method, we conducted a Proximity Ligation Assay (PLA) between CHD1 and SUZ12. Consistent with the immunostaining results, more PLA signals were detected in cells expressing CHD1^{WT} than in cells expressing CHD1^{ΔNC} (Added as Extended Data Fig. 9c, d, see below). These results have been added in the revised manuscript (lines 326–329, Extended Data Fig. 9c, d)

Extended Data Fig. 9 | Colocalization of CHD1 and SUZ12 condensates.

a, Confocal immunofluorescence images showing the colocalization of CHD1 with SUZ12 in CHD1KO mCAT HeLa transduced to express CHD1^{WT}-Venus or CHD1^{ΔNC}-Venus. Scale bars, 5 μm.

b, Quantification of the CHD1 colocalization with SUZ12 in mCAT HeLa cells. Colocalized fractions of CHD1 were quantified based on Manders' coefficient. Data are means ± SEM of forty cells in at least two independent experiments.

c, PLA using anti-GFP antibody and anti-SUZ12 antibody (top) or normal IgG (bottom) on CHD1KO mCAT HeLa cells expressing either CHD1^{WT} or CHD1^{ΔNC}-Venus. Scale bars, 5 μm.

d, Quantification of PLA signals.

We thank the reviewers for the careful evaluation of our manuscript, providing insightful feedback aimed at strengthening our manuscript. In the subsequent paragraphs, we address the comments raised by each reviewer. Our responses are in blue.

Reviewer #1 (Remarks to the Author):

In the revised version, the authors satisfactorily addressed all my initial comments with convincing original data. The current version of the manuscript is now very strong and acceptable for publication.

Response: We sincerely thank the reviewer for their positive feedback. We are glad that the revisions and additional data have satisfactorily addressed your concerns, and we appreciate your recognition of the manuscript's improved quality.

Reviewer #2 (Remarks to the Author):

The authors have addressed all of the concerns raised in my initial review. The authors have added a significant amount of new data in support of their conclusions. The revised manuscript is suitable for publication.

Response: We thank the reviewer for their positive feedback. We are glad that the additional data and revisions have addressed your concerns and that the manuscript is now considered suitable for publication.

Reviewer #3 (Remarks to the Author):

The authors provided supportive data to address some of the raised concerns, which improve the robustness of the manuscript. However, some important issues have not been addressed. Specifically, referring to the specific raised points:

1. The author did not address the discrepancy between the size of the droplets obtained from the overexpressed isoform compared to the KI system. They indeed showed that they have heterogeneous expression of the both N- and C-terminus CHD1, yet they are confident that the expression level did not impact on the protein distribution. This conclusion is actually not supported by the presented data in which they do not show quantitative measurements of the number and size of the droplets in respect to the expression level. The comparison of the

low- vs high-expressing cells do show a clear difference in the distribution and number of the formed droplets. Moreover, the Delta-N construct delineate a very peculiar distribution of the protein, which seemly resides in heterochromatin regions: why? Additionally, they did not show the relative level of OE with respect to the endogenous level, as requested.

Response: Rather than discrepancy, we considered it natural that when a protein is expressed at a higher level it will cause a shift in the size and numbers of condensation due to the checks-and-balances in the underlying molecular mechanisms. Such variability in protein condensation between the endogenous and overexpression systems is a well-documented phenomenon, and similar observations have been reported in numerous other systems.

Our conclusion that "CHD1 condensation is primarily mediated by the IDRs in the N- and C-termini" is firmly supported by the data presented throughout the manuscript, including the additional experiments (Figure 1 below) provided in response to the reviewer's earlier comments.

1(continued). Moreover, the Delta-N construct delineate a very peculiar distribution of the protein, which seemly resides in heterochromatin regions: why?

Response: We imagine the reviewer was referring to the distribution pattern of the Delta-NC construct (and not Delta-N), as shown in the Figure 1b below. The Delta-NC CHD1 variant, which failed to condense (Fig. 2a, b and below Figure 1b), displays a broader nuclear distribution, with slightly higher intensity in heterochromatin regions. This behavior is likely due to the increased affinity of Delta-NC for nucleosomes (Fig. 5) compared to the wild-type construct, which may lead to its accumulation in more condensed chromatin regions, such as heterochromatin.

Figure 1 for the reviewers only. Evaluation of condensate propensity in HeLa cells overexpressing fluorescent tag fused proteins. a, Overexpression of CHD1^{WT} in HeLa cells. Confocal images were taken at laser power of 1 % (upper) and 0.1 % (lower). Maximum laser power in LSM900 is 4 %. Scale bar, 20 μ m. **b,** Overexpression of CHD1 ^{Δ NC} in HeLa cells. Even in the cells where fluorescence is detected at the lowest laser power, indicating the highest level of protein overexpression, condensates are not observed.

1 (continued). Additionally, they did not show the relative level of OE with respect to the endogenous level, as requested.

Response: It would have been relatively simple to provide the requested data in the form of a Western blotting. However, as the reviewer would appreciate, transiently transfected cells expressed the exogenous CHD1 proteins at varying levels, due to the inherent heterogeneity and differential plasmid uptake within a cell population. This heterogeneity is readily observable in our figures involving Figure 1 for the reviewers only (above). Hence, we have specifically selected images of cell nuclei displaying saturated CHD1 expression for both CHD1^{WT}-Venus and CHD1 ^{Δ NC}-Venus, as depicted in Fig. 2b and 5b.

2. The author concluded that although different cell lines expressed different levels of CHD1, this does not impact on its distribution. This conclusion is not fully supported by the data as they did not quantify the number and size of the droplets and relate it to the overall change in protein abundance, that should be also confirmed by IF, at least. What happens if the same cancer cell line they perturb the level of the endogenous CHD1?

Response: Following the reviewer's suggestion from the first revision, we investigated the expression levels of CHD1 across various prostate cancer and colorectal cancer cell lines (see Figure 2 below). While all the cell lines expressed CHD1, there was notable variability in the expression levels. We then examined the nuclear condensation of endogenous CHD1 in three prostate cancer cell lines through immunofluorescence (IF) and observed remarkably similar punctate distribution across all lines, irrespective of their expression levels. This observation shows that *even if* there exists a relationship between protein abundance and droplet size/quantity, it did not appear to impact on the distribution of endogenous CHD1 condensates.

Figure 2 for the reviewers only. CHD1 expression in prostate and colorectal cancers. a, Western blot analysis for CHD1. HSP90 is used as a loading control. **b**, Confocal immunofluorescence images of endogenous CHD1 in prostate cancer cell lines. Scale bars, 5 μ m.

Moreover, our data overwhelmingly indicate that 1) the IDRs are crucial for CHD1 condensation, and 2) their contribution is qualitative and not quantitative. These points are further supported by exogenous expression experiments (see Figure 1 above) that showed that, even at high expression levels, CHD1 ^{Δ NC}-Venus displayed markedly reduced condensation, and was uniformly distributed throughout the nucleus. This was strikingly different to that of CHD1^{WT}-Venus, which exhibited distinct condensation patterns at both high and low expression levels (see Figure 1 above). This provides further support to our conclusion that condensation is primarily mediated by the IDRs in the N- and C-termini of CHD1, again, independent of expression levels.

We trust that this explanation sufficiently clarifies the issue and addresses the reviewer's concerns.

3. The authors stated they analyzed different KI clones, which behaved similarly without major differences. Nice to know, but they did not provide any data supporting this statement.

Response: We present here the data showing that the different KI clones behaved similarly, as illustrated below.

Figure 3 for the reviewers only. **a**, Schematic illustrating the strategy for CHD1^{WT}-Venus knock-in (KI) HeLa cell generation. Screening was performed using PCR with primers designed inside and outside the repair template. **b**, PCR results after limiting dilution. **c**, Confocal images of Spot-FRAP analysis on CHD1^{WT}-Venus KI HeLa cell clones (#1, 2, 3).

3 (continued). They instead show data on HeLa cells in which they OE the construct to then perform FRAP assay: why? This experiment does not address the original question. On the Same point, they argue that determining the biophysical properties of CHD1 is outside the scope of this work. I disagree completely as the drawn conclusions of this work rely on the potential contribution of CHD1 to form condensates, which modulated the chromatin remodeling and can influence tumor progression.

Response: Firstly, the FRAP assays on HeLa cells in previous revision were conducted in non-clonal heterogeneous cell populations (n=5) to account for potential clonal variation, which was a concern raised by the reviewer. The FRAP kinetics were consistent across these cells, indicating that the dynamic properties of CHD1 condensation were not unduly influenced by clonal variation. Concerning the reviewer's original question, which we copy as follows:

In addition, the FRAP assay, as performed, is not sufficient to describe the assembly of CHD1

in condensates through LLPS. We strongly encourage further exploration by defining the biophysical properties of endogenous CHD1, distinguishing between LLPS and other mechanisms governing condensate assembly. For example, the authors should consider the surface tensile forces of the droplets to determine the behavior of CHD1 within cells.

We maintain that the precise biophysical nature of the CHD1 condensate – specifically, whether they were the products of *bona fide* LLPS or other competing mechanisms – is beyond the scope of the current study. The reviewer may recall that this work pertains to the contribution of CHD1 IDRs in human carcinogenesis. Indeed, much of our data centered on the downstream effects of CHD1's loss-of-function through a multi-omics approach. This guided us to the discovery of co-occurrence between CHD1 and MLL mutations in diverse human cancers, which indicative of a common pathway of carcinogenesis. Crucially, the light shed on the interactome of CHD1 in this study offers the first glimpse of the underlying epigenetic basis for this hitherto unknown association. That CHD1 condensates are truly the products of LLPS does not add nor detract from the overall conclusions from this study. As such, at the suggestion of the reviewer, we have removed references to "LLPS" and "liquid-like" behavior in the manuscript.

4. The authors still consider necessary to describe the behavior of the other CHD proteins. They perform additional data that instead of clarifying the potential differences among this diverse family of remodellers, it adds only confusion. For example, how do they explain the inconsistency of the pattern distribution of the endogenous Chd7 vs the OE form, which showed a clear heterochromatin pattern? This figure is irrelevant for the manuscript, there are no quantifications (just descriptive) and inconsistent.

Response: As this study represents the first experimental examination of the condensation properties of CHD family members, we feel it an opportunity to include these data as supplemental information (Supplementary Fig. 4a), so that the scientific community may assess the properties of other CHD family members in light of our in-depth analyses of CHD1. Despite being predicted to contain IDRs, these proteins had not been previously tested for condensation behavior. Although we did not include quantitative analysis, we believe the descriptive data provide valuable insights into the potential roles of these proteins in cellular processes. Therefore, we consider these findings suitable for inclusion as supporting information. However, we agree with the reviewer that the colocalization analysis of CHD7 or CHD9 condensates with CHD1 (Supplementary Fig. 4b and 4c) is preliminary and does

not yet provide strong conclusions regarding their biological functions. As such, we have excluded these data from the revised manuscript.

5. The authors provided relevant controls to better titer the abundance of the recombinant CHD1 and the co-mixed nucleosomes, strengthening the significance of these data. Yet they should comment why another methylated nucleosome (H3K9me3) increased the droplet formation. Is it possible that the in vitro assay conditions do not permit to assess the specificity of CHD1 chromodomains towards the K4me3? This point should be discussed.

Response: We appreciate the reviewer's comment regarding the impact of H3K9me3 on droplet formation, although its effect is much weaker compared to H3K4me3. We acknowledge that the conditions, methods, and readout of in vitro droplet assays differ from in vivo assays, such as ChIP-seq, which demonstrate CHD1 specificity for H3K4me3. Alternatively, this observation may be explained by the potential binding of chromodomains to H3K9me3 (e.g., CHD3 or HP1 α , PMID: 21288002), which could contribute to CHD1 binding to H3K9me3 under certain conditions. We have added the following second sentence to the Results section:

The results indicated that H3K4me3 nucleosomes significantly increased both the number and size of CHD1 condensates, while others had minimal effects (Fig. 3a, Supplementary Fig. 5). The slight increase in CHD1 condensation with H3K9me3-modified nucleosomes may be attributed to potential interactions between chromodomains and H3K9me3 (PMID: 21288002).

6. Regarding the role of nascent RNAs in modulating CHD1 condensates, the authors argues that CHD1 behaves similarly to other chromatin factors such as MED1 and BRD4 whose condensates formation is modulated by electrostatic-driven RNA-protein interaction (PMID: 33333019). However, the authors failed to address our criticisms as they do not provide any data supporting this hypothesis. For example, if this prediction is correct, then other polyanionic molecules should give rise to similar results. In the same mentioned work, the hypothesis of charge balancing predicted that RNA synthesis should impact on condensates formation, and they showed experimentally that RNAP II inhibition increased the size of the condensates, which is the opposite of what measured in this manuscript (Extended Data Fig. 6a-e). How do the authors explain this contradictory results (respect to the charge balancing

mechanisms)?

Response: we referred to the study by Henninger et al. (2021) (PMID: 33333019), which proposed that RNA can enhance condensate formation, but excessive RNA levels may lead to condensate dissolution. Regarding the reviewer's concern about the differences in results observed with actinomycin D treatment, it is indeed possible that the experimental conditions in their study and ours differ, which may account for the discrepancies.

In Henninger et al.'s work, cells were treated with 1 μ M actinomycin D for 30 minutes, which primarily inhibits RNA elongation and leads to a burst of nascent RNA synthesis, thereby promoting condensate dissolution. In contrast, in our study, we used 20 μ M actinomycin D for 2 hours to inhibit nascent RNA synthesis and significantly reduce total RNA levels, as confirmed by the substantial reduction in RNA levels (Extended Data Fig. 6c). It is possible that the different treatment conditions led to divergent effects on RNA dynamics: Henninger et al.'s conditions inhibited RNA elongation, leading to increased nascent RNA synthesis and subsequent dissolution of condensates, while our treatment resulted in a reduction in cellular RNA levels.

We interpret these findings as consistent with the hypothesis that RNA enhances condensate formation, but excessive RNA can lead to condensate dissolution. Although the experimental conditions differ, both sets of results are aligned with this overarching idea.

6 (continued). Referring to the same experiments, they concluded that RNA synthesis inhibition reduced CHD1 condensates; however, the shown representative IF images indicated a clear decrease of CHD1 abundance, which seemingly was not detected by WB. This incongruency needs to be addressed by using independent biological replicates and by performing adequate statistical analyses.

Response: As shown in Extended Data Fig. 6a (below), the representative IF images demonstrate a clear reduction in the fluorescence intensity of CHD1 condensates (above the dashed line) after α -amanitin treatment, compared to the DMSO control. However, the fluorescence intensity of the CHD1 signal distributed throughout the nucleus (below the dashed line) remains unchanged. Since CHD1 condensates represent only a small fraction of total nuclear CHD1, the overall mean fluorescence intensity primarily reflects the distributed, non-condensed CHD1. Thus, the total mean fluorescence intensity in the nucleus (Extended

Data Fig. 6c) and the Western blotting results (Extended Data Fig. 6d) show no significant change.

Extended Data Figure 6 | Inhibition of RNA polymerases or histone methyltransferases leads to a reduction in CHD1 condensation within cells. **a**, Representative immunofluorescence images showing the CHD1 condensates under the effects of RNA polymerase II inhibition. Scale bar, 5 μ m. The fluorescence intensity profiles (bottom) were taken along the white dashed line. High condensation areas of CHD1 were defined using a cutoff intensity of 1.0×10^4 (indicated by the dashed line). **b**, The percentage of high condensation areas of CHD1 in each nucleus was plotted as individual dots and analyzed using an unpaired Student's t-test. **c**, Mean fluorescence intensity of CHD1 for each nucleus was plotted as individual dots in data from more than 42 cells across three independent experiments. Unpaired student's t-test. **d**, Western blot analysis for CHD1 and H3K4me3. **e**, Maximum fluorescence intensity of CHD1 for each nucleus was plotted as individual dots. **f,g,h**, The effects of RNA polymerase II inhibition on cell viability (**f**), total cellular RNA amount (**g**), and relative Cyclin E1 mRNA expression (**h**). **i**, Representative immunofluorescence images showing the CHD1 condensates under the effects of sinefungin (Sine), a methyl-transferase inhibitor. Scale bar, 5 μ m. **j**, Western blot analysis for CHD1 and H3K4me3 in the presence of sinefungin. **k**, Percentage of high condensation areas of CHD1 (a cutoff intensity of 6.0×10^4) in each nucleus was plotted as individual dots and analyzed using an unpaired Student's t-test (right).

6 (continued). The control of the drug efficacy in inhibiting transcription elongation is incorrect as they are measuring the steady state of total RNA (which is dominated by the large abundance of rRNAs and tRNAs) and not the RNA synthesis. In sum these set of experiments did not addressed the raised criticisms.

Response: We have supplemented our analysis by providing additional data on the reduction of *cyclin E1* mRNA levels following α -amanitin treatment (Extended Data Fig. 6h). Additionally, the reduction of total cellular RNA (Extended Data Fig. 6c) supports the efficacy of actinomycin D treatment, as this compound broadly inhibits RNA polymerases. While it is true that measuring steady-state RNA levels may have limitations in directly assessing RNA synthesis, the combination of these results strengthens our interpretation that actinomycin D and α -amanitin effectively reduces RNA synthesis.

Extended Figure 6h. RT-qPCR analysis of *Cyclin E1* expression levels normalized to *GAPDH*.

7. The protein level of the rescued experiments has not been quantified by WB or other means. This point should be carefully addressed, as it can strongly impact on the data interpretation.

Response: It seems there may be a misunderstanding, as we have provided Western blot (WB) data for the rescued experiments. Specifically, the protein levels of the rescued constructs were quantified by WB and are presented in Supplementary Figure 9a, along with a biological replicate (shown in the figure for the reviewer only in the first revision: right).

7 (continued). We understand the difficulties of having a single Ab recognizing all the protein mutants, but the author can reach quantitative information's by performing shotgun

proteomics using as reference peptides shared among all the CDH1 isoforms.

Response: We would like to clarify that WB results have been provided for all the variants generated in our study. The reviewer's comment might have been a reference to Reviewer #2's suggestion to detect of the endogenous CHD1^{E1321fs} mutant. As no antibody currently exists that can distinguish between CHD1^{WT} and CHD1^{E1321fs}, to address Reviewer #2's comment, we instead performed a knock-in (KI) reporter assay to confirm the expression of the mutant protein and demonstrate its escape from mRNA decay (Supplementary Figure 2). Reviewer #2 was satisfied with this alternative approach given the current technical limitations and had earlier accepted these new supporting data. Lastly, with regards to Reviewer #3's suggestion to use shotgun proteomics for quantitative analysis, this method requires considerable optimization, and its implementation within the scope and timeline of our current study is not feasible.

7 (continued). Similarly, the authors did not perform calibrated ChIP-seq nor orthogonal approach to properly analyze the retrieved data. Without these controls, the comparative analyses do not support the raised conclusions.

Response: The reviewer might have missed it – but we have already addressed this in our previous revision. Specifically, we have reanalyzed our data using the orthogonal approach (the S3norm method) for *in silico* normalization of ChIP-seq data, which accounts for sequencing depth and signal-to-noise ratio (Xiang, G. et al. Nucleic Acids Res 48, e43 (2020), PMID: 32086521). This method allowed for a quantitative comparison of ChIP signals between experimental groups (see Methods, lines 722-724). Our revised results, presented in Fig. 4b-d, are consistent with the original findings.

8. The additional experiments related to the *in vivo* tumorigenic assay fully addressed the raised criticisms.

Response: We thank the reviewer for accepting the new *in vivo* data.

9. The authors tuned down their conclusions on the potential role of lincRNAs in mediating CHD1 condensates, in line with the raised criticism.

Response: We thank the reviewer for accepting the revision of our conclusion pertaining to the role of lncRNAs.

10. The proteomics results through which the authors identified other chromatin factors interacting with CHD1 have been properly validated, although the limitation regarding the rescue experiments related to the proper quantification of CHD1 variants somehow undermine the relevance of these findings.

Response: We thank the reviewer for accepting our validation of CHD1-interacting chromatin factors.

10 (continued). In addition, the colocalization analyses by MCC lack the appropriate statistical tests to support the conclusions. Indeed, by considering as n the number of analyzed cells, they measured the cell-to-cell variability among the two conditions, but they did not tested the null hypothesis for which they need to consider the number of replicates. As in this case they have two biological replicates, they have to increase the numerosity of independent biological replicates before applying the appropriate nonparametric statistical test.

Response: We thank the reviewer for this helpful comment regarding the colocalization analysis using MCC. In response to the feedback, we have reanalyzed the data by adding additional 10 cells per condition. This brings the total number of analyzed cells to more than 40, across three independent biological replicates. We have applied the appropriate nonparametric statistical tests, taking into account the number of independent biological replicates. The revised data, along with the updated statistical analysis, are now presented in Figure 8d. Additionally, we have updated the Methods section to describe the revised statistical approach used for this analysis.

d
Figure 8d, Confocal images showing the colocalization of CHD1 and ASH2L, SUZ12, SC35, H3K4me3, or RNA polymerase II phosphorylated on serine 5 [Pol2(pS5)] in HeLa cells. Scale bar: 5 μ m. Colocalized fractions of CHD1 with indicated proteins were quantified based on Manders' coefficient. ASH2L and SUZ12 are recruited into CHD1 condensates. SC35, a splicing factor involved in mRNA maturation, is not recruited. H3K4me3 and Pol2(pS5) are known to interact with CHD1. Data are means \pm SEM of more than forty cells in at three independent experiments. Mann-Whitney test.

Point-By-Point Response to the Reviewer#3's Comments

Reviewer #3 (Remarks to the Author):

Although the author has not fully addressed all raised concerns, I acknowledge their effort in responding to the criticisms. That said, I remain convinced that the following points require further discussion or revision:

Protein Abundance and Droplet Size: The relationship between protein abundance and droplet size has not been experimentally addressed. Presenting non-quantitative data is insufficient to resolve this issue. The author should include a discussion section acknowledging this limitation. Importantly, no one compelled the author to rely on transient overexpression, which they use to justify inconsistencies in their data, when a more rigorous approach—such as deriving stable single clones—would have allowed for proper analysis.

Response: We thank the reviewer for providing feedback aimed at strengthening our manuscript. To address the limitation regarding the relationship between protein abundance and droplet size, we have added the following text to the Discussion: “We were unable to pinpoint the biophysical mechanisms behind CHD1 condensation in cells, such as how protein concentration influences condensate size or which specific IDR sequences drive condensation. While *in vitro* data show a concentration-dependent change in condensate size (Extended Data Fig. 2d, e), accurately assessing this relationship in cells remains challenging due to biological variability and technical constraints. Additionally, we relied on transient overexpression to qualitatively assess CHD1 mutant condensate formation, limiting quantitative comparisons of condensate size under physiological conditions in cells.”

We have included new data (Extended Data Fig. 2e) from an *in vitro* droplet assay to quantitatively demonstrate the relationship between protein abundance and droplet size under simplified conditions.

Extended Data Fig. 2e, The occupancy (left) and average size (right) of CHD1^{WT}-Venus condensates in Droplet A buffer containing 150 mM NaCl were quantified within a $4.51 \times 10^{-2} \text{ mm}^2$ area. Three fields from independent sessions were randomly selected for analysis.

We would like to clarify that the transient overexpression system was used to assess whether CHD1 and its mutants are capable of forming condensates, rather than to quantify droplet size under physiological conditions. To clearly indicate that these experiments were conducted under overexpression conditions, we have revised the main text (line 118) as follows: “In line with *in vitro* findings, cellular overexpression of exogenous CHD1^{WT} and variants revealed that both the N- and C-terminal IDRs are necessary for the nuclear condensation of CHD1, with the C-terminal IDR playing a more significant role (Fig. 2b)”. Additionally, we have included protein expression data showing that the CHD1-Venus variants are expressed at comparable levels to each other, but are elevated relative to endogenous CHD1 (see new Fig. 2b, right panels, below).

Fig. 2b, Western blot analysis performed using the Wes system (ProteinSimple), with anti-Vinculin (VINC) as the loading control (right). The CHD1-Venus variants were expressed at comparable levels to each other, and all exhibit elevated expression compared to endogenous CHD1 (indicated by the arrow). Note that the CHD1 antibody targets a C-terminal epitope, meaning it does not detect the CHD1^{ΔC} or CHD1^{ΔNC} variants.

We would like to emphasize that quantification of condensate size in cells is both technically challenging and biologically confounded even in single clone. In living cells, condensate size is influenced not only by protein concentration but also by other variables like transcriptional activity, molecular crowding, and nuclear interactions as demonstrated in Cell 2021 [<https://doi.org/10.1016/j.cell.2020.11.030>]. Consistent with this, our data in Fig. 2d show that endogenous CHD1 forms heterogeneous-size condensates, likely reflecting diverse regulatory inputs within the chromatin environment. From a technical standpoint, accurate *in vivo* size estimation is hampered by variability in imaging settings, such as laser power, pinhole size, detector gain/sensitivity, spatial resolution, and the point spread function, which can cause condensates to appear larger than their true size. To address this, we quantified regions above a fluorescence intensity threshold in Extended Data Fig. 6, minimizing arbitrary size classifications. In contrast, our *in vitro* assays using bright-field microscopy (Fig. 3, Extended Data Fig. 2) allow for more accurate, reproducible measurements of droplet size without these technical and biological confounders.

Importantly, our conclusion that CHD1 forms condensates is based not only on overexpression data but also on *in vitro* droplet assays (Fig. 2, 3, Extended Data Fig. 2) and endogenous observations using Venus knock-in CHD1 (Fig. 2d), all of which support the formation of condensates under near-physiological conditions. Furthermore, we demonstrate that CHD1 condensates are distinct from heat shock-induced aggregates (Extended Data Fig. 4), reinforcing their specificity.

Biophysical Properties of Condensates: As the author has not adequately determined the biophysical properties of the condensates, all references to liquid-liquid phase separation (LLPS) should be removed as agreed with authors. Additionally, the manuscript must explicitly state as a limitation that the biophysical mechanisms driving condensate formation could not be determined.

Response: We have removed all references to LLPS in the manuscript, as per your request. We also acknowledge in the revised Discussion section that the biophysical mechanisms behind condensate formation could not be fully determined, as follows: “We were unable to pinpoint the biophysical mechanisms behind CHD1 condensation in cells, such as how protein concentration influences condensate size or which specific IDR sequences drive condensation. While *in vitro* data show a concentration-dependent change in condensate size (Extended Data Fig. 2d, e), accurately assessing this relationship in cells remains challenging due to biological variability and technical constraints. Additionally, we relied on transient overexpression to qualitatively assess CHD1 mutant condensate formation, limiting quantitative comparisons of condensate size under physiological conditions in cells.”

CHD Protein Droplet Assays: The decision to not quantify droplet assays for the other CHD proteins suggests that the data lack consistency. Arguing that this is the first report of their ability to form condensates is not a valid justification for presenting poorly controlled experiments. Scientific conclusions must be based on robust, well-controlled data. Unless proper quantification is performed, these data should be removed. By addressing these concerns transparently and acknowledging the study's limitations, the manuscript will be significantly strengthened.

Response: We appreciate your feedback. In response, we have removed the droplet assays for the other CHD proteins from the revised manuscript. We believe this change enhances the clarity and rigor of our study.

Point-By-Point Response to the Reviewer#3's Comments

Reviewer #3 (Remarks to the Author):

The authors properly addressed all the raised points by including new data and/or by commenting on some intrinsic limitations of the study, as suggested. I am fully satisfied with the revised manuscript, and I would like to thank the authors for acknowledging the relevance of the raised points and for addressing them accordantly.

Response: We thank the reviewer for the careful evaluation and insightful feedback. We appreciate the recognition of our revisions and are glad the manuscript now meets the reviewer's expectations.